# Langevin Monte Carlo for strongly log-concave distributions: Randomized mid-point revisited

**Lu Yu**
CREST, ENSAE, IP Paris
lu.yu@ensae.fr

**Avetik Karagulyan**
KAUST
avetik.karagulyan@kaust.edu.sa

**Arnak Dalalyan**
CREST, ENSAE, IP Paris
arnak.dalalyan@ensae.fr

## Abstract

We revisit the problem of sampling from a target distribution that has a smooth strongly log-concave density everywhere in $\mathbb{R}^p$. In this context, if no additional density information is available, the randomized midpoint discretization for the kinetic Langevin diffusion is known to be the most scalable method in high dimensions with large condition numbers. Our main result is a nonasymptotic and easy to compute upper bound on the $W_2$-error of this method. To provide a more thorough explanation of our method for establishing the computable upper bound, we conduct an analysis of the midpoint discretization for the vanilla Langevin process. This analysis helps to clarify the underlying principles and provides valuable insights that we use to establish an improved upper bound for the kinetic Langevin process with the midpoint discretization. Furthermore, by applying these techniques we establish new guarantees for the kinetic Langevin process with Euler discretization, which have a better dependence on the condition number than existing upper bounds.

## 1 Introduction

The task of sampling from target distributions with smooth, strongly log-concave densities has been a long-standing challenge in various fields such as statistics, machine learning, and computational physics (Andrieu et al., 2003; Krauth, 2006; Andrieu et al., 2010). Over the years, researchers have developed several algorithms to tackle this problem, and one prominent approach are the Langevin algorithms (Rogers & Williams, 2000; Oksendal, 2013; Robert et al., 1999). Langevin algorithms leverage the Langevin equation to design efficient and effective sampling algorithms. These methods generate a Markov chain by iteratively updating the position of a particle based on the Langevin equation. By simulating the particle's motion over time, these algorithms explore the target distribution and eventually converge to samples that approximate the desired distribution (Robert et al., 1999).

The canonical sampling algorithm, Langevin Monte Carlo (LMC) (Roberts & Tweedie, 1996; Dalalyan, 2017; Durmus & Moulines, 2017; Erdogdu & Hosseinzadeh, 2021; Mousavi-Hosseini et al., 2023; Raginsky et al., 2017; Erdogdu et al., 2018; Mou et al., 2022; Erdogdu et al., 2022), is a Markov chain Monte Carlo (MCMC) method that simulates the dynamics of a fictitious particle moving through a potential energy landscape defined by the target distribution. Formally, it is the Euler-Maruyama discretization of an SDE known as the Langevin diffusion. The underlying idea can be traced back to the early 20th century when Paul Langevin introduced a stochastic differential equation (SDE) to describe the motion of a particle in a fluid (Langevin, 1908). This SDE, combines deterministic and random components to model the particle's behavior under the influence of both a deterministic force and random noise.

One popular variant of the Langevin Monte Carlo is based on discretizing the kinetic Langevin diffusion, which introduces a friction term to control the exploration-exploitation trade-off during the sampling process (Einstein, 1905; Von Smoluchowski, 1906). According to Nelson (1967), the Langevin diffusion is the rescaled limit of the kinetic Langevin diffusion. Its ergodicity and mixing-time properties are studied in Eberle et al. (2019); Dalalyan & Riou-Durand (2020). Euler-Maruyama time discretization of this SDE, called kinetic Langevin Monte Carlo (KLMC), is prevalent in the sampling literature (Cheng et al., 2018b; Dalalyan & Riou-Durand, 2020; Shen & Lee, 2019; Ma et al., 2021; Zhang et al., 2023).

The randomized midpoint discretization method, as an alternative to the Euler-Maruyama scheme for KLMC, is proposed by Shen & Lee (2019). They demonstrate the superior performance of this method in terms of both tolerance and condition number dependency. More recently, He et al. (2020) analyze probabilistic properties of the randomized midpoint discretization method for the (kinetic) Langevin diffusion. In this work, we undertake a comprehensive and thorough analysis of the randomized midpoint discretization scheme for the kinetic Langevin diffusion under strongly log-concavity. To achieve this, we introduce a novel proof technique relying on summation by part, which helps to establish improved non-asymptotic and computable upper bounds on the discretization error for this method. Our contributions can be summarized as follows.

- To lay the groundwork for our analysis, we initially delve into the midpoint discretization technique applied to the vanilla Langevin process. In this context, we introduce our novel proof technique, which plays a pivotal role in our study. Notably, in Theorem 1, we provide the convergence guarantees for RLMC in $W_2$-distance with explicit constants and a transparent reliance on the initialization. These guarantees are competitive with the best available results for LMC, and could be leveraged to derive an improved upper bound specifically tailored for RKLMC.

- We further extend these techniques to RKLMC, and provide the corresponding convergence guarantees in $W_2$-distance in Theorem 2. Compared to the previous works, our bound **a)** contains small constants and the explicit dependence on the initialization, **b)** does not require the initialization to be at the minimizer of the potential, **c)** and is free from the linear dependence on the sample size, which serves as a crucial step towards the method applied to non-convex potentials.

- Employing the same techniques, we finally examine the convergence behavior of the KLMC algorithm with the Euler-Maruyama discretization. In Theorem 3, we provide an upper bound on the accuracy of this scheme in $W_2$-distance with improved dependence on the condition number.

We offer a systematic and unified treatment of the variants of LMC, which empowers us to derive enhanced upper bounds for the $W_2$-error associated with RKLMC, RLMC, and KLMC algorithms. Furthermore, our techniques facilitate the determination of explicit constants and the dependence on initialization, providing us with a clearer basis for choosing the step size and comparing the convergence rates across these methods.

**Notation.** Denote the $p$-dimensional Euclidean space by $\mathbb{R}^p$. The letter $\boldsymbol{\theta}$ denotes the deterministic vector and its calligraphic counterpart $\boldsymbol{\vartheta}$ denotes the random vector. We use $\mathbf{I}_p$ and $\mathbf{0}_p$ to denote, respectively, the $p \times p$ identity and zero matrices. Define the relations $\mathbf{A} \preccurlyeq \mathbf{B}$ and $\mathbf{B} \succcurlyeq \mathbf{A}$ for two symmetric $p \times p$ matrices $\mathbf{A}$ and $\mathbf{B}$ to mean that $\mathbf{B} - \mathbf{A}$ is positive semi-definite. The gradient and the Hessian of a function $f : \mathbb{R}^p \to \mathbb{R}$ are denoted by $\nabla f$ and $\nabla^2 f$, respectively. Given any pair of measures $\mu$ and $\nu$ defined on $(\mathbb{R}^p, \mathcal{B}(\mathbb{R}^p))$, the Wasserstein-2 distance between $\mu$ and $\nu$ is defined as

$$W_2(\mu, \nu) = \left( \inf_{\varrho \in \Gamma(\mu, \nu)} \int_{\mathbb{R}^p \times \mathbb{R}^p} \|\boldsymbol{\theta} - \boldsymbol{\theta}'\|_2^2 \, \mathrm{d}\varrho(\boldsymbol{\theta}, \boldsymbol{\theta}') \right)^{1/2},$$

where the infimum is taken over all joint distributions $\varrho$ that have $\mu$ and $\nu$ as marginals.

## 2 UNDERSTANDING THE RANDOMIZED MIDPOINT DISCRETIZATION: THE VANILLA LANGEVIN DIFFUSION

The goal is to sample a random vector in $\mathbb{R}^p$ according to a given distribution $\pi$ of the form

$$\pi(\boldsymbol{\theta}) \propto \exp\{-f(\boldsymbol{\theta})\}, \qquad \boldsymbol{\theta} \in \mathbb{R}^p,$$

with a function $f : \mathbb{R}^p \to \mathbb{R}$, referred to as the potential. Throughout the paper, we assume that the potential function $f$ is $M$-smooth and $m$-strongly convex for some constants $0 < m \leqslant M < \infty$.

**Assumption 1.** *The function $f : \mathbb{R}^p \to \mathbb{R}$ is twice differentiable, and its Hessian matrix $\nabla^2 f$ satisfies*

$$m\mathbf{I}_p \preccurlyeq \nabla^2 f(\boldsymbol{\theta}) \preccurlyeq M\mathbf{I}_p, \qquad \forall \boldsymbol{\theta} \in \mathbb{R}^p.$$

Let $\boldsymbol{\vartheta}_0$ be a random vector drawn from a distribution $\nu$ on $\mathbb{R}^p$ and let $\boldsymbol{W} = (\boldsymbol{W}_t : t \geqslant 0)$ be a $p$-dimensional Brownian motion independent of $\boldsymbol{\vartheta}_0$. Using the potential $f$, the random variable $\boldsymbol{\vartheta}_0$ and the process $\boldsymbol{W}$, one can define the stochastic differential equation

$$d\boldsymbol{L}_t^{\mathsf{LD}} = -\nabla f(\boldsymbol{L}_t^{\mathsf{LD}})\, dt + \sqrt{2}\, d\boldsymbol{W}_t, \qquad t \geqslant 0, \qquad \boldsymbol{L}_0^{\mathsf{LD}} = \boldsymbol{\vartheta}_0. \tag{1}$$

This equation has a unique strong solution, which is a continuous-time Markov process, termed Langevin diffusion. Under some further assumptions on $f$, such as strong convexity or dissipativity, the Langevin diffusion is ergodic, geometrically mixing and has $\pi$ as its unique invariant distribution (Bhattacharya, 1978). Furthermore, the mixing properties of this process can be quantified. For instance, if $\pi$ satisfies the Poincaré inequality with constant $C_{\mathsf{P}}$, then (see e.g. Chewi et al. (2020)) the distribution $\nu_t^{\mathsf{LD}}$ of $\boldsymbol{L}_t^{\mathsf{LD}}$ satisfies

$$\mathsf{W}_2(\nu_t^{\mathsf{LD}}, \pi) \leqslant e^{-t/C_{\mathsf{P}}} \sqrt{2 C_{\mathsf{P}} \chi^2(\nu \| \pi)}, \qquad \forall t \geqslant 0.$$

These results suggest that we can sample from the distribution $\pi$ by using a suitable discretization of the Langevin diffusion. The Langevin Monte Carlo (LMC) method is based on this idea, combining the aforementioned considerations with the Euler discretization. Specifically, for small values of $h \geqslant 0$ and $\Delta_h \boldsymbol{W}_t = \boldsymbol{W}_{t+h} - \boldsymbol{W}_t$, the following approximation holds

$$\boldsymbol{L}_{t+h}^{\mathsf{LD}} = \boldsymbol{L}_t^{\mathsf{LD}} - \int_0^h \nabla f(\boldsymbol{L}_{t+s}^{\mathsf{LD}})\, ds + \sqrt{2}\, \Delta_h \boldsymbol{W}_t \approx \boldsymbol{L}_t^{\mathsf{LD}} - h\nabla f(\boldsymbol{L}_t^{\mathsf{LD}}) + \sqrt{2}\, \Delta_h \boldsymbol{W}_t.$$

By repeatedly applying this approximation with a small step-size $h$, we can construct a Markov chain $(\boldsymbol{\vartheta}_k^{\mathsf{LMC}} : k \in \mathbb{N})$ that converges to the target distribution $\pi$ as $h$ goes to zero. More precisely, $\boldsymbol{\vartheta}_k^{\mathsf{LMC}} \approx \boldsymbol{L}_{kh}^{\mathsf{LD}}$, for $k \in \mathbb{N}$, is given by

$$\boldsymbol{\vartheta}_{k+1}^{\mathsf{LMC}} = \boldsymbol{\vartheta}_k^{\mathsf{LMC}} - h\nabla f(\boldsymbol{\vartheta}_k^{\mathsf{LMC}}) + \sqrt{2}\, (\boldsymbol{W}_{(k+1)h} - \boldsymbol{W}_{kh}).$$

This method is computationally efficient and has been widely used in statistics and machine learning for sampling from high-dimensional distributions (Gal & Ghahramani, 2016; Izmailov et al., 2020; 2021). To assess the discretization error, consider the case where $\boldsymbol{L}_0^{\mathsf{LD}}$ is drawn from the invariant distribution $\pi$ and note that

$$\boldsymbol{L}_{(k+1)h}^{\mathsf{LD}} - \boldsymbol{\vartheta}_{k+1}^{\mathsf{LMC}} = \boldsymbol{L}_{kh}^{\mathsf{LD}} - \boldsymbol{\vartheta}_k^{\mathsf{LMC}} - \int_0^h \nabla f(\boldsymbol{L}_{kh+s}^{\mathsf{LD}})\, ds + h\nabla f(\boldsymbol{\vartheta}_k^{\mathsf{LMC}})$$

$$= \boldsymbol{L}_{kh}^{\mathsf{LD}} - \boldsymbol{\vartheta}_k^{\mathsf{LMC}} - h\big(\nabla f(\boldsymbol{L}_{kh}^{\mathsf{LD}}) - \nabla f(\boldsymbol{\vartheta}_k^{\mathsf{LMC}})\big) - \boldsymbol{\zeta}_k, \tag{2}$$

where $\boldsymbol{\zeta}_k = \int_0^h \big(\nabla f(\boldsymbol{L}_{kh+s}^{\mathsf{LD}}) - \nabla f(\boldsymbol{L}_{kh}^{\mathsf{LD}})\big)\, ds$ is a zero-mean random "noise" vector. Previous work on LMC demonstrated that the squared $\mathbb{L}_2$ norm of $\boldsymbol{\zeta}_k$ is of order $M^2 h^3 p$, whereas the term $\boldsymbol{L}_{kh}^{\mathsf{LD}} - \boldsymbol{\vartheta}_k^{\mathsf{LMC}} - h\big(\nabla f(\boldsymbol{L}_{kh}^{\mathsf{LD}}) - \nabla f(\boldsymbol{\vartheta}_k^{\mathsf{LMC}})\big)$ satisfies the contraction inequality

$$\big\| \boldsymbol{L}_{kh}^{\mathsf{LD}} - \boldsymbol{\vartheta}_k^{\mathsf{LMC}} - h\big(\nabla f(\boldsymbol{L}_{kh}^{\mathsf{LD}}) - \nabla f(\boldsymbol{\vartheta}_k^{\mathsf{LMC}})\big) \big\|_{\mathbb{L}_2}^2 \leqslant (1 - mh)^2 \| \boldsymbol{L}_{kh}^{\mathsf{LD}} - \boldsymbol{\vartheta}_k^{\mathsf{LMC}} \|_{\mathbb{L}_2}^2. \tag{3}$$

If we denote by $r_k$ the correlation between $\boldsymbol{\zeta}_k$ and $\boldsymbol{L}_{kh}^{\mathsf{LD}} - \boldsymbol{\vartheta}_k^{\mathsf{LMC}}$, and by $\mathrm{Err}_k$ the error $\| \boldsymbol{L}_{kh}^{\mathsf{LD}} - \boldsymbol{\vartheta}_k^{\mathsf{LMC}} \|_{\mathbb{L}_2}$, we infer from equation 2 and equation 3 that

$$\mathrm{Err}_{k+1}^2 \leqslant (1 - mh)^2 \mathrm{Err}_k^2 + CMhr_k \mathrm{Err}_k \sqrt{hp} + CM^2 h^3 p,$$

for some universal constant $C$. If we were able to check that $r_k$ is small enough so that the second term of the right-hand side can be neglected, we would get $\mathrm{Err}_{k+1}^2 \leqslant (1 - mh)^2 \mathrm{Err}_k^2 + C_1 M^2 h^3 p$, which would eventually lead to $\mathrm{Err}_{k+1}^2 \leqslant (1 - mh)^{2k} \mathrm{Err}_1^2 + C_2 M^2 h^2 (p/m)$. This would amount to

$$|r_k| \ll 1 \qquad \Longrightarrow \qquad \mathrm{Err}_{k+1} \leqslant (1 - mh)^k \mathrm{Err}_1 + C_3 Mh\sqrt{p/m}. \tag{4}$$

Unfortunately, without any additional conditions on $f$, the correlation $r_k$ cannot be shown to be small, and one can only deduce from equation 3 that $\text{Err}_{k+1} \leqslant (1 - mh)\text{Err}_k + CMh\sqrt{ph}$, which eventually yields

$$|r_k| \lll 1 \qquad \implies \qquad \text{Err}_{k+1} \leqslant (1 - mh)^k \text{Err}_1 + C_4(M/m)\sqrt{ph}. \tag{5}$$

This inequality is established under the standard assumption $Mh \leqslant 1$, which implies that the last term in equation 4 is significantly smaller than equation 5. To get such an error deflation, we need the correlations $r_k$ to be small. While this is not guaranteed for the Euler discretization, we will see that the randomized midpoint method allows us to achieve such a reduction.

Let $U$ be a random variable uniformly distributed in $[0, 1]$ and independent of the Brownian motion $\boldsymbol{W}$. The randomized midpoint method exploits the approximation

$$\boldsymbol{L}_{t+h}^{\mathsf{LD}} = \boldsymbol{L}_t^{\mathsf{LD}} - \int_0^h \nabla f(\boldsymbol{L}_{t+s}^{\mathsf{LD}})\,\mathrm{d}s + \sqrt{2}\,\Delta_h \boldsymbol{W}_t \approx \boldsymbol{L}_t^{\mathsf{LD}} - h\nabla f(\boldsymbol{L}_{t+hU}^{\mathsf{LD}}) + \sqrt{2}\,\Delta_h \boldsymbol{W}_t.$$

The noise counterpart of $\boldsymbol{\zeta}_k$ in this case is $\boldsymbol{\zeta}_k^{\mathsf{R}} = \int_0^h \nabla f(\boldsymbol{L}_{kh+s}^{\mathsf{LD}})\,\mathrm{d}s - \nabla f(\boldsymbol{L}_{kh+Uh}^{\mathsf{LD}})$. It is clearly centered and uncorrelated with all the random vectors independent of $U$ such as $\boldsymbol{L}_{kh}^{\mathsf{LD}}$, $\boldsymbol{\vartheta}_k^{\mathsf{LMC}}$ and the gradient of $f$ evaluated at these points.

The explanation above provides the intuition of the randomized midpoint method, and a hint to why it is preferable to the Euler discretization, but it cannot be taken as a formal definition of the method. The formal definition of the randomized midpoint method for the Langevin Monte Carlo (RLMC) is defined as follows: at each iteration $k = 1, 2, \ldots$,

1. we randomly, and independently of all the variables generated during the previous steps, generate a pair of random vectors $(\boldsymbol{\xi}_k', \boldsymbol{\xi}_k'')$ and a random variable $U_k$ such that
   - $U_k$ is uniformly distributed in $[0, 1]$ and independent of $(\boldsymbol{\xi}_k', \boldsymbol{\xi}_k'')$,
   - $(\boldsymbol{\xi}_k', \boldsymbol{\xi}_k'')$ are independent $\mathcal{N}_p(0, \mathbf{I}_p)$.
2. we set $\boldsymbol{\xi}_k = \sqrt{U_k}\,\boldsymbol{\xi}_k' + \sqrt{1 - U_k}\,\boldsymbol{\xi}_k''$ and define the $(k+1)$th iterate $\boldsymbol{\vartheta}^{\mathsf{RLMC}}$ by

$$\boldsymbol{\vartheta}_{k+U}^{\mathsf{RLMC}} = \boldsymbol{\vartheta}_k^{\mathsf{RLMC}} - hU_k\nabla f(\boldsymbol{\vartheta}_k^{\mathsf{RLMC}}) + \sqrt{2hU_k}\,\boldsymbol{\xi}_k' \tag{6}$$

$$\boldsymbol{\vartheta}_{k+1}^{\mathsf{RLMC}} = \boldsymbol{\vartheta}_k^{\mathsf{RLMC}} - h\nabla f(\boldsymbol{\vartheta}_{k+U}^{\mathsf{RLMC}}) + \sqrt{2h}\,\boldsymbol{\xi}_k. \tag{7}$$

With a small step-size $h$ and a large number of iterations $n$, the distribution of $\boldsymbol{\vartheta}_n^{\mathsf{RLMC}}$ can closely approximate the target distribution $\pi$. In a smooth and strongly convex setting, it is even possible to obtain a reliable estimate of the sampling error, as demonstrated in the following theorem (the the proof is included in the supplementary material).

If the step-size $h$ is small and the number of iterations $n$ is large, the distribution of $\boldsymbol{\vartheta}_n^{\mathsf{RLMC}}$ is close to the target $\pi$. Interestingly, in the smooth and strongly convex setting it is possible to get a good evaluation of the error of sampling as shown in the next theorem (the proof is deferred to the supplementary material).

**Theorem 1.** *Assume the function $f : \mathbb{R}^p \to \mathbb{R}$ satisfies Assumption 1. Let $h$ be such that $Mh + \sqrt{\kappa}\,(Mh)^{3/2} \leqslant 1/4$ with $\kappa = M/m$. Then, every $n \geqslant 1$, the distribution $\nu_n^{\mathsf{RLMC}}$ of $\boldsymbol{\vartheta}_n^{\mathsf{RLMC}}$ satisfies*

$$\mathsf{W}_2(\nu_n^{\mathsf{RLMC}}, \pi) \leqslant 1.11 e^{-mnh/2}\mathsf{W}_2(\nu_0, \pi) + \big(2.4\sqrt{\kappa Mh} + 1.77\big)Mh\sqrt{p/m}. \tag{8}$$

Prior to discussing the relation of the above error estimate to those available in the literature, let us state a consequence of it.

**Corollary 1.** *Let $\varepsilon \in (0, 1)$ be a small number. If we choose $h > 0$ and $n \in \mathbb{N}$ so that*

$$Mh = \frac{\varepsilon}{1.5 + (6.5\kappa\varepsilon)^{1/3}}, \quad and \quad n \geqslant \left(\frac{3\kappa}{\varepsilon} + \frac{3.8\kappa^{4/3}}{\varepsilon^{2/3}}\right)\left(\log(20/\varepsilon) + \frac{1}{2}\log\left(\frac{m}{p}\mathsf{W}_2^2(\nu_0, \pi)\right)\right)$$

*then[1] we have $\mathsf{W}_2(\nu_n^{\mathsf{RLMC}}, \pi) \leqslant \varepsilon\sqrt{p/m}$.*

---

[1]This follows from the fact that $(6\kappa/\varepsilon) + 4.2\kappa^{4/3}/\varepsilon^{2/3} \leqslant 2\kappa/(Mh) = 2/mh$.

Our results can be compared to the best available results for the Langevin Monte Carlo (LMC) under Assumption 1 (Durmus et al., 2019, Eq. 22). We recall that LMC is defined by a recursive relation of the same form as equation 7, with the only difference that $\nabla f(\vartheta_{k+U})$ is replaced by $\nabla f(\vartheta_k)$. The tightest known bound for LMC is given by

$$\mathsf{W}_2(\nu_n^{\mathsf{LMC}}, \pi) \leqslant (1 - mh)^{-n/2} \mathsf{W}_2(\nu_0, \pi) + \sqrt{2Mhp/m},$$

with $Mh \leqslant 1$. By choosing $2Mh = (19/20)^2 \varepsilon^2$ and

$$n \geqslant 2.22(\kappa/\varepsilon^2)\left\{ \log(20/\varepsilon) + \tfrac{1}{2} \log\left( \tfrac{m}{p} \mathsf{W}_2^2(\nu_0, \pi) \right) \right\},$$

we can ensure that $\mathsf{W}_2(\nu_n^{\mathsf{LMC}}, \pi) \leqslant \varepsilon\sqrt{p/m}$. Therefore, the complexity bound of Corollary 1 derived from our result for RLMC is better than the best-known complexity bound for LMC in the regime of $\kappa$ of smaller order than[2] $\varepsilon^{-4}$.

To the best of our knowledge, the first results on the error analysis of RLMC have been obtained in (He et al., 2020). They derived an upper bound on the discretization error (the second term on the right-hand side of equation 8) under the assumption that the initial point of the algorithm is the minimizer of the potential function $f$. Their bound takes the form $C(\sqrt{\kappa Mh}+1)Mh\sqrt{p/m}\times\sqrt{mnh}$, where $C$ is a universal but unspecified constant. Compared to our bound, the one obtained in He et al. (2020) has an additional factor $\sqrt{mnh}$. While this factor may not be very harmful in the case of geometric ergodicity where the number of iterations $n$ is chosen such that $nmh$ goes to infinity at the logarithmic rate $\log(1/\varepsilon)$, removing it can be an important step toward extending these results to potentials that are not strongly convex.

While the proof of this theorem is deferred to the supplementary material, we can outline the main argument that allowed us to remove the factor $\sqrt{nmh}$ from the error bound. To convey the main idea, let us consider three positive sequences $a_n, b_n, c_n$ satisfying, for every $n \in \mathbb{N}$,

$$a_{n+1} \leqslant (1 - \alpha)a_n + b_n \tag{9}$$
$$c_{n+1} \leqslant c_n - b_n + \mathsf{C}, \tag{10}$$

with some $\alpha \in (0, 1)$ and $\mathsf{C} > 0$. Using the standard telescoping sums argument, frequently employed for proving the convergence of convex optimization algorithms, one can infer from equation 10 that

$$\sum_{k=0}^{n} b_n \leqslant c_0 - c_{n+1} + n\mathsf{C} \leqslant c_0 + n\mathsf{C}. \tag{11}$$

On the other hand, it follows from equation 9 that

$$a_{n+1} \leqslant (1 - \alpha)^{n+1}a_0 + \sum_{k=0}^{n}(1 - \alpha)^{n-k}b_k. \tag{12}$$

Upper bounding $(1 - \alpha)^{n-k}$ by one, and using equation 11, we arrive at

$$a_{n+1} \leqslant (1 - \alpha)^{n+1}a_0 + c_0 + n\mathsf{C}. \tag{13}$$

This type of argument, used in previous papers on RKLMC (Shen & Lee, 2019), is sub-optimal and leads to the extra factor $\sqrt{nmh}$. A tighter bound can be obtained by replacing the telescoping sum argument by the summation by parts. More precisely, one can check that equation 10 and equation 12 yield

$$a_{n+1} \leqslant (1 - \alpha)^{n+1}a_0 + \sum_{k=0}^{n}(1 - \alpha)^{n-k}(c_k - c_{k+1}) + \mathsf{C}\sum_{k=0}^{n}(1 - \alpha)^{n-k}$$
$$\leqslant (1 - \alpha)^{n+1}a_0 + (1 - \alpha)^n c_0 + \alpha\sum_{k=0}^{n}(1 - \alpha)^{n-k}c_k + \frac{\mathsf{C}}{\alpha}. \tag{14}$$

The upper bound provided by equation 14 has two advantages as compared to equation 13: the term $n\mathsf{C}$ is replaced by $\mathsf{C}/\alpha$, which is generally smaller, and the dependence on the initial value is $(1 - \alpha)^n c_0$ instead of $c_0$. This comes also with a challenge consisting in upper bounding the sum present in the right-hand side of equation 14, which we managed to overcome using the strong convexity (or, more precisely, the Polyak-Lojasiewicz condition). The full details are deferred to the supplementary material.

---

[2]The condition $\kappa = o(\varepsilon^{-4})$ is obtained by simple algebra from the condition $\kappa^{4/3}/\varepsilon^{2/3} = o(\kappa/\varepsilon^2)$.

## 3  RANDOMIZED MIDPOINT METHOD FOR THE KINETIC LANGEVIN DIFFUSION

The randomized midpoint method, introduced and studied in Shen & Lee (2019), aims at providing a discretization of the kinetic Langevin process that reduces the bias of sampling as compared to more conventional discretizations. Recall that the kinetic Langevin process $\boldsymbol{L}^{\mathsf{KLD}}$ is a solution to a second-order stochastic differential equation that can be informally written as

$$\frac{1}{\gamma}\ddot{\boldsymbol{L}}_t^{\mathsf{KLD}} + \dot{\boldsymbol{L}}_t^{\mathsf{KLD}} = -\nabla f(\boldsymbol{L}_t^{\mathsf{KLD}}) + \sqrt{2}\,\dot{\boldsymbol{W}}_t, \tag{15}$$

with initial conditions $\boldsymbol{L}_0^{\mathsf{KLD}} = \boldsymbol{\vartheta}_0$ and $\dot{\boldsymbol{L}}_0^{\mathsf{KLD}} = \boldsymbol{v}_0$. In equation 15, $\gamma > 0$, $\boldsymbol{W}$ is a standard $p$-dimensional Brownian motion and dots are used to designate derivatives with respect to time $t \geqslant 0$. This can be formalized using Itô's calculus and introducing the velocity field $\boldsymbol{V}^{\mathsf{KLD}}$ so that the joint process $(\boldsymbol{L}^{\mathsf{KLD}}, \boldsymbol{V}^{\mathsf{KLD}})$ satisfies

$$\mathrm{d}\boldsymbol{L}_t^{\mathsf{KLD}} = \boldsymbol{V}_t^{\mathsf{KLD}}\,\mathrm{d}t; \quad \frac{1}{\gamma}\mathrm{d}\boldsymbol{V}_t^{\mathsf{KLD}} = -\big(\boldsymbol{V}_t^{\mathsf{KLD}} + \nabla f(\boldsymbol{L}_t^{\mathsf{KLD}})\big)\,\mathrm{d}t + \sqrt{2}\,\mathrm{d}\boldsymbol{W}_t. \tag{16}$$

Similar to the vanilla Langevin diffusion equation 1, the kinetic Langevin diffusion $(\boldsymbol{L}^{\mathsf{KLD}}, \boldsymbol{V}^{\mathsf{KLD}})$ is a Markov process that exhibits ergodic properties when the potential $f$ is strongly convex (see (Eberle et al., 2019) and references therein). The invariant density of this process is given by

$$p_*(\boldsymbol{\theta},\boldsymbol{v}) \propto \exp\{-f(\boldsymbol{\theta}) - \tfrac{1}{2\gamma}\|\boldsymbol{v}\|^2\}, \qquad \text{for all} \quad \boldsymbol{\theta},\boldsymbol{v} \in \mathbb{R}^p.$$

Note that the marginal of $p_*$ corresponds to $\boldsymbol{\theta}$ coincides with the target density $\pi$. However, unlike the vanilla Langevin diffusion, the kinetic Langevin is not reversible. It is interesting to note that the distribution of the process $\boldsymbol{L}^{\mathsf{KLD}}$ approaches that of the vanilla Langevin process as $\gamma$ approaches infinity (see e.g. (Nelson, 1967)). Therefore, $\boldsymbol{L}^{\mathsf{LD}}$ and $\boldsymbol{L}^{\mathsf{KLD}}$ are often referred to as the overdamped and underdamped Langevin processes, respectively (where increasing the friction parameter $\gamma$ is characterized as damping).

The kinetic Langevin diffusion $\boldsymbol{L}^{\mathsf{KLD}}$ is particularly attractive for sampling because its distribution $\nu_t^{\mathsf{KLD}}$ converges to the invariant distribution exponentially fast. This is especially true for strongly convex potentials, as proven in[3] (Dalalyan & Riou-Durand, 2020, Prop. 1), where it is shown that the following inequality holds:

$$\mathsf{W}_2\left(\mathbf{C}\begin{bmatrix}\boldsymbol{V}_t^{\mathsf{KLD}}\\\boldsymbol{L}_t^{\mathsf{KLD}}\end{bmatrix}, \mathbf{C}\begin{bmatrix}\boldsymbol{v}\\\boldsymbol{\vartheta}\end{bmatrix}\right) \leqslant e^{-mt}\mathsf{W}_2\left(\mathbf{C}\begin{bmatrix}\boldsymbol{V}_0\\\boldsymbol{L}_0\end{bmatrix}, \mathbf{C}\begin{bmatrix}\boldsymbol{v}\\\boldsymbol{\vartheta}\end{bmatrix}\right), \quad \mathbf{C} = \begin{bmatrix}\mathbf{I}_p & \mathbf{0}_p\\\mathbf{I}_p & \gamma\mathbf{I}_p\end{bmatrix}$$

for every $t \geqslant 0$, provided that $\gamma \geqslant m + M$.

To discretize this continuous-time process and make it applicable to the sampling problem, Shen & Lee (2019) proposed the following procedure: at each iteration $k = 1, 2, \ldots,$

1. randomly, and independently of all the variables generated at the previous steps, generate random vectors $(\boldsymbol{\xi}_k', \boldsymbol{\xi}_k'', \boldsymbol{\xi}_k''')$ and a random variable $U_k$ such that
   - $U_k$ is uniformly distributed in $[0, 1]$,
   - conditionally to $U_k = u$, $(\boldsymbol{\xi}_k', \boldsymbol{\xi}_k'', \boldsymbol{\xi}_k''')$ has the same joint distribution as $\big(\boldsymbol{B}_u - e^{-\gamma hu}\boldsymbol{G}_u, \boldsymbol{B}_1 - e^{-\gamma h}\boldsymbol{G}_1, \gamma e^{-\gamma h}\boldsymbol{G}_1\big)$, where $\boldsymbol{B}$ is a $p$-dimensional Brownian motion and $\boldsymbol{G}_t = \int_0^t e^{\gamma hs}\,\mathrm{d}\boldsymbol{B}_s$.

2. set $\psi(x) = (1 - e^{-x})/x$ and define the $(k+1)$th iterate of $\boldsymbol{\vartheta}^{\mathsf{RKLMC}}$ by

$$\boldsymbol{\vartheta}_{k+U} = \boldsymbol{\vartheta}_k + Uh\psi(\gamma Uh)\boldsymbol{v}_k - Uh\big(1 - \psi(\gamma Uh)\big)\nabla f(\boldsymbol{\vartheta}_k) + \sqrt{2h}\,\boldsymbol{\xi}_k'$$

$$\boldsymbol{\vartheta}_{k+1} = \boldsymbol{\vartheta}_k + h\psi(\gamma h)\boldsymbol{v}_k - \gamma h^2(1 - U)\psi\big(\gamma h(1 - U)\big)\nabla f(\boldsymbol{\vartheta}_{k+U}) + \sqrt{2h}\boldsymbol{\xi}_k''$$

$$\boldsymbol{v}_{k+1} = e^{-\gamma h}\boldsymbol{v}_k - \gamma h e^{-\gamma h(1-U)}\nabla f(\boldsymbol{\vartheta}_{k+U}) + \sqrt{2h}\,\boldsymbol{\xi}_k'''.$$

Although the sequence $(\boldsymbol{v}_k^{\mathsf{RKLMC}}, \boldsymbol{\vartheta}_k^{\mathsf{RKLMC}})$ approximates $(\boldsymbol{V}_{kh}^{\mathsf{KLD}}, \boldsymbol{L}_{kh}^{\mathsf{KLD}})$, it is not immediately apparent. The supplementary material clarifies this point. We state now the main result of this paper, providing a simple upper bound for the error of the RKLMC algorithm.

---

[3]For the sake of the self-containedness of this paper, we reproduce the proof of this inequality in Proposition 1 deferred to the Appendix.

**Theorem 2.** *Assume the function* $f : \mathbb{R}^p \to \mathbb{R}$ *satisfies Assumption 1. Choose* $\gamma$ *and* $h$ *so that* $\gamma \geqslant 5M$ *and* $\gamma h \leqslant 0.1\kappa^{-1/6}$, *where* $\kappa = M/m$. *Assume that* $\boldsymbol{\vartheta}_0$ *is independent of* $\boldsymbol{v}_0$ *and that* $\boldsymbol{v}_0 \sim \mathcal{N}_p(0, \gamma \mathbf{I}_p)$. *Then, for any* $n \geqslant 1$, *the distribution* $\nu_n^{\mathsf{RKLMC}}$ *of* $\boldsymbol{\vartheta}_n^{\mathsf{RKLMC}}$ *satisfies*

$$\mathsf{W}_2(\nu_n^{\mathsf{RKLMC}}, \pi) \leqslant 1.6\varrho^n \mathsf{W}_2(\nu_0, \pi) + 0.1\sqrt{\varrho^n \mathbb{E}[f(\boldsymbol{\vartheta}_0) - f(\boldsymbol{\theta}_*)]/m}$$
$$+ 0.2(\gamma h)^3 \sqrt{\kappa p/m} + 10(\gamma h)^{3/2}\sqrt{p/m},$$

*where* $\varrho = \exp(-mh)$, *and* $\boldsymbol{\theta}_* = \arg\min_{\boldsymbol{\theta} \in \mathbb{R}^p} f(\boldsymbol{\theta})$.

This result has several strengths and limitations, which are discussed below, after the corollary providing the number of required iterations to attain a predetermined level of accuracy.

**Corollary 2.** *Let* $\varepsilon \in (0, 1)$ *be a small constant. If* $\gamma = 5M$, $\boldsymbol{\vartheta}_0 = \boldsymbol{\theta}_*$ *and we choose* $h > 0$ *and* $n \in \mathbb{N}$ *so that*

$$\gamma h = \frac{\varepsilon^{2/3}}{5 + 0.6(\varepsilon^2\kappa)^{1/6}}, \quad \text{and} \quad n \geqslant \kappa \varepsilon^{-2/3}\big(25 + 3(\varepsilon^2\kappa)^{1/6}\big)\log(20/\varepsilon),$$

*then we have* $\mathsf{W}_2(\nu_n^{\mathsf{RKLMC}}, \pi) \leqslant \varepsilon\sqrt{p/m}$.

The corollary presented above gives the best-known convergence rate for the number of gradient evaluations required to achieve a prescribed error level in the case of a gradient Lipschitz potential, without any additional assumptions on its structure or smoothness. This rate, $\kappa\varepsilon^{-2/3}(1 + (\varepsilon^2\kappa)^{1/6})$, was first discovered by Shen & Lee (2019) (see also (He et al., 2020)). By employing our proposed proof technique described in Section 2, the result in Theorem 2 gets rid of the factor $nmh$ from the discretization error, which was present in the previous upper bounds of the sampling error. Furthermore, our bound contains only small and explicit constants. Finally, our result does not require the RKLMC algorithm to be initialized at the minimizer of the potential, which is important for extending the method to non-convex potentials.

On the downside, the condition $\gamma \geqslant 5M$ is stronger than the corresponding conditions used in prior work on the KLMC (without randomization). Indeed, these prior results generally require $\gamma \geqslant 2M$. Having a proof of Theorem 2 that reduces the factor 5 in $\gamma \geqslant 5M$ would lead to significant savings in running time.

## 4 IMPROVED ERROR BOUND FOR THE KINETIC LANGEVIN WITH EULER DISCRETIZATION

The proof techniques presented in the previous section can be used to derive an upper bound on the error of the kinetic Langevin Monte Carlo (KLMC) algorithm. KLMC is a discretized version of KLD equation 16, where the term $\nabla f(\boldsymbol{L}_t)$ is replaced by $\nabla f(\boldsymbol{L}_{kh})$ on each interval $[kh, (k+1)h]$. The resulting error bound, given in the following theorem, exhibits a better dependence on $\kappa$ than previously established bounds.

**Theorem 3.** *Let* $f : \mathbb{R}^p \to \mathbb{R}$ *satisfy* $m\mathbf{I}_p \preccurlyeq \nabla^2 f(\boldsymbol{\theta}) \preccurlyeq M\mathbf{I}_p$ *for every* $\boldsymbol{\theta} \in \mathbb{R}^p$. *Choose* $\gamma$ *and* $h$ *so that* $\gamma \geqslant 5M$ *and* $\sqrt{\kappa}\,\gamma h \leqslant 0.1$, *where* $\kappa = M/m$. *Assume that* $\boldsymbol{\vartheta}_0$ *is independent of* $\boldsymbol{v}_0$ *and that* $\boldsymbol{v}_0 \sim \mathcal{N}_p(0, \gamma\mathbf{I}_p)$. *Then, for any* $n \geqslant 1$, *the distribution* $\nu_n^{\mathsf{KLMC}}$ *of* $\boldsymbol{\vartheta}_n^{\mathsf{KLMC}}$ *satisfies*

$$\mathsf{W}_2(\nu_n^{\mathsf{KLMC}}, \pi) \leqslant 2\varrho^n \mathsf{W}_2(\nu_0, \pi) + 0.05\sqrt{\varrho^n \mathbb{E}[f(\boldsymbol{\vartheta}_0) - f(\boldsymbol{\theta}_*)]/m} + 0.9\gamma h\sqrt{\kappa p/m},$$

*where* $\varrho = \exp(-mh)$, *and* $\boldsymbol{\theta}_* = \arg\min_{\boldsymbol{\theta} \in \mathbb{R}^p} f(\boldsymbol{\theta})$.

Bounds on the error of KLMC under convexity assumption, or other related conditions, can be found in recent papers (Cheng et al., 2018b; Dalalyan & Riou-Durand, 2020; Monmarché, 2021; Monmarché, 2023). Our result has the advantage of providing an upper bound with the best known dependence on the condition number $\kappa$ and having relatively small numerical constants, as shown in the next corollary.

**Corollary 3.** *Let* $\varepsilon \in (0, 0.1)$. *If* $\gamma = 5M$, $\boldsymbol{\vartheta}_0 = \boldsymbol{\theta}_*$ *and we choose* $h > 0$ *and* $n \in \mathbb{N}$ *so that*

$$\gamma h = \varepsilon\kappa^{-1/2}, \quad \text{and} \quad n \geqslant 5\kappa^{3/2}\varepsilon^{-1}\log(20/\varepsilon)$$

*then we have* $\mathsf{W}_2(\nu_n^{\mathsf{KLMC}}, \pi) \leqslant \varepsilon\sqrt{p/m}$.

It is worth noting that our error bounds, along with the other bounds mentioned previously under strong convexity, rely on the synchronous coupling between the KLMC and the KLD. However, in the case of the vanilla Langevin, it has been shown in Durmus et al. (2019) that the dependence of the error bound on $\kappa$ can be improved by considering other couplings (in their case, the coupling is hidden in the analytical arguments). We conjecture that the dependence on $\kappa$ in the kinetic Langevin Monte Carlo algorithm can also be improved through non-synchronous coupling. Specifically, we conjecture that the number of iterations required to achieve a $W_2$-error bounded by $\varepsilon\sqrt{p/m}$ should scale as $\kappa/\varepsilon$ rather than $\kappa^{3/2}/\varepsilon$, as obtained in previous work and in Theorem 3.

## 5 NUMERICAL EXPERIMENTS

In this section, we compare the performance of LMC, KLMC, RLMC, and RKLMC algorithms. We apply the four algorithms to the posterior density of penalized logistic regression, defined by $\pi(\boldsymbol{\vartheta}) \propto \exp(-f(\boldsymbol{\vartheta}))$, with the potential function

$$f(\boldsymbol{\vartheta}) = \frac{\lambda}{2}\|\boldsymbol{\vartheta}\|^2 + \frac{1}{n_{\text{data}}}\sum_{i=1}^{n_{\text{data}}}\log(1 + \exp(-y_i \boldsymbol{x}_i^\top \boldsymbol{\vartheta}))\,,$$

where $\lambda > 0$ denotes the tuning parameter. The data $\{\boldsymbol{x}_i, y_i\}_{i=1}^m$, composed of binary labels $y_i \in \{-1, 1\}$ and features $\boldsymbol{x}_i \in \mathbb{R}^p$ generated from $x_{i,j} \overset{iid}{\sim} \mathcal{N}(0,1), \mathcal{N}(0,5)$, and $\mathcal{N}(0,10)$, corresponding to the plots from left to right, respectively. In our experiments, we have chosen $\lambda = 1/100$, $p = 3$ and $n_{\text{data}} = 100$.

Figure 1 shows the $W_2$-distance measured along the first dimension between the empirical distributions of the samples from the four algorithms and the target distribution[4], with different choices of $h$. These numerical results confirm our theoretical results. Indeed, we see that the randomized midpoint versions of LMC and KLMC perform better than their vanilla counterparts when the condition number is not too large (the leftmost plot). This order changes when $\kappa$ becomes large, as we see in the rightmost plot, where KLMC outperforms the other algorithms.

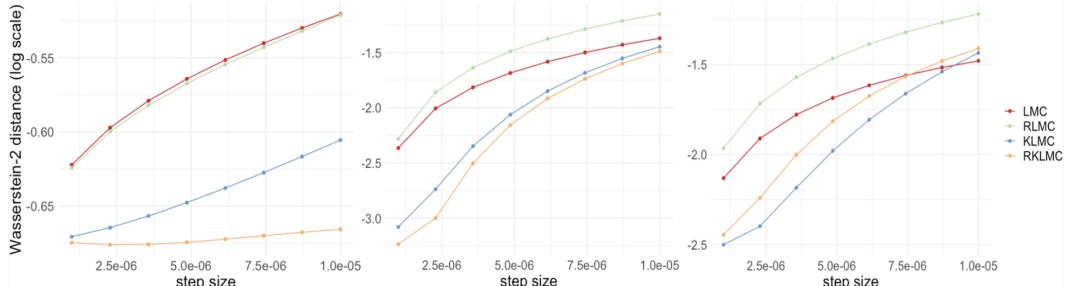

Figure 1: Error of {L,RL,KL,RKL}MC with different choice of step size.

## 6 DISCUSSION OF ASSUMPTIONS AND LIMITATIONS

The results presented in this paper provide easily computable guarantees for performing sampling with assured accuracy. These guarantees are conservative, implying that the actual sampling error may be smaller than $\varepsilon$ even if the upper bounds stated in our theorems are larger than $\varepsilon$. However, these bounds represent the most reliable technique available in the existing literature. The importance of having such guarantees is further emphasized by the lack of reliable practical measures to assess the quality of sampling methods. To better understand the computational complexity implied by our bounds for various Monte Carlo algorithms, we present in Table 1 the number of gradient evaluations required to achieve the accuracy of $\varepsilon\sqrt{p/m}$ for different combinations of $(\varepsilon, \kappa)$.

**Strong convexity** The assumption of strong convexity is often seen as too restrictive. In our theorems, strong convexity is used for three purposes: (a) to ensure the contraction of the continuous-time

---

[4]Here, we execute the LMC algorithm with a small step size over an extended duration to approximate the true distribution.

Table 1: The number of iterations that are sufficient for the algorithms {L,RL,KL,RKL}MC to achieve an error in $W_2$ distance bounded by $\varepsilon\sqrt{p/m}$, provided that they are initialized at the minimum of the potential $f$.

| $(\varepsilon, \kappa)$ | $(0.1^1, 10^1)$ | $(0.1^1, 10^3)$ | $(0.1^1, 10^5)$ | $(0.1^1, 10^7)$ | $(0.1^1, 10^9)$ | $(0.1^1, 10^{11})$ |
|---|---|---|---|---|---|---|
| LMC | $1.2 \times 10^4$ | $1.2 \times 10^6$ | $1.2 \times 10^8$ | $1.2 \times 10^{10}$ | $1.2 \times 10^{12}$ | $1.2 \times 10^{14}$ |
| RLMC | $3.6 \times 10^3$ | $1.1 \times 10^6$ | $4.5 \times 10^8$ | $2.0 \times 10^{11}$ | $9.3 \times 10^{13}$ | $4.3 \times 10^{16}$ |
| KLMC | $8.4 \times 10^3$ | $8.4 \times 10^6$ | $8.4 \times 10^9$ | $8.4 \times 10^{12}$ | $8.4 \times 10^{15}$ | $8.4 \times 10^{18}$ |
| RKLMC | $1.0 \times 10^4$ | $1.1 \times 10^6$ | $1.1 \times 10^8$ | $1.3 \times 10^{10}$ | $2.2 \times 10^{12}$ | $4.2 \times 10^{14}$ |
| $(\varepsilon, \kappa)$ | $(0.1^3, 10^1)$ | $(0.1^3, 10^3)$ | $(0.1^3, 10^5)$ | $(0.1^3, 10^7)$ | $(0.1^3, 10^9)$ | $(0.1^3, 10^{11})$ |
| LMC | $2.2 \times 10^8$ | $2.2 \times 10^{10}$ | $2.2 \times 10^{12}$ | $2.2 \times 10^{14}$ | $2.2 \times 10^{16}$ | $2.2 \times 10^{18}$ |
| RLMC | $3.8 \times 10^5$ | $6.8 \times 10^7$ | $2.0 \times 10^{10}$ | $8.4 \times 10^{12}$ | $3.8 \times 10^{15}$ | $1.7 \times 10^{18}$ |
| KLMC | $1.6 \times 10^6$ | $1.6 \times 10^9$ | $1.6 \times 10^{12}$ | $1.6 \times 10^{15}$ | $1.6 \times 10^{18}$ | $1.6 \times 10^{21}$ |
| RKLMC | $4.5 \times 10^5$ | $4.5 \times 10^7$ | $4.5 \times 10^{\ 9}$ | $4.5 \times 10^{11}$ | $4.7 \times 10^{13}$ | $5.7 \times 10^{15}$ |
| $(\varepsilon, \kappa)$ | $(0.1^5, 10^1)$ | $(0.1^5, 10^3)$ | $(0.1^5, 10^5)$ | $(0.1^5, 10^7)$ | $(0.1^5, 10^9)$ | $(0.1^5, 10^{11})$ |
| LMC | $3.2 \times 10^{12}$ | $3.2 \times 10^{14}$ | $3.2 \times 10^{16}$ | $3.2 \times 10^{18}$ | $3.2 \times 10^{20}$ | $3.2 \times 10^{22}$ |
| RLMC | $4.6 \times 10^7$ | $5.5 \times 10^9$ | $9.9 \times 10^{11}$ | $3.0 \times 10^{14}$ | $1.2 \times 10^{17}$ | $5.5 \times 10^{19}$ |
| KLMC | $2.3 \times 10^8$ | $2.3 \times 10^{11}$ | $2.3 \times 10^{14}$ | $2.3 \times 10^{17}$ | $2.3 \times 10^{20}$ | $2.3 \times 10^{23}$ |
| RKLMC | $1.5 \times 10^7$ | $1.5 \times 10^9$ | $1.5 \times 10^{11}$ | $1.5 \times 10^{13}$ | $1.5 \times 10^{15}$ | $1.5 \times 10^{17}$ |

Langevin dynamics, (b) to relate the potential's values to its gradient through the Polyak-Lojasiewicz condition $\|\nabla f(\boldsymbol{\theta})\|^2 \geqslant 2m(f(\boldsymbol{\theta}) - f(\boldsymbol{\theta}_*))$ (Polyak, 1963; Łojasiewicz, 1963), and (c) to provide the following simple upper bound on the 2-Wasserstein distance $W_2(\delta_{\boldsymbol{\theta}_*}, \pi) \leqslant \sqrt{p/m}$ (Durmus & Moulines, 2019, Prop. 1). The last two inequalities can be satisfied for many non-convex functions, but the same is not true for the contraction of the Langevin dynamics.

Alternatively, we can assume that the function is only strongly convex outside a ball of radius $R > 0$, whereas within the ball it is smooth but otherwise arbitrary. This approach requires an additional factor of order $e^{MR^2}$ in the number of iterations necessary to achieve a specified error level (Cheng et al., 2018a; Ma et al., 2019). We can also assume that the Markov semi-group has a spectral gap and use this gap in the risk bounds. However, this approach goes against the spirit of our paper, which aims to provide guarantees that are easy to interpret and verify.

Another important point to note is that the results obtained under the assumption of strong convexity can be used as ready-made results in other frameworks as well. For instance, this is applicable to weakly convex potentials or potentials supported on a compact set (Dalalyan et al., 2022; Dwivedi et al., 2018; Brosse et al., 2017).

**Smoothness** Smoothness of $f$ is a critical assumption for the results obtained in this paper. However, in statistical applications, this assumption may not hold, such as when using a Laplace prior. In such cases, various approaches have been proposed, mainly involving gradient approximation techniques, as explored in the literature (Durmus et al., 2018; Chatterji et al., 2020). Our results open the door for similar extensions of the randomized midpoint method for such scenarios.

It should also be stressed that if the potential is more than twice differentiable with a bounded tensor of higher-order derivatives, then it is possible to design Monte Carlo algorithms that perform better than the LMC and the KLMC (Dalalyan & Karagulyan, 2019; Dalalyan & Riou-Durand, 2020; Ma et al., 2021). The same is true if the function $f$ has some specific structure (Mou et al., 2021).

**Functional inequalities** Functional inequalities such as the Poincaré and the log-Sobolev inequalities provide a convenient framework for analyzing sampling methods derived from continuous-time Markov processes. This line of research was developed in a series of papers (Chewi et al., 2020; Vempala & Wibisono, 2019; Chewi et al., 2022). Extending the techniques of this paper from strong log-concavity to the framework of distributions satisfying one of the aforementioned functional inequalities is a non-trivial task.

**Other distances** The Wasserstein-2 distance, utilized in this paper, serves as a natural metric for measuring the error in sampling due to its connection with optimal transport. However, it is worth noting that recent literature on gradient-based sampling has explored other metrics such as total variation distance, KL divergence, and $\chi^2$ divergence (Ma et al., 2021; Vempala & Wibisono, 2019; Durmus et al., 2019; Chewi et al., 2020; Balasubramanian et al., 2022; Zhang et al., 2023). An interesting direction for future research involves establishing error guarantees for the randomized midpoint method with respect to these alternative distances.

ACKNOWLEDGMENTS

This work was partially supported by the grant Investissements d'Avenir (ANR-11-IDEX0003/Labex Ecodec/ANR-11-LABX-0047) and the center Hi! PARIS.

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

CONTENTS

# A  THE PROOF OF THE UPPER BOUND ON THE ERROR OF RLMC

This section is devoted to the proof of the upper bound on the error of sampling, measured in $W_2$-distance, of the randomized mid-point method for the vanilla Langevin Langevin diffusion. Since no other sampling method is considered in this section, without any risk of confusion, we will use the notation $\boldsymbol{\vartheta}_k$ instead of $\boldsymbol{\vartheta}_k^{\mathsf{RLMC}}$ to refer to the $k$-th iterate of the RLMC. We will also use the shorthand notation

$$f_k = f(\boldsymbol{\vartheta}_k), \qquad \nabla f_k := \nabla f(\boldsymbol{\vartheta}_k), \qquad \text{and} \qquad \nabla f_{k+U} := \nabla f(\boldsymbol{\vartheta}_{k+U}).$$

## A.1  PROOF OF THEOREM 1

Let $\boldsymbol{\vartheta}_0 \sim \nu_0$ and $\boldsymbol{L}_0 \sim \pi$ be two random vectors in $\mathbb{R}^p$ defined on the same probability space. At this stage, the joint distribution of these vectors is arbitrary; we will take an infimum over all possible joint distributions with given marginals at the end of the proof. Note right away that the condition $Mh + \sqrt{\kappa}(Mh)^{3/2} \leqslant 1/4$ implies that $Mh + (Mh)^{3/2} \leqslant 1/4$, which also yields $Mh \leqslant 0.18$.

Assume that on the same probability space, we can define a Brownian motion $\boldsymbol{W}$, independent of $(\boldsymbol{\vartheta}_0, \boldsymbol{L}_0)$, and an infinite sequence of iid random variables, uniformly distributed in $[0, 1]$, $U_0, U_1, \ldots$, independent of $(\boldsymbol{\vartheta}_0, \boldsymbol{L}_0, \boldsymbol{W})$. We define the Langevin diffusion

$$\boldsymbol{L}_t = \boldsymbol{L}_0 - \int_0^t \nabla f(\boldsymbol{L}_s)\mathrm{d}s + \sqrt{2}\,\boldsymbol{W}_t. \tag{17}$$

We also set

$$\boldsymbol{\vartheta}_{k+U} = \boldsymbol{\vartheta}_k - hU_k\nabla f_k + \sqrt{2}\left(\boldsymbol{W}_{(k+U_k)h} - \boldsymbol{W}_{kh}\right)$$
$$\boldsymbol{\vartheta}_{k+1} = \boldsymbol{\vartheta}_k - h\nabla f_{k+U} + \sqrt{2}\left(\boldsymbol{W}_{(k+1)h} - \boldsymbol{W}_{kh}\right).$$

One can check that this sequence $\{\boldsymbol{\vartheta}_k\}$ has exactly the same distribution as the sequence defined in equation 6 and equation 7. Therefore,

$$\mathsf{W}_2^2(\nu_{k+1}, \pi) \leqslant \mathbb{E}[\|\boldsymbol{\vartheta}_{k+1} - \boldsymbol{L}_{(k+1)h}\|_2^2] := \|\boldsymbol{\vartheta}_{k+1} - \boldsymbol{L}_{(k+1)h}\|_{\mathbb{L}_2}^2 := x_{k+1}^2.$$

We will also consider the Langevin process on the time interval $[0, h]$ given by

$$\boldsymbol{L}_t' = \boldsymbol{L}_0' - \int_0^t \nabla f(\boldsymbol{L}_s')\,\mathrm{d}s + \sqrt{2}\left(\boldsymbol{W}_{kh+t} - \boldsymbol{W}_{kh}\right), \qquad \boldsymbol{L}_0' = \boldsymbol{\vartheta}_k.$$

Note that the Brownian motion is the same as in equation 17.

Let us introduce one additional notation, the average of $\boldsymbol{\vartheta}_{k+1}$ with respect to $U_k$,

$$\bar{\boldsymbol{\vartheta}}_{k+1} = \mathbb{E}[\boldsymbol{\vartheta}_{k+1}|\boldsymbol{\vartheta}_k, \boldsymbol{W}, \boldsymbol{L}_0].$$

Since $\boldsymbol{L}_{(k+1)h}$ is independent of $U_k$, it is clear that

$$x_{k+1}^2 = \|\boldsymbol{\vartheta}_{k+1} - \bar{\boldsymbol{\vartheta}}_{k+1}\|_{\mathbb{L}_2}^2 + \|\bar{\boldsymbol{\vartheta}}_{k+1} - \boldsymbol{L}_{(k+1)h}\|_{\mathbb{L}_2}^2.$$

Furthermore, the triangle inequality yields

$$\|\bar{\boldsymbol{\vartheta}}_{k+1} - \boldsymbol{L}_{(k+1)h}\|_{\mathbb{L}_2} \leqslant \|\bar{\boldsymbol{\vartheta}}_{k+1} - \boldsymbol{L}_h'\|_{\mathbb{L}_2} + \|\boldsymbol{L}_h' - \boldsymbol{L}_{(k+1)h}\|_{\mathbb{L}_2}.$$

From the exponential ergodicity of the Langevin diffusion (Bhattacharya, 1978), we get

$$\|\boldsymbol{L}_h' - \boldsymbol{L}_{(k+1)h}\|_{\mathbb{L}_2} \leqslant e^{-mh}\|\boldsymbol{L}_0' - \boldsymbol{L}_{kh}\|_{\mathbb{L}_2} = e^{-mh}\|\boldsymbol{\vartheta}_k - \boldsymbol{L}_{kh}\|_{\mathbb{L}_2} = e^{-mh}x_k.$$

Therefore, we get

$$x_{k+1}^2 \leqslant \|\boldsymbol{\vartheta}_{k+1} - \bar{\boldsymbol{\vartheta}}_{k+1}\|_{\mathbb{L}_2}^2 + \left(\|\bar{\boldsymbol{\vartheta}}_{k+1} - \boldsymbol{L}_h'\|_{\mathbb{L}_2} + e^{-2mh}x_k\right)^2$$
$$= \left(e^{-mh}x_k + \|\bar{\boldsymbol{\vartheta}}_{k+1} - \boldsymbol{L}_h'\|_{\mathbb{L}_2}\right)^2 + \|\boldsymbol{\vartheta}_{k+1} - \bar{\boldsymbol{\vartheta}}_{k+1}\|_{\mathbb{L}_2}^2. \tag{18}$$

The last term of the right-hand side can be bounded as follows

$$\|\boldsymbol{\vartheta}_{k+1} - \bar{\boldsymbol{\vartheta}}_{k+1}\|_{\mathbb{L}_2} = h\|\nabla f_{k+U} - \mathbb{E}_U[\nabla f_{k+U}]\|_{\mathbb{L}_2}$$
$$\leqslant h\|\nabla f_{k+U} - \nabla f(\boldsymbol{\vartheta}_k)\|_{\mathbb{L}_2}.$$

Using the definition of $\boldsymbol{\vartheta}_{k+U}$, we get

$$\|\boldsymbol{\vartheta}_{k+1} - \bar{\boldsymbol{\vartheta}}_{k+1}\|_{\mathbb{L}_2}^2 \leqslant (Mh)^2\left((1/3)h^2\|\nabla f_k\|_{\mathbb{L}_2}^2 + hp\right). \tag{19}$$

We will also need the following lemma, the proof of which is postponed.

**Lemma 1.** *If $Mh \leqslant 0.18$, then $\|\bar{\boldsymbol{\vartheta}}_{k+1} - \boldsymbol{L}'_h\|_{\mathbb{L}_2} \leqslant (Mh)^2 \{0.7h\|\nabla f_k\|_{\mathbb{L}_2} + 1.2\sqrt{hp}\}$.*

One can check by induction that if for some $A \in [0, 1]$ and for two positive sequences $\{B_k\}$ and $\{C_k\}$ the inequality $x_{k+1}^2 \leqslant \{(1 - A)x_k + C_k\}^2 + B_k^2$ holds for every integer $k \geqslant 0$, then[5]

$$x_n \leqslant (1 - A)^n x_0 + \sum_{k=0}^{n} (1 - A)^{n-k} C_k + \left\{ \sum_{k=0}^{n} (1 - A)^{2(n-k)} B_k^2 \right\}^{1/2} \tag{20}$$

In view of equation 20, equation 18, equation 19 and Lemma 1, for $\rho = e^{-mh}$, we get

$$\begin{aligned}
x_n &\leqslant \rho^n x_0 + (Mh)^2 \sum_{k=0}^{n} \rho^{n-k} \left(0.7h\|\nabla f_k\|_{\mathbb{L}_2} + 1.2\sqrt{hp}\right) \\
&\quad + Mh \left\{ \sum_{k=0}^{n} \rho^{2(n-k)} \left((1/3)h^2\|\nabla f_k\|_{\mathbb{L}_2}^2 + hp\right) \right\}^{1/2} \\
&\leqslant \rho^n x_0 + 0.7(Mh)^2 h \sum_{k=0}^{n} \rho^{n-k} \|\nabla f_k\|_{\mathbb{L}_2} + 1.32 \frac{M^2 h\sqrt{hp}}{m} \\
&\quad + \frac{Mh^2}{\sqrt{3}} \left\{ \sum_{k=0}^{n} \rho^{2(n-k)} \|\nabla f(\boldsymbol{\vartheta}_k)\|_{\mathbb{L}_2}^2 \right\}^{1/2} + 0.92 Mh\sqrt{p/m}. \tag{21}
\end{aligned}$$

We need a last lemma for finding a suitable upper bound on the right-hand side of the last display.

**Lemma 2.** *If $Mh \leqslant 0.18$ and $k \geqslant 1$, then the following inequalities hold*

$$h^2 \sum_{k=0}^{n} \rho^{n-k} \|\nabla f(\boldsymbol{\vartheta}_k)\|_{\mathbb{L}_2}^2 \leqslant 1.7 Mh\rho^n \|\boldsymbol{\vartheta}_0\|_{\mathbb{L}_2}^2 + 4.4 Mh(p/m) \leqslant 0.31\rho^n \|\boldsymbol{\vartheta}_0\|_{\mathbb{L}_2}^2 + 0.8(p/m).$$

The claim of this lemma and together with equation 21 entail that

$$\begin{aligned}
x_n &\leqslant \rho^n x_0 + 0.7(Mh)^2 h \sum_{k=0}^{n} \rho^{n-k} \|\nabla f_k\|_{\mathbb{L}_2} + \frac{Mh^2}{\sqrt{3}} \left\{ \sum_{k=0}^{n} \rho^{2(n-k)} \|\nabla f(\boldsymbol{\vartheta}_k)\|_{\mathbb{L}_2}^2 \right\}^{1/2} \\
&\quad + (1.32\sqrt{\kappa Mh} + 0.92) Mh\sqrt{p/m} \\
&\leqslant \rho^n x_0 + \left(0.74\sqrt{\kappa Mh} + 0.58\right) Mh \left\{ h^2 \sum_{k=0}^{n} \rho^{n-k} \|\nabla f(\boldsymbol{\vartheta}_k)\|_{\mathbb{L}_2}^2 \right\}^{1/2} \\
&\quad + \left(1.32\sqrt{\kappa Mh} + 0.92\right) Mh\sqrt{p/m} \\
&\leqslant \rho^n x_0 + (0.42\sqrt{\kappa Mh} + 0.33) Mh\rho^{n/2} \|\boldsymbol{\vartheta}_0\|_{\mathbb{L}_2} + \left(1.98\sqrt{\kappa Mh} + 1.44\right) Mh\sqrt{p/m}.
\end{aligned}$$

Assuming that $h$ is such that $(\sqrt{\kappa Mh}+1)Mh \leqslant 1/4$ and noting that $\|\boldsymbol{\vartheta}_0\|_{\mathbb{L}_2} \leqslant \mathsf{W}_2(\nu_0, \pi) + \sqrt{p/m}$, we arrive at the desired inequality.

## A.2 Proof of technical lemmas

In this section, we present the proofs of two technical lemmas that have been used in the proof of the main theorem. The first lemma provides an upper bound on the error of the averaged iterate $\bar{\boldsymbol{\vartheta}}_{k+1}$ and the continuous time diffusion $\boldsymbol{L}'$ that starts from $\boldsymbol{\vartheta}_k$ and runs until the time $h$. This upper bound involves the norm of the gradient of the potential $f$ evaluated at $\boldsymbol{\vartheta}_k$. The second lemma aims at bounding the discounted sums of the squared norms of these gradients.

---

[5]This is an extension of (Dalalyan & Karagulyan, 2019, Lemma 7). It essentially relies on the elementary $\sqrt{(a + b)^2 + c^2} \leqslant a + \sqrt{b^2 + c^2}$, which should be used to prove the induction step.

### A.2.1 PROOF OF LEMMA 1 (ONE-STEP MEAN DISCRETISATION ERROR)

We have

$$
\begin{aligned}
\|\bar{\boldsymbol{\vartheta}}_{k+1} - \boldsymbol{L}'_h\|_{\mathbb{L}_2} &= \left\|\boldsymbol{\vartheta}_k - h\mathbb{E}_U[\nabla f_{k+U}] - \boldsymbol{L}'_0 + \int_0^h \nabla f(\boldsymbol{L}'_s)\,\mathrm{d}s\right\|_{\mathbb{L}_2} \\
&= \left\|h\mathbb{E}_U[\nabla f_{k+U} - \nabla f(\boldsymbol{L}'_{Uh})]\right\|_{\mathbb{L}_2} \\
&\leqslant Mh\left\|\boldsymbol{\vartheta}_{k+U} - \boldsymbol{L}'_{Uh}\right\|_{\mathbb{L}_2} \\
&= Mh\left\|\boldsymbol{\vartheta}_k - U_k h\nabla f_k - \boldsymbol{L}'_0 + \int_0^{U_k h} \nabla f(\boldsymbol{L}'_s)\mathrm{d}s\right\|_{\mathbb{L}_2} \\
&= Mh\left\|\int_0^{U_k h} (\nabla f(\boldsymbol{L}'_s) - \nabla f_k)\,\mathrm{d}s\right\|_{\mathbb{L}_2} \\
&\leqslant Mh\int_0^h \|\nabla f(\boldsymbol{L}'_s) - \nabla f_k\|_{\mathbb{L}_2}\,\mathrm{d}s \\
&= Mh\int_0^h \left\|\nabla f(\boldsymbol{L}'_s) - \nabla f(\boldsymbol{L}'_0)\right\|_{\mathbb{L}_2}\,\mathrm{d}s.
\end{aligned}
\tag{22}
$$

Let us define $\varphi(t) = \|\nabla f(\boldsymbol{L}'_t) - \nabla f(\boldsymbol{L}'_0)\|_{\mathbb{L}_2}$. Using the Lipschitz continuity of $\nabla f$ and the definition of $\boldsymbol{L}'$, we arrive at

$$
\begin{aligned}
\varphi(t)^2 &\leqslant M^2\left\{\left\|\int_0^t \nabla f(\boldsymbol{L}'_s)\,\mathrm{d}s\right\|_{\mathbb{L}_2}^2 + 2tp\right\} \\
&\leqslant M^2\left\{\left(t\|\nabla f_k\|_{\mathbb{L}_2} + \int_0^t \left\|\nabla f(\boldsymbol{L}'_s) - \nabla f(\boldsymbol{L}'_0)\right\|_{\mathbb{L}_2}\,\mathrm{d}s\right)^2 + 2tp\right\} \\
&\leqslant M^2\left\{\int_0^t \left\|\nabla f(\boldsymbol{L}'_s) - \nabla f(\boldsymbol{L}'_0)\right\|_{\mathbb{L}_2}\,\mathrm{d}s + \sqrt{t^2\|\nabla f_k\|_{\mathbb{L}_2}^2 + 2tp}\right\}^2
\end{aligned}
$$

or, equivalently,

$$
\varphi(t) \leqslant M\int_0^t \varphi(s)\,\mathrm{d}s + M\sqrt{t^2\|\nabla f_k\|_{\mathbb{L}_2}^2 + 2tp}.
$$

Using the Grönwall inequality, we get

$$
\varphi(t) \leqslant Me^{Mt}\sqrt{t^2\|\nabla f_k\|_{\mathbb{L}_2}^2 + 2tp}.
$$

Combining this inequality with the bound obtained in equation 22, and using the inequality $e^{Mh} \leqslant 1.2$, we arrive at

$$
\begin{aligned}
\|\bar{\boldsymbol{\vartheta}}_{k+1} - \boldsymbol{L}'_h\|_{\mathbb{L}_2} &\leqslant 1.2M^2 h\int_0^h \sqrt{s^2\|\nabla f_k\|_{\mathbb{L}_2}^2 + 2sp}\,\mathrm{d}s \\
&\leqslant 1.2M^2 h\sqrt{h}\left\{\int_0^h \left(s^2\|\nabla f_k\|_{\mathbb{L}_2}^2 + 2sp\right)\mathrm{d}s\right\}^{1/2} \\
&\leqslant 1.2M^2 h\sqrt{h}\left\{(h^3/3)\|\nabla f_k\|_{\mathbb{L}_2}^2 + h^2 p\right\}^{1/2}.
\end{aligned}
$$

This completes the proof.

### A.2.2 PROOF OF LEMMA 2 (DISCOUNTED SUM OF SQUARED GRADIENTS)

We have

$$
\begin{aligned}
f_{k+1} &\leqslant f_k + \nabla f_k^\top (\boldsymbol{\vartheta}_{k+1} - \boldsymbol{\vartheta}_k) + \frac{M}{2}\|\boldsymbol{\vartheta}_{k+1} - \boldsymbol{\vartheta}_k\|_2^2 \\
&\leqslant f_k - h\nabla f_k^\top \nabla f_{k+U} + \sqrt{2}\nabla f_k^\top \boldsymbol{\xi}_h + \frac{M}{2}\|h\nabla f_{k+U} - \sqrt{2}\boldsymbol{\xi}_k\|_2^2 \\
&\leqslant f_k - h\|\nabla f_k\|_2^2 + Mh\|\nabla f_k\|_2\|\boldsymbol{\vartheta}_{k+U} - \boldsymbol{\vartheta}_k\|_2 + \sqrt{2}\nabla f_k^\top \boldsymbol{\xi}_h + \frac{M}{2}\|h\nabla f_{k+U} - \sqrt{2}\boldsymbol{\xi}_k\|_2^2.
\end{aligned}
\tag{23}
$$

One checks that
$$\|\boldsymbol{\vartheta}_{k+U} - \boldsymbol{\vartheta}_k\|_{\mathbb{L}_2}^2 = h^2\|U\nabla f_k\|_{\mathbb{L}_2}^2 + 2hp\mathbb{E}[U] = (h^2/3)\|\nabla f_k\|_{\mathbb{L}_2}^2 + hp \leqslant 0.011\frac{\|\nabla f_k\|_{\mathbb{L}_2}^2}{M^2} + hp$$
and, therefore,
$$\begin{aligned}
M\|\nabla f_k\|_2\|\boldsymbol{\vartheta}_{k+U} - \boldsymbol{\vartheta}_k\|_2 &\leqslant \left(0.011\|\nabla f_k\|_{\mathbb{L}_2}^4 + M^2 hp\|\nabla f_k\|_2^2\right)^{1/2}\\
&\leqslant 0.105\|\nabla f_k\|_{\mathbb{L}_2}^2 + 4.55M^2 hp\\
&\leqslant 0.105\|\nabla f_k\|_{\mathbb{L}_2}^2 + 0.82Mp.
\end{aligned}$$

Furthermore,
$$\begin{aligned}
\|h\nabla f_{k+U} - \sqrt{2}\,\boldsymbol{\xi}_k\|_{\mathbb{L}_2} &\leqslant \|h\nabla f_k - \sqrt{2}\,\boldsymbol{\xi}_k\|_{\mathbb{L}_2} + h\|\nabla f_{k+U} - \nabla f_k\|_{\mathbb{L}_2}\\
&\leqslant \sqrt{h^2\|\nabla f_k\|_{\mathbb{L}_2}^2 + hp} + Mh\|\boldsymbol{\vartheta}_{k+U} - \boldsymbol{\vartheta}_k\|_{\mathbb{L}_2}\\
&\leqslant \sqrt{h^2\|\nabla f_k\|_{\mathbb{L}_2}^2 + hp} + h\sqrt{0.011\|\nabla f_k\|_{\mathbb{L}_2}^2 + M^2 hp}\\
&\leqslant \sqrt{h^2\|\nabla f_k\|_{\mathbb{L}_2}^2 + hp} + \sqrt{0.011h^2\|\nabla f_k\|_{\mathbb{L}_2}^2 + 0.18^2 hp}
\end{aligned}$$
implying that
$$\begin{aligned}
\frac{M}{2}\|h\nabla f_{k+U} - \sqrt{2}\,\boldsymbol{\xi}_k\|_{\mathbb{L}_2}^2 &\leqslant \frac{M}{2}\left(1.37h^2\|\nabla f_k\|_{\mathbb{L}_2}^2 + 1.4hp\right)\\
&\leqslant 0.124h^2\|\nabla f_k\|_{\mathbb{L}_2}^2 + 0.7Mhp.
\end{aligned}$$
Combining these inequalities with equation 23, we get
$$\mathbb{E}[f_{k+1}] \leqslant \mathbb{E}[f_k] - 0.771h\|\nabla f_k\|_{\mathbb{L}_2}^2 + 1.52Mhp. \tag{24}$$
Set $S_n(f) = \sum_{k=0}^n \rho^{n-k} f_k$ and $S_n(\nabla f^2) = \sum_{k=0}^n \rho^{n-k}\|\nabla f_k\|_{\mathbb{L}_2}^2$. Using Lemma 3, we get
$$\mathbb{E}[f_{n+1}] - \rho^n\mathbb{E}[f_0] + \rho S_n(f) \leqslant S_n(f) - 0.771h S_n(\nabla f^2) + \frac{1.52Mhp}{1-\rho}$$
Since $mh \geqslant 1 - \rho \geqslant 0.915mh$, we get
$$\begin{aligned}
0.771h S_n(\nabla f^2) &\leqslant \rho^n\mathbb{E}[f_0] + (1-\rho)S_n(f) + 1.67\kappa p\\
&\leqslant \rho^n\mathbb{E}[f_0] + mh S_n(f) + 1.67\kappa p\\
&\leqslant \rho^n\mathbb{E}[f_0] + 0.5h S_n(\nabla f^2) + 1.67\kappa p
\end{aligned}$$
where the last line follows from the Polyak-Lojasiewicz inequality. Rearranging the terms, we get
$$h S_n(\nabla f^2) \leqslant 3.7\rho^n\mathbb{E}[f_0] + 6.2\kappa p \tag{25}$$
Note that equation 25 is obtained under the Polyak-Lojasiewicz condition, without explicitly using the strong convexity of $f$. However, using the latter property, we can obtain a similar inequality with slightly better constants.

Indeed, equation 24 yields
$$h\mathbb{E}[\|\nabla f_k\|_2^2] \leqslant 1.3\left(\mathbb{E}[f_k] - \mathbb{E}[f_{k+1}]\right) + 1.98Mhp. \tag{26}$$
In what follows, without loss of generality, we assume that $f(\boldsymbol{\theta}^*) = \min_{\boldsymbol{\theta}} f(\boldsymbol{\theta}) = 0$. In view of equation 26, we have
$$\begin{aligned}
h\sum_{j=0}^k \rho^{k-j}\|\nabla f(\boldsymbol{\vartheta}_j)\|_{\mathbb{L}_2}^2 &\leqslant 1.3\sum_{j=0}^k \rho^{k-j}\left(\mathbb{E}[f(\boldsymbol{\vartheta}_j) - f(\boldsymbol{\vartheta}_{j+1})]\right) + \frac{1.98Mhp}{1 - e^{-mh}}\\
&\leqslant 1.3\left(\rho^k\mathbb{E}[f(\boldsymbol{\vartheta}_0)] - \mathbb{E}[f_{k+1}]\right) + 1.3\sum_{j=1}^k \rho^{k-j}(1-\rho)\mathbb{E}[f(\boldsymbol{\vartheta}_j)] + 2.1\kappa p\\
&\leqslant 1.3\rho^{k+1}\mathbb{E}[f(\boldsymbol{\vartheta}_0)] + 1.3(1-\rho)\sum_{j=0}^k \rho^{k-j}\mathbb{E}[f(\boldsymbol{\vartheta}_j)] + 2.1\kappa p\\
&\leqslant \frac{1.3M}{2}\rho^{k+1}\|\boldsymbol{\vartheta}_0\|_{\mathbb{L}_2}^2 + \frac{1.3}{2}M(1-\rho)\sum_{j=0}^k \rho^{k-j}\|\boldsymbol{\vartheta}_j\|_{\mathbb{L}_2}^2 + 2.1\kappa p.
\end{aligned}$$

We have, in addition

$$\|\boldsymbol{\vartheta}_{k+1}\|_{\mathbb{L}_2}^2 = \|\boldsymbol{\vartheta}_k - h\nabla f_k\|_{\mathbb{L}_2}^2 + 2hp \leqslant (1-mh)^2\|\boldsymbol{\vartheta}_k\|_{\mathbb{L}_2}^2 + 2hp.$$

Therefore,

$$\|\boldsymbol{\vartheta}_k\|_{\mathbb{L}_2}^2 \leqslant (1-mh)^{2k}\|\boldsymbol{\vartheta}_0\|_{\mathbb{L}_2}^2 + \frac{2hp}{2mh - (mh)^2} \leqslant (1-mh)^{2k}\|\boldsymbol{\vartheta}_0\|_{\mathbb{L}_2}^2 + \frac{1.1p}{m}.$$

Using this inequality in conjunction with the fact that $1 - mh \leqslant \rho$, we arrive at

$$h\sum_{j=0}^{k} \rho^{k-j}\|\nabla f(\boldsymbol{\vartheta}_j)\|_{\mathbb{L}_2}^2 \leqslant \frac{1.3M}{2}\rho^{k+1}\|\boldsymbol{\vartheta}_0\|_{\mathbb{L}_2}^2 + \frac{1.3}{2}M\rho^{2k}\|\boldsymbol{\vartheta}_0\|_{\mathbb{L}_2}^2 + 2.9\kappa p.$$

This completes the proof of the lemma.

## B  THE PROOF OF THE UPPER BOUND ON THE ERROR OF KLMC

The goal of this section is to present the proof of the bound on the error of sampling of the "standard" discretization of the kinetic Langevin diffusion. With a slight abuse of language, we will call it Euler-Maruyama discretized kinetic Langevin diffusion, or kinetic Langevin Monte Carlo (KLMC). To avoid complicated notation, and since there is no risk of confusion, throughout this section $\boldsymbol{\vartheta}_k$ and $\boldsymbol{v}_k$ will refer to $\boldsymbol{\vartheta}_k^{\mathsf{KLMC}}$ and $\boldsymbol{v}_k^{\mathsf{KLMC}}$, respectively. We will also use the following shorthand notation:

$$f_n = f(\boldsymbol{\vartheta}_n), \quad \boldsymbol{g}_n = \nabla f_n = \nabla f(\boldsymbol{\vartheta}_n), \quad \eta = \gamma h, \quad M_\gamma = M/\gamma.$$

The advantage of dealing with $\eta$ instead of $h$ is that the former is scale-free.

Note that the iterates of KLMC satisfy

$$\boldsymbol{v}_{n+1} = (1 - \alpha\eta)\boldsymbol{v}_n - \alpha\eta\,\boldsymbol{g}_n + \sqrt{2\gamma\eta}\,\sigma\boldsymbol{\xi}_n \tag{27}$$

$$\boldsymbol{\vartheta}_{n+1} = \boldsymbol{\vartheta}_n + \gamma^{-1}\eta\big(\alpha\boldsymbol{v}_n - \beta\eta\boldsymbol{g}_n + \sqrt{2\gamma\eta}\,\tilde{\sigma}\bar{\boldsymbol{\xi}}_n\big), \tag{28}$$

where

$$\alpha = \frac{1 - e^{-\eta}}{\eta} \in (0, 1), \qquad \beta = \frac{e^{-\eta} - 1 + \eta}{\eta^2} \in (0, 1/2),$$

$$\sigma^2 = \frac{1 - e^{-2\eta}}{2\eta} \in (0, 1), \qquad \tilde{\sigma}^2 = \frac{2(1 - 2\eta + 2\eta^2 - e^{-2\eta})}{(2\eta)^3} \in (0, 1/3)$$

and $\boldsymbol{\xi}_n, \bar{\boldsymbol{\xi}}_n$ are two $\mathcal{N}_p(\mathbf{0}, \mathbf{I}_p)$-distributed random vectors independent of $(\boldsymbol{\vartheta}_n, \boldsymbol{v}_n)$.

Since we assume throughout this section that $2Mh \leqslant 0.1$, $\gamma \geqslant 2M$ and $\kappa \geqslant 10$, we have

$$\alpha = \frac{1 - \exp(-\eta)}{\eta} \geq 0.95, \quad \text{and} \quad mh = \frac{Mh}{\kappa} \leqslant \frac{Mh}{10} \leqslant \frac{1}{200}.$$

The latter, in particular, implies the following bound for $\varrho$:

$$1 - mh \leqslant \varrho = e^{-mh} \leqslant 1 - 0.99mh = 1 - 0.99m\eta/\gamma. \tag{29}$$

For any sequence $\omega = (\omega_n)_{n\in\mathbb{N}}$ of real numbers, we denote by $S_n(\omega)$ the $\rho$-discounted sum $\sum_{k=0}^n \rho^{n-k}\omega_k$. Below we present a simple lemma for the function $S_n(\cdot)$ that we will use repeatedly in this proof.

**Lemma 3** (Summation by parts)**.** *Suppose $\omega = (\omega_n)_{n\in\mathbb{N}}$ is a sequence of real numbers and define $S_n^{+1}(\omega) := \sum_{k=0}^n \varrho^{n-k}\omega_{k+1}$. Then, the following identity is true*

$$S_n^{+1}(\omega) = \omega_{n+1} - \varrho^{n+1}\omega_0 + \varrho S_n(\omega).$$

*Proof.* The proof is based on simple algebra:

$$S_n^{+1}(\omega) = \omega_{n+1} + \sum_{j=1}^n \varrho^{n-j+1}\omega_j = \omega_{n+1} + \varrho\left(S_n(\omega) - \varrho^n\omega_0\right). \qquad \square$$

### B.1  EXPONENTIAL MIXING OF CONTINUOUS-TIME KINETIC LANGEVIN DIFFUSION

Consider the kinetic Langevin diffusions

$$d\boldsymbol{L}_t = \boldsymbol{V}_t\,dt \qquad d\boldsymbol{V}_t = -\gamma\boldsymbol{V}_t\,dt - \gamma\nabla f(\boldsymbol{L}_t)dt + \sqrt{2\gamma}\,d\boldsymbol{W}_t \tag{30}$$

**Proposition 1.** *Let $\boldsymbol{V}_0, \boldsymbol{L}_0$ and $\boldsymbol{L}_0'$ be random vectors in $\mathbb{R}^p$. Let $(\boldsymbol{V}_t, \boldsymbol{L}_t)$ and $(\boldsymbol{V}_t', \boldsymbol{L}_t')$ be kinetic Langevin diffusions defined in equation 30 driven by the same Brownian motion and starting from $(\boldsymbol{V}_0, \boldsymbol{L}_0)$ and $(\boldsymbol{V}_0', \boldsymbol{L}_0')$ respectively. It holds for any $t \geqslant 0$ that*

$$\left\| \mathbf{C} \begin{bmatrix} \boldsymbol{V}_t - \boldsymbol{V}_t' \\ \boldsymbol{L}_t - \boldsymbol{L}_t' \end{bmatrix} \right\| \leqslant e^{-\{m\wedge(\gamma-M)\}t} \left\| \mathbf{C} \begin{bmatrix} \boldsymbol{V}_0 - \boldsymbol{V}_0' \\ \boldsymbol{L}_0 - \boldsymbol{L}_0' \end{bmatrix} \right\|, \quad \text{with} \quad \mathbf{C} = \begin{bmatrix} \mathbf{I}_p & \mathbf{0}_p \\ \mathbf{I}_p & \gamma\mathbf{I}_p \end{bmatrix}.$$

*Proof of Proposition 1.* Set $\boldsymbol{Y}_t := \boldsymbol{V}_t - \boldsymbol{V}'_t + \gamma(\boldsymbol{L}_t - \boldsymbol{L}'_t)$, $\boldsymbol{Z}_t := \boldsymbol{V}_t - \boldsymbol{V}'_t$, that is

$$\begin{bmatrix} \boldsymbol{Z}_t \\ \boldsymbol{Y}_t \end{bmatrix} = \mathbf{C} \begin{bmatrix} \boldsymbol{V}_t - \boldsymbol{V}'_t \\ \boldsymbol{L}_t - \boldsymbol{L}'_t \end{bmatrix}.$$

We note that by the Taylor expansion, we have

$$\nabla f(\boldsymbol{L}_t) - \nabla f(\boldsymbol{L}'_t) = \mathbf{H}_t(\boldsymbol{L}_t - \boldsymbol{L}'_t),$$

where $\mathbf{H}_t := \int_0^1 \nabla^2 f(\boldsymbol{L}_t - x(\boldsymbol{L}_t - \boldsymbol{L}'_t)) \, \mathrm{d}x$. By the definition of $(\boldsymbol{V}_t, \boldsymbol{L}_t)$ and $(\boldsymbol{V}'_t, \boldsymbol{L}'_t)$, we find

$$\begin{aligned} \frac{\mathrm{d}}{\mathrm{d}t}(\boldsymbol{V}_t - \boldsymbol{V}'_t + \gamma(\boldsymbol{L}_t - \boldsymbol{L}'_t)) &= -\gamma \mathbf{H}_t(\boldsymbol{L}_t - \boldsymbol{L}'_t) \\ &= -\mathbf{H}_t(\boldsymbol{Y}_t - \boldsymbol{Z}_t). \end{aligned}$$

Similarly, we obtain

$$\begin{aligned} \frac{\mathrm{d}}{\mathrm{d}t}(\boldsymbol{V}_t - \boldsymbol{V}'_t) &= -\gamma(\boldsymbol{V}_t - \boldsymbol{V}'_t) - \gamma \mathbf{H}_t(\boldsymbol{L}_t - \boldsymbol{L}'_t) \\ &= -\gamma \boldsymbol{Z}_t - \mathbf{H}_t(\boldsymbol{Y}_t - \boldsymbol{Z}_t). \end{aligned}$$

This implies

$$\begin{aligned} \frac{d}{dt}\left[ \|\boldsymbol{Y}_t\|^2 + \|\boldsymbol{Z}_t\|^2 \right] &= 2\boldsymbol{Y}_t^\top(-\mathbf{H}_t\boldsymbol{Y}_t + \mathbf{H}_t\boldsymbol{Z}_t) + 2\boldsymbol{Z}_t^\top(-\gamma\boldsymbol{Z}_t - \mathbf{H}_t\boldsymbol{Y}_t + \mathbf{H}_t\boldsymbol{Z}_t) \\ &\leqslant 2\left( -m\|\boldsymbol{Y}_t\|^2 - \gamma\|\boldsymbol{Z}_t^2\| + M\|\boldsymbol{Z}_t\|^2 \right) \\ &\leqslant -2(m \wedge (\gamma - M))\left\| \begin{bmatrix} \boldsymbol{Z}_t \\ \boldsymbol{Y}_t \end{bmatrix} \right\|^2. \end{aligned}$$

Invoking Gronwall's inequality, we get

$$\left\| \begin{bmatrix} \boldsymbol{Z}_t \\ \boldsymbol{Y}_t \end{bmatrix} \right\| \leqslant \exp\left( -\{m \wedge (\gamma - M)\}t \right) \left\| \begin{bmatrix} \boldsymbol{Z}_0 \\ \boldsymbol{Y}_0 \end{bmatrix} \right\|$$

as desired. $\qquad\square$

## B.2 PROOF OF THEOREM 3

Let $\boldsymbol{\vartheta}_n, \boldsymbol{v}_n$ be the iterates of the KLMC algorithm. Let $(\boldsymbol{L}_t, \boldsymbol{V}_t)$ be the kinetic Langevin diffusion, coupled with $(\boldsymbol{\vartheta}_n, \boldsymbol{v}_n)$ through the same Brownian motion $(\boldsymbol{W}_t; t \geqslant 0)$ and starting from a random point $(\boldsymbol{L}_0, \boldsymbol{V}_0) \propto \exp(-f(\boldsymbol{y}) + \frac{1}{2\gamma}\|\boldsymbol{v}\|_2^2)$ such that $\boldsymbol{V}_0 = \boldsymbol{v}_0$. This means that

$$\boldsymbol{v}_{n+1} = \boldsymbol{v}_n e^{-\eta} - \gamma \int_0^h e^{-\gamma(h-s)} \, \mathrm{d}s \nabla f(\boldsymbol{\vartheta}_n) + \sqrt{2}\gamma \int_0^h e^{-\gamma(h-s)} \, \mathrm{d}\boldsymbol{W}_s$$

$$\boldsymbol{\vartheta}_{n+1} = \boldsymbol{\vartheta}_n + \int_0^h \left( \boldsymbol{v}_n e^{-\gamma u} - \gamma \int_0^u e^{-\gamma(u-s)} \, \mathrm{d}s \nabla f(\boldsymbol{\vartheta}_n) + \sqrt{2}\gamma \int_0^u e^{-\gamma(u-s)} \, \mathrm{d}\boldsymbol{W}_s \right) \mathrm{d}u.$$

We also consider the kinetic Langevin diffusion, $(\boldsymbol{L}', \boldsymbol{V}')$, defined on $[0, h]$ with the starting point $(\boldsymbol{\vartheta}_n, \boldsymbol{v}_n)$ and driven by the Brownian motion $(\boldsymbol{W}_{nh+t} - \boldsymbol{W}_{nh}; t \in [0, h])$. It satisfies

$$\boldsymbol{V}'_t = \boldsymbol{v}_n e^{-\gamma t} - \gamma \int_0^t e^{-\gamma(t-s)} \nabla f(\boldsymbol{L}'_s) \, \mathrm{d}s + \sqrt{2}\gamma \int_0^t e^{-\gamma(t-s)} \, \mathrm{d}\boldsymbol{W}_s$$

$$\boldsymbol{L}'_t = \boldsymbol{\vartheta}_n + \int_0^t \boldsymbol{V}'_s \, \mathrm{d}s.$$

Our goal will be to bound the term $x_n$ defined by

$$x_n = \left\| \mathbf{C} \begin{bmatrix} \boldsymbol{v}_n - \boldsymbol{V}_{nh} \\ \boldsymbol{\vartheta}_n - \boldsymbol{L}_{nh} \end{bmatrix} \right\|_{\mathbb{L}_2} \quad \text{with} \quad \mathbf{C} = \begin{bmatrix} \mathbf{I}_p & \mathbf{0}_p \\ \mathbf{I}_p & \gamma\mathbf{I}_p \end{bmatrix}. \tag{31}$$

The triangle inequality yields

$$
\begin{aligned}
x_{n+1} &\leqslant \left\| \mathbf{C} \begin{bmatrix} \boldsymbol{v}_n - \boldsymbol{V}_h' \\ \boldsymbol{\vartheta}_n - \boldsymbol{L}_h' \end{bmatrix} \right\|_{\mathbb{L}_2} + \left\| \mathbf{C} \begin{bmatrix} \boldsymbol{V}_h' - \boldsymbol{V}_{nh} \\ \boldsymbol{L}_h' - \boldsymbol{L}_{nh} \end{bmatrix} \right\|_{\mathbb{L}_2} \\
&\leqslant \left\| \mathbf{C} \begin{bmatrix} \boldsymbol{v}_n - \boldsymbol{V}_h' \\ \boldsymbol{\vartheta}_n - \boldsymbol{L}_h' \end{bmatrix} \right\|_{\mathbb{L}_2} + \varrho x_n \\
&\leqslant \varrho x_n + \sqrt{2}\, \|\boldsymbol{v}_{n+1} - \boldsymbol{V}_h'\|_{\mathbb{L}_2} + \gamma \|\boldsymbol{\vartheta}_{n+1} - \boldsymbol{L}_h'\|_{\mathbb{L}_2},
\end{aligned}
\tag{32}
$$

where the second inequality follows from Proposition 1 (see also the proof of (Dalalyan & Riou-Durand, 2020, Prop. 1)), while the third inequality is a consequence of the elementary inequality $\sqrt{a^2 + (a+b)^2} \leqslant \sqrt{2}\, a + b$ for $a, b \geqslant 0$.

The next lemma gives an upper bound on the terms appearing in the right-hand side of equation 32.

**Lemma 4.** *If $\nabla f$ is $M$-Lipschitz continuous, then for every step-size $\eta = \gamma h \geqslant 0$ and every $\gamma \geqslant 0$, the following holds*

$$
\|\boldsymbol{v}_{n+1} - \boldsymbol{V}_h'\|_{\mathbb{L}_2} \leqslant \tfrac{1}{6}\big\{ 2\sqrt{\gamma p \eta} + 3\|\boldsymbol{v}_n\|_{\mathbb{L}_2} + \eta\|\boldsymbol{g}_n\|_{\mathbb{L}_2} \big\} M_\gamma \eta^2 e^{M_\gamma \eta^2/2}
$$

$$
\gamma \|\boldsymbol{\vartheta}_{n+1} - \boldsymbol{L}_h'\|_{\mathbb{L}_2} \leqslant \tfrac{1}{6}\big( 0.6\sqrt{\gamma p \eta} + \|\boldsymbol{v}_n\|_{\mathbb{L}_2} + 0.25\eta\|\boldsymbol{g}_n\|_{\mathbb{L}_2} + \big) M_\gamma \eta^3 e^{M_\gamma \eta^2/2},
$$

*where $M_\gamma = M/\gamma$.*

For $\eta \leqslant 0.2$ and $\gamma \geqslant 2M$, Lemma 4 implies

$$
\|\boldsymbol{v}_{n+1} - \boldsymbol{V}_h'\|_{\mathbb{L}_2} \leqslant M_\gamma \eta^2 (0.15\sqrt{\gamma p} + 0.51\|\boldsymbol{v}_n\|_{\mathbb{L}_2} + 0.17\eta\|\boldsymbol{g}_n\|_{\mathbb{L}_2})
$$

$$
\gamma \|\boldsymbol{\vartheta}_{n+1} - \boldsymbol{L}_h'\|_{\mathbb{L}_2} \leqslant M_\gamma \eta^3 (0.046\sqrt{\gamma p} + 0.17\|\boldsymbol{v}_n\|_{\mathbb{L}_2} + 0.043\eta\|\boldsymbol{g}_n\|_{\mathbb{L}_2}).
$$

Therefore,

$$
\sqrt{2}\, \|\boldsymbol{v}_{n+1} - \boldsymbol{V}_h'\|_{\mathbb{L}_2} + \gamma \|\boldsymbol{\vartheta}_{n+1} - \boldsymbol{L}_h'\|_{\mathbb{L}_2} \leqslant M_\gamma \eta^2 \big( 0.23\sqrt{\gamma p} + 0.74\|\boldsymbol{v}_n\|_{\mathbb{L}_2} + 0.25\eta\|\boldsymbol{g}_n\|_{\mathbb{L}_2} \big).
\tag{33}
$$

Combining equation 32 and equation 33, we get

$$
x_{n+1} \leqslant \varrho x_n + M_\gamma \eta^2 \big( 0.23\sqrt{\gamma p} + 0.74\|\boldsymbol{v}_n\|_{\mathbb{L}_2} + 0.25\eta\|\boldsymbol{g}_n\|_{\mathbb{L}_2} \big).
$$

From the last display, we infer that

$$
x_n \leqslant \varrho^n x_0 + M_\gamma \eta^2 \sum_{k=0}^{n-1} \varrho^{n-1-k} \big( 0.23\sqrt{\gamma p} + 0.74\|\boldsymbol{v}_k\|_{\mathbb{L}_2} + 0.25\eta\|\boldsymbol{g}_k\|_{\mathbb{L}_2} \big).
$$

This implies that

$$
\begin{aligned}
x_n &\leqslant \varrho^n x_0 + \frac{0.23 M_\gamma \eta^2 \sqrt{\gamma p}}{1 - \varrho} + 0.74 M_\gamma \eta^2 \sum_{k=1}^{n} \varrho^{n-k} \big( \|\boldsymbol{v}_{k-1}\|_{\mathbb{L}_2} + 0.33\eta\|\boldsymbol{g}_{k-1}\|_{\mathbb{L}_2} \big) \\
&\leqslant \varrho^n x_0 + \frac{0.23 M_\gamma \eta^2 \sqrt{\gamma p}}{1 - \varrho} + \frac{0.74 M_\gamma \eta^2}{\sqrt{1 - \varrho}} \left\{ \sum_{k=1}^{n} \varrho^{n-k} \big( \|\boldsymbol{v}_{k-1}\|_{\mathbb{L}_2}^2 + 0.33\eta^2\|\boldsymbol{g}_{k-1}\|_{\mathbb{L}_2}^2 \big) \right\}^{1/2}.
\end{aligned}
$$

In view of equation 29, $\varrho \leqslant 1 - 0.99 m\eta/\gamma$ and

$$
x_n \leqslant \varrho^n x_0 + 0.233 \kappa \eta \sqrt{\gamma p} + 0.74 M_\gamma \eta \left\{ \frac{\gamma \eta}{m \varrho} \sum_{k=0}^{n} \varrho^{n-k} \big( \|\boldsymbol{v}_k\|_{\mathbb{L}_2}^2 + 0.33\eta^2\|\boldsymbol{g}_k\|_{\mathbb{L}_2}^2 \big) \right\}^{1/2}.
$$

**Proposition 2.** *Assume that $\boldsymbol{v}_0 \sim \mathcal{N}(0, \gamma \mathbf{I}_p)$ is independent of $\boldsymbol{\vartheta}_0$. If $\gamma \geqslant 5M$, $\kappa \geqslant 10$ and $\eta \leqslant 1/10$ then*

$$
\eta \sum_{k=0}^{n} \varrho^{n-k} \|\boldsymbol{g}_k\|_{\mathbb{L}_2}^2 \leqslant 4.42 \varrho^n \gamma\, \mathbb{E}[f_0] + \frac{1.11\gamma^2 p}{m} + 4.98\big( x_n + 0.96\sqrt{\gamma p} \big)^2
$$

$$
\eta \sum_{k=0}^{n} \varrho^{n-k} \|\boldsymbol{v}_k\|_{\mathbb{L}_2}^2 \leqslant 3.93 \varrho^n \gamma \mathbb{E}[f_0] + \frac{1.87\gamma^2 p}{m} + 3.2\big( x_n + 0.96\sqrt{\gamma p} \big)^2.
$$

We can apply Proposition 2 and $\varrho \geqslant 0.998$ to infer that

$$
\begin{aligned}
x_n &\leqslant \varrho^n x_0 + 0.233\kappa\eta\sqrt{\gamma p} + 0.74M_\gamma\eta\left\{\frac{\gamma}{m\varrho}\left(3.98\varrho^n\gamma\,\mathbb{E}[f_0] + \frac{1.87\gamma^2 p}{m} + 3.25\left(x_n + 0.96\sqrt{\gamma p}\right)^2\right)\right\}^{1/2} \\
&\leqslant \varrho^n x_0 + 0.233\kappa\eta\sqrt{\gamma p} + 0.74M_\gamma\eta\left\{\frac{\gamma}{m}\left(3.99\varrho^n\gamma\,\mathbb{E}[f_0] + \frac{1.88\gamma^2 p}{m} + 3.26\left(x_n + 0.96\sqrt{\gamma p}\right)^2\right)\right\}^{1/2} \\
&\leqslant \varrho^n x_0 + +0.233\kappa\eta\sqrt{\gamma p} + \frac{1.54M\eta}{\sqrt{m}}\sqrt{\varrho^n\mathbb{E}[f_0]} + 0.62\sqrt{\kappa}\,\eta x_n + 0.74M_\gamma\eta\left(\frac{1.88\gamma^3 p}{m^2} + \frac{3\gamma^2 p}{m}\right)^{1/2} \\
&\leqslant \varrho^n x_0 + 0.62\sqrt{\kappa}\,\eta x_n + \frac{1.54M\eta}{\sqrt{m}}\sqrt{\varrho^n\mathbb{E}[f_0]} + \frac{M\eta\sqrt{\gamma p}}{m}\left(0.233 + 0.74\sqrt{1.94}\right).
\end{aligned}
$$

Therefore, under the condition $\sqrt{\kappa}\,\eta \leqslant 0.1$,

$$
x_n \leqslant 1.07\varrho^n x_0 + \frac{1.65M\eta}{\sqrt{m}}\sqrt{\varrho^n\mathbb{E}[f_0]} + \frac{1.35M\eta\sqrt{\gamma p}}{m}.
$$

Finally, one can check that $2x_n^2 \geqslant \gamma^2\|\boldsymbol{\vartheta}_n - \bar{\boldsymbol{\vartheta}}_n\|_{\mathbb{L}_2}^2 \geqslant \gamma^2\mathsf{W}_2^2(\nu_n^{\mathsf{KLMC}}, \pi)$ and $x_0 = \gamma\|\boldsymbol{\vartheta}_0 - \boldsymbol{L}_0\|_{\mathbb{L}_2} = \gamma\mathsf{W}_2(\nu_0, \pi)$. This completes the proof of the theorem.

### B.3  PROOF OF PROPOSITION 2 (DISCOUNTED SUMS OF SQUARED GRADIENTS AND VELOCITIES)

To ease the notation, we set $z_n := \mathbb{E}[\boldsymbol{v}_n^\top \boldsymbol{g}_n]$ and define

$$
\begin{aligned}
S_n(z) &:= \sum_{k=0}^n \varrho^{n-k} z_k, & S_n(g^2) &:= \sum_{k=0}^n \varrho^{n-k}\|\boldsymbol{g}_k\|_{\mathbb{L}_2}^2, \\
S_n(f) &:= \sum_{k=0}^n \varrho^{n-k}\mathbb{E}[f_k], & S_n(v^2) &:= \sum_{k=0}^n \varrho^{n-k}\|\boldsymbol{v}_k\|_{\mathbb{L}_2}^2.
\end{aligned}
$$

Throughout the proof, we will need some technical results that will be stated as lemmas and their proof will be postponed to Appendix B.5.

**Lemma 5.** *If for some $M \geqslant 0$, the gradient $\nabla f$ is $M$-Lipschitz continuous, then for every step-size $h > 0$ and every $\gamma > 0$ it holds for the KLMC iterates defined in equation 28 that*

$$
\left|z_{n+1} - (1 - \alpha\eta)z_n + \alpha\eta\|\boldsymbol{g}_n\|_{\mathbb{L}_2}^2\right| \leqslant \eta M_\gamma\left(\|\boldsymbol{v}_n\|_{\mathbb{L}_2}^2 + \tfrac{5}{8}\eta^2\|\boldsymbol{g}_n\|_{\mathbb{L}_2}^2 + \tfrac{4}{3}\eta\gamma p - \tilde{\alpha}\eta z_n\right)
$$

*for some positive number $\tilde{\alpha}\eta \leqslant 0.14$.*

Since $M_\gamma \leqslant 1/5$ and $\eta \leqslant 0.1$, we have

$$
\alpha - \tfrac{5}{8}M_\gamma\eta^2 \geqslant \tfrac{1}{\eta}(1 - e^{-\eta}) - \tfrac{1}{8}\eta^2 \geqslant 10(1 - e^{-0.1}) - \tfrac{1}{8}0.1^2 \geqslant 0.94.
$$

Therefore, we can rewrite the claim of Lemma 5 with the notation $\tilde{\beta} = \eta M_\gamma\tilde{\alpha}$ as follows:

$$
z_{n+1} \leqslant (1 - \alpha\eta - \tilde{\beta}\eta)z_n + 0.2\eta\|\boldsymbol{v}_n\|_{\mathbb{L}_2}^2 + 0.67\gamma p\eta^2 - 0.94\eta\|\boldsymbol{g}_n\|_{\mathbb{L}_2}^2.
$$

**Lemma 6.** *Let $\tilde{\beta} \leqslant 0.014$ and $\eta \in [0, 0.1]$. If $z_0 = 0$ and the sequences $\{z_n\} \subset \mathbb{R}$, $\{\boldsymbol{v}_n\} \subset \mathbb{R}^p$ and $\{\boldsymbol{g}_n\} \subset \mathbb{R}^p$ satisfy the inequality*

$$
z_{n+1} \leqslant (e^{-\eta} - \tilde{\beta}\eta)z_n + 0.2\eta\|\boldsymbol{v}_n\|_{\mathbb{L}_2}^2 + 0.67\gamma p\eta^2 - 0.94\eta\|\boldsymbol{g}_n\|_{\mathbb{L}_2} \tag{34}
$$

*then for every $\varrho \in [0, 1]$ such that $\varrho \geqslant e^{-\eta}$, it holds that*

$$
S_n(g^2) \leqslant 1.09\alpha(S_n(z))_- + 0.213S_n(v^2) + \frac{0.73\eta\gamma p}{1 - \varrho} - \frac{1.07z_{n+1}}{\eta}, \tag{35}
$$

*where $(S_n(z))_- = \max(0, S_n(z))$ is the negative part of $S_n(z)$ and $\alpha = (1 - e^{-\eta})/\eta$.*

In order to get rid of the last term in equation 35, we need a bound on $(S_n(z))_-$. To this end, we use the smoothness of the function $f$, in conjunction with equation 28, to infer that

$$
\begin{aligned}
2\gamma\mathbb{E}[f_{n+1} - f_n] &\leqslant 2\gamma\mathbb{E}[\boldsymbol{g}_n^\top(\boldsymbol{\vartheta}_{n+1} - \boldsymbol{\vartheta}_n)] + M_\gamma\|\gamma(\boldsymbol{\vartheta}_{n+1} - \boldsymbol{\vartheta}_n)\|_{\mathbb{L}_2}^2 \\
&\leqslant 2\alpha\eta(1 - \beta M_\gamma\eta^2)z_n - \beta\eta^2(2 - \beta M_\gamma\eta^2)\|\boldsymbol{g}_n\|_{\mathbb{L}_2}^2 + M_\gamma\alpha^2\eta^2\|\boldsymbol{v}_n\|_{\mathbb{L}_2}^2 + \tfrac{2}{3}M_\gamma\eta^3\gamma p \\
&\leqslant 2\alpha_0\eta z_n - 0.96\eta^2\|\boldsymbol{g}_n\|_{\mathbb{L}_2}^2 + 0.0182\eta\|\boldsymbol{v}_n\|_{\mathbb{L}_2}^2 + \tfrac{2}{15}\eta^3\gamma p
\end{aligned}
$$

with $\alpha_0 = \alpha(1 - \beta M_\gamma\eta^2) \geqslant \alpha(1 - 0.1 \times 0.1^2) \geqslant 0.999\alpha$.

**Lemma 7.** *Let $\alpha_0, \gamma, \eta > 0$. If the sequences $\{F_n\} \subset \mathbb{R}$, $\{\boldsymbol{g}_n\} \subset \mathbb{R}^p$ and $\{\boldsymbol{v}_n\} \subset \mathbb{R}^p$ satisfy $F_n \geqslant 0$ and*

$$
2(F_{n+1} - F_n) \leqslant 2\alpha_0\eta z_n + 0.0182\eta\|\boldsymbol{v}_n\|_{\mathbb{L}_2}^2 + \tfrac{2}{15}\eta^3\gamma p \tag{36}
$$

*then, for every $\varrho \in (0, 1)$, it holds that*

$$
\alpha_0(\eta S_n(z))_- \leqslant \varrho^n F_0 + (1 - \varrho)\gamma S_n(F) + 0.0182\eta S_n(v^2) + \frac{2\eta^3\gamma p}{15(1 - \varrho)}.
$$

In view of the strong convexity of the potential function and the assumption that $f(\boldsymbol{\theta}_*) = 0$, the Polyak-Lojasiewicz inequality

$$
f_n \leqslant \tfrac{1}{2m}\|\boldsymbol{g}_n\|^2
$$

holds true. This implies that $(1 - \varrho)S_n(f) \leqslant (\eta/\gamma)mS_n(f) \leqslant \tfrac{1}{2}(\eta/\gamma)S_n(g^2)$. Combining this inequality with the claim of Lemma 7, applied to $F_n = \gamma\mathbb{E}[f_n]$, we get

$$
\boxed{0.999\alpha(S_n(z))_- \leqslant \frac{\varrho^n\gamma}{\eta}\mathbb{E}[f_0] + 0.5S_n(g^2) + 0.0182S_n(v^2) + \frac{0.14\eta\gamma^2 p}{m}.} \tag{37}
$$

Let us now combine equation 35 and equation 37:

$$
\begin{aligned}
S_n(g^2) &\leqslant \frac{1.1\varrho^n\gamma}{\eta}\mathbb{E}[f_0] + 0.55S_n(g^2) + 0.02S_n(v^2) + \frac{0.16\eta\gamma^2 p}{m} \\
&\quad + 0.213S_n(v^2) + \frac{0.73\gamma^2 p}{m} + \frac{1.07|z_{n+1}|}{\eta} \\
&\leqslant \frac{1.1\varrho^n\gamma}{\eta}\mathbb{E}[f_0] + 0.55S_n(g^2) + 0.223S_n(v^2) + \frac{0.75\gamma^2 p}{m} + \frac{1.07|z_{n+1}|}{\eta}.
\end{aligned}
$$

Subtracting $0.55S_n(g^2)$ from both sides and dividing by $0.45$, we obtain

$$
\boxed{S_n(g^2) \leqslant \frac{2.45\varrho^n\gamma\mathbb{E}[f_0]}{\eta} + 0.5S_n(v^2) + \frac{1.7\gamma^2 p}{m} + \frac{2.38|z_{n+1}|}{\eta}.} \tag{38}
$$

Let us now derive a bound for $S_n(v^2)$. We start with the following property, which is a direct consequence of the definition of $\boldsymbol{v}_{n+1}$:

$$
\|\boldsymbol{v}_{n+1}\|_{\mathbb{L}_2}^2 - \|\boldsymbol{v}_n\|_{\mathbb{L}_2}^2 \leqslant -\alpha\eta(2 - \alpha\eta)\|\boldsymbol{v}_n\|_{\mathbb{L}_2}^2 - 2\alpha\eta(1 - \alpha\eta)z_n + \alpha^2\eta^2\|\boldsymbol{g}_n\|_{\mathbb{L}_2}^2 + 2\eta\gamma p.
$$

Using the same technique as before and applying Lemma 3, we deduce the following:

$$
\begin{aligned}
(\varrho - 1)S_n(v^2) - \varrho^n\|\boldsymbol{v}_0\|_{\mathbb{L}_2}^2 &= (\varrho - 1)S_n(v^2) - \varrho^n\gamma p \\
&\leqslant -\alpha\eta(2 - \alpha\eta)S_n(v^2) - 2\alpha\eta(1 - \alpha\eta)S_n(z) + \alpha^2\eta^2 S_n(g^2) + \frac{2\eta\gamma p}{1 - \varrho}.
\end{aligned}
$$

Therefore, since $\varrho \geqslant 1 - \tfrac{m}{\gamma}\eta$,

$$
(2\alpha - \alpha^2\eta - \tfrac{m}{\gamma})\eta S_n(v^2) \leqslant -2\alpha\eta(1 - \alpha\eta)S_n(z) + (\alpha\eta)^2 S_n(g^2) + \frac{2.021\gamma^2 p}{m}.
$$

Since $\alpha = (1 - e^{-\eta})/\eta$ with $\eta \leqslant 0.1$, from the last display we infer that

$$
S_n(v^2) \leqslant 1.02\alpha\big(S_n(z)\big)_- + 0.51\eta S_n(g^2) + \frac{1.13\gamma^2 p}{m\eta}.
$$

Combining this inequality with equation 37, implies

$$S_n(v^2) \leqslant \frac{1.03\varrho^n\gamma}{\eta}[f_0] + 0.562S_n(g^2) + 0.019S_n(v^2) + \frac{0.2\eta\gamma^2 p}{m} + 0.51\eta S_n(g^2) + \frac{1.13\gamma^2 p}{m\eta}$$

$$\leqslant \frac{1.03\varrho^n\gamma}{\eta}f_0 + 0.62S_n(g^2) + 0.019S_n(v^2) + \frac{1.13\gamma^2 p}{m\eta}$$

Therefore, subtracting $0.019S_n(v^2)$ and dividing by $(1 - 0.019)$, we get

$$S_n(v^2) \leqslant \frac{1.1\varrho^n\gamma}{\eta}f_0 + 0.64S_n(g^2) + \frac{1.16\gamma^2 p}{m\eta} \tag{39}$$

Combining equation 38 and equation 39, we arrive at

$$S_n(v^2) \leqslant \frac{1.1\varrho^n\gamma}{\eta}\mathbb{E}[f_0] + \frac{1.16\gamma^2 p}{m\eta} + 0.64\Big(\frac{2.45\varrho^n\gamma}{\eta}\mathbb{E}[f_0] + 0.5S_n(v^2) + \frac{1.7\gamma^2 p}{m} + \frac{2.38|z_{n+1}|}{\eta}\Big)$$

$$\leqslant \frac{2.67\varrho^n\gamma}{\eta}f_0 + \frac{1.27\gamma^2 p}{m\eta} + \frac{1.53|z_{n+1}|}{\eta} + 0.32S_n(v^2).$$

Therefore, subtracting $0.32S_n(v^2)$ and dividing by $(1 - 0.32)$, we get

$$S_n(v^2) \leqslant \frac{3.93\varrho^n\gamma}{\eta}f_0 + \frac{1.87\gamma^2 p}{m\eta} + \frac{2.25|z_{n+1}|}{\eta}.$$

Once again, combining with equation 38, we get

$$S_n(g^2) \leqslant \frac{2.45\varrho^n\gamma}{\eta}\mathbb{E}[f_0] + 0.5\Big(\frac{3.93\varrho^n\gamma}{\eta}f_0 + \frac{1.87\gamma^2 p}{m\eta} + \frac{2.25|z_{n+1}|}{\eta}\Big) + \frac{1.7\gamma^2 p}{m} + \frac{2.38|z_{n+1}|}{\eta}$$

that leads to

$$S_n(g^2) \leqslant \frac{4.42\varrho^n\gamma}{\eta}\mathbb{E}[f_0] + \frac{1.11\gamma^2 p}{m\eta} + \frac{3.51|z_{n+1}|}{\eta}. \tag{40}$$

The last lemma we need is the one providing an upper bound on $|z_{n+1}|$.

**Lemma 8.** *For every $\eta \leqslant 0.1$ and $\gamma \geqslant 5M$, we have*

$$|z_{n+1}| \leqslant \big(1.19x_n + 1.14\sqrt{\gamma p}\big)^2,$$

*where $x_n$ is given by equation 31.*

Using Lemma 8 in conjunction with equation 39 and equation 40, we arrive at the inequalities stated in Proposition 2.

### B.4 PROOF OF LEMMA 4 (ONE-STEP DISCRETIZATION ERROR)

We use the notation

$$\psi_0(t) = e^{-\gamma t}, \qquad \psi_1(t) = \frac{1 - e^{-\gamma t}}{\gamma}, \qquad \psi_2(t) = \frac{e^{-\gamma t} - 1 + \gamma t}{\gamma}$$

and note that

$$\psi_1(t) \leqslant t, \qquad \psi_2(t) \leqslant 0.5\gamma t^2.$$

Furthermore,

$$\|\boldsymbol{v}_{n+1} - \boldsymbol{V}_h'\|_{\mathbb{L}_2} = \gamma\bigg\|\int_0^h e^{-\gamma(h-s)}\big(\nabla f(\boldsymbol{L}_s') - \nabla f(\boldsymbol{\vartheta}_n)\big)\,\mathrm{d}s\bigg\|_{\mathbb{L}_2}$$

$$\leqslant \gamma\int_0^h e^{-\gamma(h-s)}\big\|\nabla f(\boldsymbol{L}_s') - \nabla f(\boldsymbol{\vartheta}_n)\big\|_{\mathbb{L}_2}\,\mathrm{d}s$$

$$\leqslant M\gamma\int_0^h \big\|\boldsymbol{L}_s' - \boldsymbol{\vartheta}_n\big\|_{\mathbb{L}_2}\,\mathrm{d}s, \tag{41}$$

where the last implication is due to the $M$-smoothness of the potential function $f$. On the one hand, for every $s \in [0, h]$, we have

$$
\begin{aligned}
\boldsymbol{L}'_s - \boldsymbol{\vartheta}_n &= \int_0^s \boldsymbol{V}'_u \, \mathrm{d}u \\
&= \psi_1(s)\, \boldsymbol{v}_n - \gamma \int_0^s \int_0^u e^{-\gamma(u-t)} \big(\nabla f(\boldsymbol{L}'_t) - \nabla f(\boldsymbol{\vartheta}_n)\big) \, \mathrm{d}t \mathrm{d}u - \psi_2(s)\, \boldsymbol{g}_n \\
&\quad + \sqrt{2}\,\gamma \int_0^s \int_0^u e^{-\gamma(u-t)} \, \mathrm{d}\boldsymbol{W}_t \, \mathrm{d}u \\
&= \psi_1(s)\, \boldsymbol{v}_n - \psi_2(s)\, \boldsymbol{g}_n + \sqrt{2}\,\gamma \int_0^s \psi_1(t) \, \mathrm{d}\boldsymbol{W}_t \\
&\quad - \gamma \int_0^s \psi_1(s-t) \big(\nabla f(\boldsymbol{L}'_t) - \nabla f(\boldsymbol{\vartheta}_n)\big) \, \mathrm{d}t.
\end{aligned}
$$

Therefore,

$$
\big\|\boldsymbol{L}'_s - \boldsymbol{\vartheta}_n\big\|_{\mathbb{L}_2} \leqslant s \,\|\boldsymbol{v}_n\|_{\mathbb{L}_2} + 0.5\gamma s^2 \,\|\boldsymbol{g}_n\|_{\mathbb{L}_2} + \sqrt{2ps/3}\,\gamma s + M\gamma \int_0^s (s-t)\big\|\boldsymbol{L}'_t - \boldsymbol{\vartheta}_n\big\|_{\mathbb{L}_2} \, \mathrm{d}t.
$$

The last inequality combined with $s - t \leqslant h - t$ allows us to use the Grönwall lemma, which implies that

$$
\begin{aligned}
\big\|\boldsymbol{L}'_s - \boldsymbol{\vartheta}_n\big\|_{\mathbb{L}_2} &\leqslant \big(s \,\|\boldsymbol{v}_n\|_{\mathbb{L}_2} + 0.5\gamma s^2 \,\|\boldsymbol{g}_n\|_{\mathbb{L}_2} + \sqrt{(2/3)ps}\,\gamma s\big) e^{M\gamma s(h-0.5s)} \\
&\leqslant \big(s\|\boldsymbol{v}_n\|_{\mathbb{L}_2} + 0.5\gamma s^2 \,\|\boldsymbol{g}_n\|_{\mathbb{L}_2} + \sqrt{(2/3)ps}\,\gamma s\big) e^{0.5M\gamma h^2}.
\end{aligned}
\tag{42}
$$

Combining the last display with equation 41, we get

$$
\begin{aligned}
\big\|\boldsymbol{v}_{n+1} - \boldsymbol{V}'_h\big\|_{\mathbb{L}_2} &\leqslant \big\{ \tfrac{1}{2}\|\boldsymbol{v}_n\|_{\mathbb{L}_2} + \tfrac{1}{6}\eta\|\boldsymbol{g}_n\|_{\mathbb{L}_2} + 0.33\sqrt{\gamma p\eta} \big\} Mh^2 \gamma e^{M\gamma h^2/2} \\
&\leqslant \big\{ \tfrac{1}{2}\|\boldsymbol{v}_n\|_{\mathbb{L}_2} + \tfrac{1}{6}\eta\|\boldsymbol{g}_n\|_{\mathbb{L}_2} + 0.33\sqrt{\gamma p\eta} \big\} M_\gamma \eta^2 e^{M_\gamma \eta^2/2}.
\end{aligned}
$$

This completes the proof of the first inequality. To prove the second one, we again use the update rules of $\boldsymbol{\theta}_{n+1}$ and $\boldsymbol{L}'$:

$$
\begin{aligned}
\big\|\boldsymbol{\vartheta}_{n+1} - \boldsymbol{L}'_h\big\|_{\mathbb{L}_2} &= \gamma \bigg\| \int_0^h \int_0^t e^{-\gamma^2(t-s)} \big(\nabla f(\boldsymbol{L}'_s) - \nabla f(\boldsymbol{\vartheta}_n)\big) \, \mathrm{d}s \, \mathrm{d}t \bigg\|_{\mathbb{L}_2} \\
&\leqslant \gamma \int_0^h \int_0^t \big\|\nabla f(\boldsymbol{L}'_s) - \nabla f(\boldsymbol{\vartheta}_n)\big\|_{\mathbb{L}_2} \, \mathrm{d}s \, \mathrm{d}t \\
&\leqslant M\gamma \int_0^h \int_0^t \big\|\boldsymbol{L}'_s - \boldsymbol{\vartheta}_n\big\|_{\mathbb{L}_2} \, \mathrm{d}s \, \mathrm{d}t.
\end{aligned}
$$

The last term can be bounded using equation 42. This yields

$$
\begin{aligned}
\gamma\big\|\boldsymbol{\vartheta}_{n+1} - \boldsymbol{L}'_h\big\|_{\mathbb{L}_2} &\leqslant M_\gamma \gamma^3 e^{M_\gamma \eta^2/2} \int_0^h \int_0^t \big(s\|\boldsymbol{v}_n\|_{\mathbb{L}_2} + 0.5\gamma s^2\|\boldsymbol{g}_n\|_{\mathbb{L}_2} + \sqrt{(2/3)ps^3}\,\gamma\big) \, \mathrm{d}s \, \mathrm{d}t \\
&\leqslant M_\gamma \eta^3 e^{M_\gamma \eta^2/2} \big(\tfrac{1}{6}\|\boldsymbol{v}_n\|_{\mathbb{L}_2} + \tfrac{1}{24}\eta\|\boldsymbol{g}_n\|_{\mathbb{L}_2} + 0.1\sqrt{p\eta\gamma}\big) \\
&\leqslant (1/6)M_\gamma \eta^3 e^{M_\gamma \eta^2/2} \big(\|\boldsymbol{v}_n\|_{\mathbb{L}_2} + 0.25\eta\|\boldsymbol{g}_n\|_{\mathbb{L}_2} + 0.6\sqrt{p\eta\gamma}\big)
\end{aligned}
$$

as desired.

## B.5 PROOFS OF THE TECHNICAL LEMMAS USED IN PROPOSITION 2

### B.5.1 PROOF OF LEMMA 5

Since $z_n = \mathbb{E}[\boldsymbol{g}_n^\top \boldsymbol{v}_n]$, we have

$$
\big| z_{n+1} - z_n - \mathbb{E}[\boldsymbol{g}_n^\top (\boldsymbol{v}_{n+1} - \boldsymbol{v}_n)] \big| = \big| \mathbb{E}\big[(\boldsymbol{g}_{n+1} - \boldsymbol{g}_n)^\top \boldsymbol{v}_{n+1}\big] \big|.
\tag{43}
$$

On the one hand, definition equation 27 of $\boldsymbol{v}_{n+1}$ yields

$$\mathbb{E}[\boldsymbol{g}_n^\top(\boldsymbol{v}_{n+1} - \boldsymbol{v}_n)] = -\alpha\eta\mathbb{E}[\boldsymbol{g}_n^\top\boldsymbol{v}_n] - \alpha\eta\|\boldsymbol{g}_n\|_{\mathbb{L}_2}^2. \tag{44}$$

On the other hand, the Cauchy-Schwartz inequality implies

$$\begin{aligned}
\left|\mathbb{E}\left[(\boldsymbol{g}_{n+1} - \boldsymbol{g}_n)^\top\boldsymbol{v}_{n+1}\right]\right| &\leqslant \left\|\boldsymbol{g}_{n+1} - \boldsymbol{g}_n\right\|_{\mathbb{L}_2}\|\boldsymbol{v}_{n+1}\|_{\mathbb{L}_2} \\
&\leqslant M\|\boldsymbol{\vartheta}_{n+1} - \boldsymbol{\vartheta}_n\|_{\mathbb{L}_2}\|\boldsymbol{v}_{n+1}\|_{\mathbb{L}_2}.
\end{aligned}$$

Similarly, using update rules equation 27 and equation 28 of the KLMC, and the triangle inequality we get

$$\begin{aligned}
\gamma^2\|\boldsymbol{\vartheta}_{n+1} - \boldsymbol{\vartheta}_n\|_{\mathbb{L}_2}^2 &\leqslant \eta^2\|\boldsymbol{v}_n\|_{\mathbb{L}_2}^2 + \tfrac{1}{4}\eta^4\|\boldsymbol{g}_n\|_{\mathbb{L}_2}^2 - 2\alpha\beta\eta^2 z_n + \tfrac{2}{3}\eta^3\gamma p \\
\|\boldsymbol{v}_{n+1}\|_{\mathbb{L}_2}^2 &\leqslant \|\boldsymbol{v}_n\|_{\mathbb{L}_2}^2 + \eta^2\|\boldsymbol{g}_n\|_{\mathbb{L}_2}^2 - 2\alpha\eta(1 - \alpha\eta)z_n + 2\eta\gamma p.
\end{aligned}$$

Hence,

$$\begin{aligned}
\tfrac{\gamma}{\eta}\|\boldsymbol{\vartheta}_{n+1} - \boldsymbol{\vartheta}_n\|_{\mathbb{L}_2}\|\boldsymbol{v}_{n+1}\|_{\mathbb{L}_2} &\leqslant \frac{(\gamma/\eta)^2\|\boldsymbol{\vartheta}_{n+1} - \boldsymbol{\vartheta}_n\|_{\mathbb{L}_2}^2 + \|\boldsymbol{v}_{n+1}\|_{\mathbb{L}_2}^2}{2} \\
&\leqslant \|\boldsymbol{v}_n\|_{\mathbb{L}_2}^2 + \tfrac{5}{8}\eta^2\|\boldsymbol{g}_n\|_{\mathbb{L}_2}^2 - \alpha\eta(\beta + 1 - \alpha\eta)z_n + \tfrac{4}{3}\eta\gamma p. \tag{45}
\end{aligned}$$

Therefore, combining equation 43, equation 44 and equation 45, we get

$$\begin{aligned}
\left|z_{n+1} - (1 - \alpha\eta)z_n + \alpha\eta\|\boldsymbol{g}_n\|_{\mathbb{L}_2}^2\right| &\leqslant \eta M_\gamma\left(\|\boldsymbol{v}_n\|_{\mathbb{L}_2}^2 + \tfrac{5}{8}\eta^2\|\boldsymbol{g}_n\|_{\mathbb{L}_2}^2 + \tfrac{4}{3}\eta\gamma p\right) \\
&\quad - \eta^2 M_\gamma\underbrace{\alpha(\beta + 1 - \alpha\eta)}_{:=\tilde{\alpha}}z_n
\end{aligned}$$

with $\tilde{\alpha}\eta \leqslant \alpha\eta(1.5 - \alpha\eta) \leqslant 0.14$, as desired.

### B.5.2 PROOF OF LEMMA 6

We apply inequality equation 34 for every index $k \leqslant n$ multiply each side by $\varrho^{n-k}$:

$$\varrho^{n-k}z_{k+1} \leqslant (e^{-\eta} - \tilde{\beta}\eta)\varrho^{n-k}z_k + 0.2\varrho^{n-k}\eta\|\boldsymbol{v}_k\|_{\mathbb{L}_2}^2 + 0.67\eta^2\gamma p\varrho^{n-k} - 0.94\eta\varrho^{n-k}\|\boldsymbol{g}_k\|_{\mathbb{L}_2}^2.$$

Summing over $k$, applying Lemma 3 and taking into account that $z_0 = 0$, we get

$$\begin{aligned}
z_{n+1} + \varrho S_n(z) &\leqslant (e^{-\eta} - \tilde{\beta}\eta)S_n(z) + 0.2\eta S_n(v^2) + \frac{0.67\eta^2\gamma p}{1 - \varrho} - 0.94\eta S_n(g^2) \\
&\leqslant (e^{-\eta} - \tilde{\beta}\eta)S_n(z) + 0.2\eta S_n(v^2) + \frac{0.68\eta^2\gamma p}{1 - \varrho} - 0.94\eta S_n(g^2).
\end{aligned}$$

This implies that

$$0.94\eta S_n(g^2) \leqslant (\varrho - e^{-\eta} + \tilde{\beta}\eta)(-S_n(z)) + 0.2\eta S_n(v^2) + \frac{0.68\eta^2\gamma p}{1 - \varrho} - z_{n+1}.$$

Note that $\rho - e^{-\eta} \geqslant 0$ and $\varrho - e^{-\eta} + \tilde{\beta}\eta \leqslant 1 - e^{-\eta} + 0.014\eta \leqslant 1.02(e^{-\eta} - 1) = 1.02\alpha\eta$. Therefore,

$$0.94\eta S_n(g^2) \leqslant 1.02\alpha\eta(S_n(z))_- + 0.2\eta S_n(v^2) + \frac{0.68\eta^2\gamma p}{1 - \varrho} - z_{n+1}.$$

Dividing both sides of the last display by $0.94\eta$, we get

$$S_n(g^2) \leqslant 1.09\alpha(S_n(z))_- + 0.213 S_n(v^2) + \frac{0.73\eta\gamma p}{1 - \varrho} - \frac{1.07 z_{n+1}}{\eta}.$$

This completes the proof of the lemma.

### B.5.3 PROOF OF LEMMA 7

We write inequality equation 36 for all indices $k$ and multiply both sides of it by $\varrho^{n-k}$. Summing the obtained inequalities and applying Lemma 3, we obtain the following:

$$2(\varrho - 1)S_n(F) - 2\varrho^n F_0 \leqslant 2\alpha_0 \eta S_n(z) + 0.0182\eta S_n(v^2) + \frac{2\eta^3 \gamma p}{15(1 - \varrho)},$$

where the left-hand side is obtained using Lemma 3 and the fact that $F_n \geqslant 0$. Rearranging the terms and dividing by 2, we obtain

$$-\alpha_0 \eta S_n(z) \leqslant \varrho^n F_0 + (1 - \varrho)S_n(F) + 0.0182\eta S_n(v^2) + \frac{2\eta^3 \gamma p}{15(1 - \varrho)}.$$

Since the right-hand side of the last display is nonnegative, we infer that

$$\alpha_0(\eta S_n(z))_- \leqslant \varrho^n F_0 + (1 - \varrho)S_n(F) + 0.0182\eta S_n(v^2) + \frac{2\eta^3 \gamma p}{15(1 - \varrho)},$$

which coincides with the claim of the lemma.

### B.5.4 PROOF OF LEMMA 8

In view of Lemma 5, we have

$$
\begin{aligned}
|z_{n+1}| &\leqslant (e^{-\eta} + 0.3\eta^2)|z_n| + \eta\|\boldsymbol{g}_n\|_{\mathbb{L}_2}^2 + \eta\big(0.2\|\boldsymbol{v}_n\|_{\mathbb{L}_2}^2 + 0.2\eta^2\|\boldsymbol{g}_n\|_{\mathbb{L}_2}^2 + 0.027\eta\gamma p\big) \\
&\leqslant 0.56(\|\boldsymbol{g}_n\|_{\mathbb{L}_2} + \|\boldsymbol{v}_n\|_{\mathbb{L}_2})^2 + 0.0027\gamma p \\
&\leqslant 0.56\big(\|\boldsymbol{g}_n - \nabla f(\boldsymbol{L}_{nh})\|_{\mathbb{L}_2} + \|\boldsymbol{v}_n - \boldsymbol{V}_{nh}\|_{\mathbb{L}_2} + \sqrt{Mp} + \sqrt{\gamma p}\big)^2 + 0.0027\gamma p,
\end{aligned}
$$

where we have used the facts $\|\nabla f(\boldsymbol{L}_{nh})\|_{\mathbb{L}_2} = \int \|\nabla f\|^2 \, d\pi \leqslant Mp$ (Dalalyan & Karagulyan, 2019, Lemma 3) and $\mathbb{E}[\|\boldsymbol{V}_{nh}\|^2] = \gamma p$. Finally, one can note that

$$
\begin{aligned}
\|\boldsymbol{g}_n - \nabla f(\boldsymbol{L}_{nh})\|_{\mathbb{L}_2} + \|\boldsymbol{v}_n - \boldsymbol{V}_{nh}\|_{\mathbb{L}_2} &\leqslant 0.5\gamma\|\boldsymbol{\vartheta}_n - \boldsymbol{L}_{nh}\|_{\mathbb{L}_2} + \|\boldsymbol{v}_n - \boldsymbol{V}_{nh}\|_{\mathbb{L}_2} \\
&\leqslant 0.5\|\boldsymbol{v}_n - \boldsymbol{V}_{nh} + \gamma(\boldsymbol{\vartheta}_n - \boldsymbol{L}_{nh})\|_{\mathbb{L}_2} + 1.5\|\boldsymbol{v}_n - \boldsymbol{V}_{nh}\|_{\mathbb{L}_2} \\
&\leqslant \sqrt{5/2}\, x_n.
\end{aligned}
$$

Therefore,

$$|z_{n+1}| \leqslant \big(1.19x_n + 1.09\sqrt{\gamma p}\big)^2 + 0.0027\gamma p \leqslant \big(1.19x_n + 1.14\sqrt{\gamma p}\big)^2.$$

This completes the proof of the lemma.

## C    THE PROOF OF THE UPPER BOUND ON THE ERROR OF RKLMC

Consider the underdamped Langevin diffusion

$$d\boldsymbol{L}_t = \boldsymbol{V}_t\, dt, \qquad \text{where} \qquad d\boldsymbol{V}_t = -\gamma \boldsymbol{V}_t\, dt - \gamma \nabla f(\boldsymbol{L}_t)\, dt + \sqrt{2}\gamma\, d\boldsymbol{W}_t \tag{46}$$

for every $t \geqslant 0$, with given initial conditions $\boldsymbol{L}_0$ and $\boldsymbol{V}_0$. Throughout this section, we assume that $\boldsymbol{V}_0 \sim \mathcal{N}_p(0, \gamma \mathbf{I}_p)$ is independent of $\boldsymbol{L}_0$, and the couple $(\boldsymbol{V}_0, \boldsymbol{L}_0)$ is independent of the Brownian motion $\boldsymbol{W}$. We also assume that $\boldsymbol{L}_0$ is drawn from the target distribution $\pi$; this implies that the process $(\boldsymbol{L}_t, \boldsymbol{V}_t)$ is stationary.

In the sequel, we use the following shorthand notation

$$\eta = \gamma h, \quad g = \nabla f, \qquad f_n = f(\boldsymbol{\vartheta}_n), \qquad \boldsymbol{g}_n = g(\boldsymbol{\vartheta}_n), \qquad \boldsymbol{g}_{n+U} = g(\boldsymbol{\vartheta}_{n+U}), \qquad M_\gamma = M/\gamma.$$

The randomized midpoint discretization—proposed and studied in (Shen & Lee, 2019)—of the kinetic Langevin process equation 51, can be written as

$$\boldsymbol{\vartheta}_{n+U} = \boldsymbol{\vartheta}_n + \frac{1 - e^{-U\eta}}{\gamma}\boldsymbol{v}_n - \int_0^{Uh}(1 - e^{-\gamma(Uh-s)})\, ds\, \nabla f_n + \sqrt{2}\int_0^{Uh}(1 - e^{-\gamma(Uh-s)})\, d\bar{\boldsymbol{W}}_s$$

$$\boldsymbol{\vartheta}_{n+1} = \boldsymbol{\vartheta}_n + \frac{1 - e^{-\eta}}{\gamma}\boldsymbol{v}_n - \eta\frac{1 - e^{-\eta(1-U)}}{\gamma}\nabla f_{n+U} + \sqrt{2}\int_0^h(1 - e^{-\gamma(h-s)})\, d\bar{\boldsymbol{W}}_s$$

$$\boldsymbol{v}_{n+1} = \boldsymbol{v}_n e^{-\eta} - \eta e^{-\gamma(h-Uh)}\nabla f_{n+U} + \sqrt{2}\gamma\int_0^h e^{-\gamma(h-s)}\, d\bar{\boldsymbol{W}}_s \tag{47}$$

where $\bar{\boldsymbol{W}}_s = \boldsymbol{W}_{nh+s} - \boldsymbol{W}_{nh}$. We rewrite these relations in the shorter form

$$\boldsymbol{\vartheta}_{n+U} = \boldsymbol{\vartheta}_n + \gamma^{-1}\eta\big(U\bar{\alpha}_1\boldsymbol{v}_n - U^2\eta\bar{\beta}_1\boldsymbol{g}_n + U\sqrt{2U\gamma\eta}\,\bar{\sigma}_1\boldsymbol{\xi}_1\big) \tag{48}$$

$$\boldsymbol{\vartheta}_{n+1} = \boldsymbol{\vartheta}_n + \gamma^{-1}\eta\big(\bar{\alpha}_2\boldsymbol{v}_n - \eta\bar{\beta}_2\boldsymbol{g}_{n+U} + \sqrt{2\gamma\eta}\,\bar{\sigma}_2\boldsymbol{\xi}_2\big) \tag{49}$$

$$\boldsymbol{v}_{n+1} = \boldsymbol{v}_n - \eta\bar{\alpha}_2\boldsymbol{v}_n - 2\eta\bar{\beta}_3\boldsymbol{g}_{n+U} + \sqrt{2\gamma\eta}\,\bar{\sigma}_3\boldsymbol{\xi}_3 \tag{50}$$

where $\bar{\alpha}_1, \bar{\beta}_1, \bar{\beta}_2, \bar{\beta}_3$ and $\bar{\sigma}_1$ are positive random variables (with randomness inherited from $U$ only) satisfying

$$\bar{\alpha}_1 \leqslant 1, \qquad \bar{\beta}_1 \leqslant 1/2, \qquad \bar{\beta}_2 \leqslant 1 - U \leqslant 1, \qquad \bar{\beta}_3 \leqslant 1/2, \qquad \bar{\sigma}_1^2 \leqslant 1/3$$

and $\mathbb{E}[\bar{\beta}_2] \in [0.468, 0.5]$. Similarly, $\bar{\alpha}_2, \bar{\sigma}_2$ and $\bar{\sigma}_3$ are positive real numbers depending on $\gamma$ and $h$ such that

$$\bar{\alpha}_2 \leqslant 1, \qquad \bar{\sigma}_2^2 \leqslant 1/3, \qquad \bar{\sigma}_3^2 \leqslant 1.$$

We define

$$\bar{\boldsymbol{v}}_{n+1} := \mathbb{E}_U[\boldsymbol{v}_{n+1}], \qquad \bar{\boldsymbol{\vartheta}}_{n+1} := \mathbb{E}_U[\boldsymbol{\vartheta}_{n+1}].$$

The solution to SDE equation 46 starting from $(\boldsymbol{v}_n, \boldsymbol{\vartheta}_n)$ at the $n$-th iteration at time $h$ admits the following integral formulation

$$\boldsymbol{L}_t' = \boldsymbol{\vartheta}_n + \int_0^t \boldsymbol{V}_s'\, ds$$

$$\boldsymbol{V}_t' = \boldsymbol{v}_n e^{-\gamma t} - \gamma\int_0^t e^{-\gamma(t-s)}\nabla f(\boldsymbol{L}_s')\, ds + \sqrt{2}\,\gamma\int_0^t e^{-\gamma(t-s)}d\boldsymbol{W}_{nh+s}. \tag{51}$$

These expressions will be used in the proofs provided in the present section. Furthermore, without loss of generality, we assume that the $f(\boldsymbol{\theta}_*) = \min_{\boldsymbol{\theta}\in\mathbb{R}^p} f(\boldsymbol{\theta}) = 0$.

### C.1    SOME PRELIMINARY RESULTS

We start with some technical results required to prove Theorem 2. They mainly assess the discretisation error as well as discounted sums of squared gradients and velocities.

**Lemma 9** (Precision of the mid-point). *For every $h > 0$, it holds that*

$$\|\boldsymbol{\vartheta}_{n+U} - \boldsymbol{L}'_{Uh}\|_{\mathbb{L}_2} \leqslant \gamma^{-1} M_\gamma \eta^3 e^{M_\gamma \eta^2/2} \Big(0.065\eta\|\boldsymbol{g}_n\|_{\mathbb{L}_2} + (1/6)\|\boldsymbol{v}_n\|_{\mathbb{L}_2} + \sqrt{\eta\gamma p/54}\Big).$$

**Lemma 10** (Discretization error). *Let $(\boldsymbol{L}'_t, \boldsymbol{V}'_t)$ be the exact solution of the kinetic Langevin diffusion starting from $(\boldsymbol{\vartheta}_n, \boldsymbol{v}_n)$. If $\gamma \geqslant M$ and $h > 0$, it holds that*

$$\gamma\|\bar{\boldsymbol{\vartheta}}_{n+1} - \boldsymbol{L}'_h\|_{\mathbb{L}_2} \leqslant \frac{M_\gamma^2 \eta^5 e^{M_\gamma \eta^2/2}}{\sqrt{3}} \Big(0.065\eta\|\boldsymbol{g}_n\|_{\mathbb{L}_2} + (1/6)\|\boldsymbol{v}_n\|_{\mathbb{L}_2} + \sqrt{\eta\gamma p/54}\Big)$$

$$\gamma\|\boldsymbol{\vartheta}_{n+1} - \bar{\boldsymbol{\vartheta}}_{n+1}\|_{\mathbb{L}_2} \leqslant M_\gamma \eta^3 \big(0.26\|\boldsymbol{v}_n\|_{\mathbb{L}_2} + 0.106\sqrt{\eta\gamma p}\big) + \frac{\eta^2}{\sqrt{3}}(0.12 M_\gamma \eta^2 + 1)\|\boldsymbol{g}_n\|_{\mathbb{L}_2}$$

$$\|\bar{\boldsymbol{v}}_{n+1} - \boldsymbol{V}'_h\|_{\mathbb{L}_2} \leqslant M_\gamma^2 \eta^4 e^{M_\gamma \eta^2/2} \Big(0.065\eta\|\boldsymbol{g}_n\|_{\mathbb{L}_2} + (1/6)\|\boldsymbol{v}_n\|_{\mathbb{L}_2} + \sqrt{\eta\gamma p/54}\Big)$$

$$\|\boldsymbol{v}_{n+1} - \bar{\boldsymbol{v}}_{n+1}\|_{\mathbb{L}_2} \leqslant M_\gamma \eta^2 \big(0.82\|\boldsymbol{v}_n\|_{\mathbb{L}_2} + 0.41\sqrt{\eta\gamma p}\big) + \frac{\eta^2}{\sqrt{3}}(0.55 M_\gamma \eta + 1)\big\|\boldsymbol{g}_n\big\|_{\mathbb{L}_2}.$$

**Corollary 4.** *If $\gamma \geqslant 2M$ and $\eta \leqslant 1/5$, it holds that*

$$\begin{aligned}
\gamma\|\bar{\boldsymbol{\vartheta}}_{n+1} - \boldsymbol{L}'_h\|_{\mathbb{L}_2} &\leqslant M_\gamma^2 \eta^5 \big(0.038\eta\|\boldsymbol{g}_n\|_{\mathbb{L}_2} + 0.098\|\boldsymbol{v}_n\|_{\mathbb{L}_2} + 0.084\sqrt{\eta\gamma p}\big), \\
\gamma\|\boldsymbol{\vartheta}_{n+1} - \bar{\boldsymbol{\vartheta}}_{n+1}\|_{\mathbb{L}_2} &\leqslant \eta^2 \big(0.578\|\boldsymbol{g}_n\|_{\mathbb{L}_2} + 0.02\|\boldsymbol{v}_n\|_{\mathbb{L}_2} + 0.005\sqrt{\eta\gamma p}\big), \\
\|\bar{\boldsymbol{v}}_{n+1} - \boldsymbol{V}'_h\|_{\mathbb{L}_2} &\leqslant M_\gamma^2 \eta^4 \big(0.066\eta\|\boldsymbol{g}_n\|_{\mathbb{L}_2} + 0.168\|\boldsymbol{v}_n\|_{\mathbb{L}_2} + 0.137\sqrt{\eta\gamma p}\big), \\
\|\boldsymbol{v}_{n+1} - \bar{\boldsymbol{v}}_{n+1}\|_{\mathbb{L}_2} &\leqslant \eta^2 \big(0.591\big\|\boldsymbol{g}_n\big\|_{\mathbb{L}_2} + 0.164\|\boldsymbol{v}_n\|_{\mathbb{L}_2} + 0.082\sqrt{\eta\gamma p}\big).
\end{aligned}$$

**Proposition 3.** *If $\gamma \geqslant 5M$ and $\eta \leqslant 1/5$, then, for any $n \in \mathbb{N}$, the iterates of the RKLMC satisfy*

$$\eta \sum_{k=0}^{n} \varrho^{n-k}\|\boldsymbol{v}_k\|_{\mathbb{L}_2}^2 \leqslant 18.8\varrho^n \gamma \mathbb{E}[f_0] + 3.92(x_n + 1.5\sqrt{\gamma p})^2 + \frac{10.6\gamma^2 p}{m},$$

$$\eta \sum_{k=0}^{n} \varrho^{n-k}\|\boldsymbol{g}_k\|_{\mathbb{L}_2}^2 \leqslant 21.7\varrho^n \gamma \mathbb{E}[f_0] + 4.88(x_n + 1.5\sqrt{\gamma p})^2 + \frac{11.2\gamma^2 p}{m},$$

*where $\varrho = \exp(-mh)$ and $x_n = \big(\|\boldsymbol{v}_n - \boldsymbol{V}_{nh}\|_{\mathbb{L}_2}^2 + \|\boldsymbol{v}_n - \boldsymbol{V}_{nh} + \gamma(\boldsymbol{\vartheta}_n - \boldsymbol{L}_{nh})\|_{\mathbb{L}_2}^2\big)^{1/2}$.*

*Proof of Proposition 3.* We use the same shorthand notation as in the previous proofs and assume without loss of generality that $\boldsymbol{\theta}_* = 0$. Let us define $z_k = \mathbb{E}[\boldsymbol{v}_k^\top \boldsymbol{g}_k]$, and

$$\begin{aligned}
S_n(z) &:= \sum_{k=0}^{n} \varrho^{n-k} z_k, & S_n(g^2) &:= \sum_{k=0}^{n} \varrho^{n-k}\|\boldsymbol{g}_k\|_{\mathbb{L}_2}^2, \\
S_n(f) &:= \sum_{k=0}^{n} \varrho^{n-k}\mathbb{E}[f_k], & S_n(v^2) &:= \sum_{k=0}^{n} \varrho^{n-k}\|\boldsymbol{v}_k\|_{\mathbb{L}_2}^2.
\end{aligned}$$

We will need the following lemma, the proof of which is postponed.

**Lemma 11.** *For any $\gamma > 0$ and $h > 0$ satisfying $\gamma \geqslant 5M$ and any $\eta \leqslant 1/5$, the iterates of the randomized midpoint discretization of the kinetic Langevin diffusion satisfy*

$$\|\boldsymbol{v}_{n+1}\|_{\mathbb{L}_2}^2 \leqslant (1 - 1.47\eta)\|\boldsymbol{v}_n\|_{\mathbb{L}_2}^2 - 2\bar{\alpha}_2 \eta\mathbb{E}[\boldsymbol{v}_n^\top \boldsymbol{g}_n] + 2\eta^2\|\boldsymbol{g}_n\|_{\mathbb{L}_2}^2 + 2.12\gamma\eta p \tag{52}$$

$$\mathbb{E}[\boldsymbol{v}_{n+1}^\top \boldsymbol{g}_{n+1}] \leqslant 0.51\eta\|\boldsymbol{v}_n\|_{\mathbb{L}_2}^2 + (1 - \bar{\alpha}_2 \eta)\mathbb{E}[\boldsymbol{v}_n^\top \boldsymbol{g}_n] - 0.97\eta\|\boldsymbol{g}_n\|_{\mathbb{L}_2}^2 + 0.9\eta^2\gamma p \tag{53}$$

$$\gamma\mathbb{E}[f_{n+1} - f_n] \leqslant 0.28\eta^2\|\boldsymbol{v}_n\|_{\mathbb{L}_2}^2 + \bar{\alpha}_2 \eta\mathbb{E}[\boldsymbol{v}_n^\top \boldsymbol{g}_n] - 0.46\eta^2\|\boldsymbol{g}_n\|_{\mathbb{L}_2}^2 + 0.09\eta^3\gamma p.$$

From the first inequality equation 52 in Lemma 11, we infer that

$$S_n^{+1}(v^2) \leqslant (1 - 1.47\eta)S_n(v^2) - 2\bar{\alpha}_2 \eta S_n(z) + 2\eta^2 S_n(g^2) + 2.12\gamma^2 p/m.$$

In view of Lemma 3 and the fact that $\|\boldsymbol{v}_0\|_{\mathbb{L}_2}^2 = \gamma p$, this implies that

$$\begin{aligned}
1.47\eta S_n(v^2) + 2\bar{\alpha}_2 \eta S_n(z) &\leqslant S_n(v^2) - S_n^{+1}(v^2) + 2\eta^2 S_n(g^2) + 2.12\gamma^2 p/m \\
&\leqslant (1 - \varrho)S_n(v^2) + 2\eta^2 S_n(g^2) + 2.12\gamma^2 p/m + \gamma p.
\end{aligned}$$

Note that $1 - \varrho \leqslant \frac{m\eta}{\gamma} \leqslant 0.02\eta$. Therefore, we obtain

$$(1.47 - 0.02)S_n(v^2) + 2\bar{\alpha}_2 S_n(z) \leqslant 2\eta S_n(g^2) + \frac{2.14\gamma^2 p}{m\eta},$$

that is equivalent to

$$S_n(v^2) \leqslant 1.38\bar{\alpha}_2 S_n(z)_- + 1.38\eta S_n(g^2) + \frac{1.48\gamma^2 p}{m\eta}. \tag{54}$$

The second step is to use the second inequality equation 53 of Lemma 11. Note that $m\eta/\gamma \leqslant 1/500$ implies $1 - \varrho \geqslant 0.998 m\eta/\gamma$. It then follows that

$$S_n^{+1}(z) = (1 - \bar{\alpha}_2\eta)S_n(z) - 0.97\eta S_n(g^2) + 0.51\eta S_n(v^2) + 0.9\eta\gamma^2 p/m.$$

This inequality, combined with Lemma 3, yields

$$0.97\eta S_n(g^2) \leqslant -(\bar{\alpha}_2\eta + \varrho - 1)S_n(z) + 0.51\eta S_n(v^2) + |z_{n+1}| + 0.9\eta\gamma^2 p/m$$
$$\leqslant \bar{\alpha}_2\eta S_n(z)_- + 0.51\eta S_n(v^2) + |z_{n+1}| + 0.9\eta\gamma^2 p/m.$$

This can be rewritten as

$$S_n(g^2) \leqslant 1.03\bar{\alpha}_2 S_n(z)_- + 0.53 S_n(v^2) + \frac{1.03|z_{n+1}|}{\eta} + \frac{0.93\gamma^2 p}{m}. \tag{55}$$

Let us now proceed with a similar treatment for the last inequality of Lemma 11. Applying Lemma 3, we get $S_n^{+1}(f) \geqslant \varrho S_n(f) - \varrho^{n+1}\mathbb{E}[f_0] \geqslant (1 - m\eta/\gamma)S_n(f) - \varrho^{n+1}\mathbb{E}[f_0]$, which leads to

$$-m\eta S_n(f) \leqslant \varrho^{n+1}\gamma\mathbb{E}[f_0] + 0.28\eta^2 S_n(v^2) + \bar{\alpha}_2\eta S_n(z) - 0.46\eta^2 S_n(g^2) + 0.09\frac{\eta^2\gamma^2 p}{m}.$$

From this inequality, and the Polyak-Lojasievicz condition, one can infer that

$$\bar{\alpha}_2 S_n(z)_- \leqslant \varrho^{n+1}\gamma\mathbb{E}[f_0]/\eta + 0.28\eta S_n(v^2) + (0.5 - 0.46\eta)S_n(g^2) + 0.09\frac{\eta\gamma^2 p}{m}. \tag{56}$$

Combining equation 56 with equation 54, we get

$$S_n(v^2) \leqslant 1.38\Big(\varrho^{n+1}\gamma\mathbb{E}[f_0]/\eta + 0.28\eta S_n(v^2) + (0.5 - 0.46\eta)S_n(g^2) + 0.09\frac{\eta\gamma^2 p}{m}\Big)$$
$$+ 1.38\eta S_n(g^2) + \frac{1.48\gamma^2 p}{m\eta}.$$

Since $\eta \leqslant 0.2$, it follows then

$$S_n(v^2) \leqslant 0.8\Big(S_n(g^2) + \frac{1.8\varrho^n\gamma}{\eta}\mathbb{E}[f_0] + \frac{2\gamma^2 p}{m\eta}\Big). \tag{57}$$

Similarly, combining equation 56 and equation 55, we get

$$S_n(g^2) \leqslant 1.03\Big(\varrho^n\gamma\mathbb{E}[f_0]/\eta + 0.28\eta S_n(v^2) + (0.5 - 0.46\eta)S_n(g^2) + 0.09\frac{\eta\gamma^2 p}{m}\Big)$$
$$+ 0.53 S_n(v^2) + \frac{1.03|z_{n+1}|}{\eta} + \frac{0.93\gamma^2 p}{m}.$$

Since $\eta \leqslant 0.2$, it follows then

$$S_n(g^2) \leqslant 1.05 S_n(v^2) + \frac{1.94\varrho^n\gamma}{\eta}\mathbb{E}[f_0] + \frac{1.94|z_{n+1}|}{\eta} + \frac{0.94\gamma^2 p}{m}. \tag{58}$$

Equations equation 57 and equation 58 together yield

$$S_n(g^2) \leqslant 0.84\Big(S_n(g^2) + \frac{1.8\varrho^n\gamma}{\eta}\mathbb{E}[f_0] + \frac{2\gamma^2 p}{m\eta}\Big) + \frac{1.94\varrho^n\gamma}{\eta}\mathbb{E}[f_0] + \frac{1.94|z_{n+1}|}{\eta} + \frac{0.94\gamma^2 p}{m}$$
$$\leqslant 0.84 S_n(g^2) + \frac{3.46\varrho^n\gamma}{\eta}\mathbb{E}[f_0] + \frac{1.94|z_{n+1}|}{\eta} + \frac{1.78\gamma^2 p}{m\eta}.$$

Hence, we get

$$S_n(g^2) \leqslant \frac{21.7\varrho^n\gamma}{\eta}\,\mathbb{E}[f_0] + \frac{12.2|z_{n+1}|}{\eta} + \frac{11.2\gamma^2 p}{m\eta}.$$

Using once again equation equation 57, we arrive at

$$S_n(v^2) \leqslant 0.8\Big(\frac{21.7\varrho^n\gamma}{\eta}\,\mathbb{E}[f_0] + \frac{12.2|z_{n+1}|}{\eta} + \frac{11.2\gamma^2 p}{m\eta} + \frac{1.8\varrho^{n+1}}{\eta}\,\mathbb{E}[f_0] + \frac{2\gamma^2 p}{m\eta}\Big),$$

which is equivalent to

$$S_n(v^2) \leqslant \frac{18.8\varrho^n\gamma}{\eta}\,\mathbb{E}[f_0] + \frac{9.8|z_{n+1}|}{\eta} + \frac{10.6\gamma^2 p}{m\eta}.$$

To complete the proof of the proposition, it remains to establish the suitable upper bound on $|z_{n+1}|$. To this end, we note that

$$\begin{aligned}
\|\boldsymbol{g}_n\|_{\mathbb{L}_2} &\leqslant M\,\|\boldsymbol{\vartheta}_n - \boldsymbol{L}_{nh}\|_{\mathbb{L}_2} + \sqrt{Mp} \\
&\leqslant 0.2\,\|\gamma(\boldsymbol{\vartheta}_n - \boldsymbol{L}_{nh})\|_{\mathbb{L}_2} + \sqrt{0.2\gamma p} \\
&\leqslant 0.3(x_n + 1.5\sqrt{\gamma p}) \\
\|\boldsymbol{v}_n\|_{\mathbb{L}_2} &\leqslant \|\boldsymbol{v}_n - \boldsymbol{V}_{nh}\|_{\mathbb{L}_2} + \sqrt{\gamma p} \\
&\leqslant \|\boldsymbol{v}_n - \boldsymbol{V}_{nh}\|_{\mathbb{L}_2} + \sqrt{\gamma p} \\
&\leqslant x_n + \sqrt{\gamma p}.
\end{aligned}$$

Then, following the same steps as those used in the proof of the second inequality of Lemma 11, one can infer that

$$\begin{aligned}
|z_{n+1}| &\leqslant |z_n| + 0.97\eta\|\boldsymbol{g}_n\|_{\mathbb{L}_2}^2 + 0.51\eta\|\boldsymbol{v}_n\|_{\mathbb{L}_2}^2 + 0.09\eta^2\gamma p \\
&\leqslant \|\boldsymbol{g}_n\|_{\mathbb{L}_2}\|\boldsymbol{v}_n\|_{\mathbb{L}_2} + 0.1\|\boldsymbol{g}_n\|_{\mathbb{L}_2}^2 + 0.051\|\boldsymbol{v}_n\|_{\mathbb{L}_2}^2 + 0.001\gamma p \\
&\leqslant 1.1\|\boldsymbol{g}_n\|_{\mathbb{L}_2}^2 + 0.301\|\boldsymbol{v}_n\|_{\mathbb{L}_2}^2 + 0.001\gamma p \\
&\leqslant 0.099(x_n + 1.5\sqrt{\gamma p})^2 + 0.301(x_n + 1.1\sqrt{p})^2 \\
&\leqslant 0.4\big(x_n + 1.5\sqrt{p}\big)^2.
\end{aligned}$$

This completes the proof of the proposition. $\qquad\square$

## C.2 PROOF OF THEOREM 2

Let $\boldsymbol{\vartheta}_{n+U}, \boldsymbol{\vartheta}_{n+1}, \boldsymbol{v}_{n+1}$ be the iterates of Algorithm. Let $(\boldsymbol{L}_t, \boldsymbol{V}_t)$ be the kinetic Langevin diffusion, coupled with $(\boldsymbol{\vartheta}_n, \boldsymbol{v}_n)$ through the same Brownian motion and starting from a random point $(\boldsymbol{L}_0, \boldsymbol{V}_0) \propto \exp(-f(\boldsymbol{\theta}) - \frac{1}{2}\|\boldsymbol{v}\|^2)$ such that $\boldsymbol{V}_0 = \boldsymbol{v}_0$. Let $(\boldsymbol{L}_t', \boldsymbol{V}_t')$ be the kinetic Langevin diffusion defined on $[0, h]$ using the same Brownian motion and starting from $(\boldsymbol{\vartheta}_n, \boldsymbol{v}_n)$.

Our goal will be to bound the term $x_n$ defined by

$$x_n = \left\|\mathbf{C}\begin{bmatrix}\boldsymbol{v}_n - \boldsymbol{V}_{nh} \\ \boldsymbol{\vartheta}_n - \boldsymbol{L}_{nh}\end{bmatrix}\right\|_{\mathbb{L}_2} \quad \text{with} \quad \mathbf{C} = \begin{bmatrix}\mathbf{I}_p & \mathbf{0}_p \\ \mathbf{I}_p & \gamma\mathbf{I}_p\end{bmatrix}.$$

To this end, define

$$\bar{\boldsymbol{v}}_{n+1} = \mathbb{E}_U[\boldsymbol{v}_{n+1}], \qquad \bar{\boldsymbol{\vartheta}}_{n+1} = \mathbb{E}_U[\boldsymbol{\vartheta}_{n+1}].$$

Since $(\boldsymbol{V}_{(n+1)h}, \boldsymbol{L}_{(n+1)h})$ are independent of $U$, we have

$$x_{n+1}^2 = \left\|\mathbf{C}\begin{bmatrix}\boldsymbol{v}_{n+1} - \bar{\boldsymbol{v}}_{n+1} \\ \boldsymbol{\vartheta}_{n+1} - \bar{\boldsymbol{\vartheta}}_{n+1}\end{bmatrix}\right\|_{\mathbb{L}_2}^2 + \left\|\mathbf{C}\begin{bmatrix}\bar{\boldsymbol{v}}_{n+1} - \boldsymbol{V}_{(n+1)h} \\ \bar{\boldsymbol{\vartheta}}_{n+1} - \boldsymbol{L}_{(n+1)h}\end{bmatrix}\right\|_{\mathbb{L}_2}^2.$$

Using the triangle inequality and Proposition 1 (See also Proposition 1 from (Dalalyan & Riou-Durand, 2020)), we get

$$\left\| \mathbf{C} \begin{bmatrix} \bar{\boldsymbol{v}}_{n+1} - \boldsymbol{V}_{(n+1)h} \\ \bar{\boldsymbol{\vartheta}}_{n+1} - \boldsymbol{L}_{(n+1)h} \end{bmatrix} \right\|_{\mathbb{L}_2} \leqslant \left\| \mathbf{C} \begin{bmatrix} \bar{\boldsymbol{v}}_{n+1} - \boldsymbol{V}_h' \\ \bar{\boldsymbol{\vartheta}}_{n+1} - \boldsymbol{L}_h' \end{bmatrix} \right\|_{\mathbb{L}_2} + \left\| \mathbf{C} \begin{bmatrix} \bar{\boldsymbol{V}}_h' - \boldsymbol{V}_{(n+1)h} \\ \boldsymbol{L}_h' - \boldsymbol{L}_{(n+1)h} \end{bmatrix} \right\|_{\mathbb{L}_2}$$

$$\leqslant \left\| \mathbf{C} \begin{bmatrix} \bar{\boldsymbol{v}}_{n+1} - \boldsymbol{V}_h' \\ \bar{\boldsymbol{\vartheta}}_{n+1} - \boldsymbol{L}_h' \end{bmatrix} \right\|_{\mathbb{L}_2} + \varrho x_n$$

where $\varrho = e^{-mh}$. Combining these inequalities, we get

$$x_{n+1}^2 \leqslant \left( \varrho x_n + y_{n+1} \right)^2 + z_{n+1}^2$$

where

$$y_{n+1} = \left\| \mathbf{C} \begin{bmatrix} \bar{\boldsymbol{v}}_{n+1} - \boldsymbol{V}_h' \\ \bar{\boldsymbol{\vartheta}}_{n+1} - \boldsymbol{L}_h' \end{bmatrix} \right\|_{\mathbb{L}_2}, \qquad z_{n+1} = \left\| \mathbf{C} \begin{bmatrix} \boldsymbol{v}_{n+1} - \bar{\boldsymbol{v}}_{n+1} \\ \boldsymbol{\vartheta}_{n+1} - \bar{\boldsymbol{\vartheta}}_{n+1} \end{bmatrix} \right\|_{\mathbb{L}_2}^2.$$

This yields[6]

$$x_n \leqslant \varrho^n x_0 + \sum_{k=1}^n \varrho^{n-k} y_k + \left( \sum_{k=1}^n \varrho^{2(n-k)} z_k^2 \right)^{1/2}$$

$$\leqslant \varrho^n x_0 + \left( \frac{1}{1-\varrho} \sum_{k=1}^n \varrho^{n-k} y_k^2 \right)^{1/2} + \left( \sum_{k=1}^n \varrho^{2(n-k)} z_k^2 \right)^{1/2},$$

where the second inequality follows from the Cauchy-Schwarz inequality and the formula of the sum of a geometric progression. Using the fact that $\|\mathbf{C}[a,b]^\top\|^2 = \|a\|^2 + \|a + \gamma b\|^2 \leqslant 3\|a\|^2 + 2\gamma^2\|b\|^2$, we arrive at

$$y_{n+1}^2 \leqslant 3\|\bar{\boldsymbol{v}}_{n+1} - \boldsymbol{V}_h'\|_{\mathbb{L}_2}^2 + 2\gamma^2\|\bar{\boldsymbol{\vartheta}}_{n+1} - \boldsymbol{L}_h'\|_{\mathbb{L}_2}^2,$$

$$z_{n+1}^2 \leqslant 3\|\boldsymbol{v}_{n+1} - \bar{\boldsymbol{v}}_{n+1}\|_{\mathbb{L}_2}^2 + 2\gamma^2\|\boldsymbol{\vartheta}_{n+1} - \bar{\boldsymbol{\vartheta}}_{n+1}\|_{\mathbb{L}_2}^2.$$

We then have

$$x_n \leqslant \varrho^n x_0 + \left( \frac{1.001\gamma}{m\eta} \sum_{k=1}^n \varrho^{n-k} (3\|\bar{\boldsymbol{v}}_k - \boldsymbol{V}_h'\|_{\mathbb{L}_2}^2 + 2\gamma^2\|\bar{\boldsymbol{\vartheta}}_k - \boldsymbol{L}_h'\|_{\mathbb{L}_2}^2) \right)^{1/2}$$

$$+ \left( \sum_{k=1}^n \varrho^{2(n-k)} (3\|\boldsymbol{v}_k - \bar{\boldsymbol{v}}_k\|_{\mathbb{L}_2}^2 + 2\gamma^2\|\boldsymbol{\vartheta}_k - \bar{\boldsymbol{\vartheta}}_k\|_{\mathbb{L}_2}^2) \right)^{1/2}. \tag{59}$$

By Corollary 4, we find

$$\|\bar{\boldsymbol{v}}_k - \boldsymbol{V}_h'\|_{\mathbb{L}_2}^2 \leqslant M_\gamma^4 \eta^8 \left( 0.066\eta\|\boldsymbol{g}_{k-1}\|_{\mathbb{L}_2} + 0.168\|\boldsymbol{v}_{k-1}\|_{\mathbb{L}_2} + 0.137\sqrt{\eta\gamma p} \right)^2$$

$$\leqslant 0.2^3 M_\gamma \eta^8 \times 0.0514 \left( \eta^2\|\boldsymbol{g}_{k-1}\|_{\mathbb{L}_2}^2 + \|\boldsymbol{v}_{k-1}\|_{\mathbb{L}_2}^2 + \eta\gamma p \right),$$

$$\gamma^2\|\bar{\boldsymbol{\vartheta}}_k - \boldsymbol{L}_h'\|_{\mathbb{L}_2}^2 \leqslant M_\gamma^4 \eta^{10} \left( 0.038\eta\|\boldsymbol{g}_{k-1}\|_{\mathbb{L}_2} + 0.098\|\boldsymbol{v}_{k-1}\|_{\mathbb{L}_2} + 0.084\sqrt{\eta\gamma p} \right)^2$$

$$\leqslant 0.2^3 M_\gamma \eta^8 \times 0.0002 \left( \eta^2\|\boldsymbol{g}_{k-1}\|_{\mathbb{L}_2}^2 + \|\boldsymbol{v}_{k-1}\|_{\mathbb{L}_2}^2 + \eta\gamma p \right)$$

$$\|\boldsymbol{v}_k - \bar{\boldsymbol{v}}_k\|_{\mathbb{L}_2}^2 \leqslant \eta^4 \left( 0.591\|\boldsymbol{g}_{k-1}\|_{\mathbb{L}_2} + 0.164\|\boldsymbol{v}_{k-1}\|_{\mathbb{L}_2} + 0.082\sqrt{\eta\gamma p} \right)^2$$

$$\leqslant \eta^4 \times 0.39 \left( \|\boldsymbol{g}_{k-1}\|_{\mathbb{L}_2}^2 + \|\boldsymbol{v}_{k-1}\|_{\mathbb{L}_2}^2 + \eta\gamma p \right),$$

$$\gamma^2\|\boldsymbol{\vartheta}_k - \bar{\boldsymbol{\vartheta}}_k\|_{\mathbb{L}_2}^2 \leqslant \eta^4 \left( 0.578\|\boldsymbol{g}_{k-1}\|_{\mathbb{L}_2} + 0.02\|\boldsymbol{v}_{k-1}\|_{\mathbb{L}_2} + 0.005\sqrt{\eta\gamma p} \right)^2$$

$$\leqslant \eta^4 \times 0.32 \left( \|\boldsymbol{g}_{k-1}\|_{\mathbb{L}_2}^2 + \|\boldsymbol{v}_{k-1}\|_{\mathbb{L}_2}^2 + \eta\gamma p \right)$$

---

[6]One can check by induction, that if for some sequences $x_n, y_n, z_n$ and some $\varrho \in (0,1)$ it holds that $x_{n+1}^2 \leqslant (\varrho x_n + y_{n+1})^2 + z_{n+1}^2$, then necessarily $x_n \leqslant \varrho^n x_0 + \sum_{k=1}^n \varrho^{n-k} y_k + (\sum_{k=1}^n \varrho^{2(n-k)} z_k^2)^{1/2}$ for every $n \in \mathbb{N}$.

Therefore, we infer from equation 59 that

$$x_n \leqslant \varrho^n x_0 + \left( \frac{\gamma}{m\eta} \sum_{k=0}^{n} 0.2^2 M_\gamma \eta^8 \times 0.031 \varrho^{n-k} \left( \eta^2 \|\boldsymbol{g}_k\|_{\mathbb{L}_2}^2 + \|\boldsymbol{v}_k\|_{\mathbb{L}_2}^2 + \eta\gamma p \right) \right)^{1/2}$$
$$+ \left( \sum_{k=0}^{n} 1.82 \eta^4 \varrho^{2(n-k)} \left( \|\boldsymbol{g}_k\|_{\mathbb{L}_2}^2 + \|\boldsymbol{v}_k\|_{\mathbb{L}_2}^2 + \eta\gamma p \right) \right)^{1/2}.$$

From Proposition 3 it then follows that

$$\eta \sum_{k=0}^{n} \varrho^{n-k} \left( \eta^2 \|\boldsymbol{g}_k\|_{\mathbb{L}_2}^2 + \|\boldsymbol{v}_k\|_{\mathbb{L}_2}^2 + \eta\gamma p \right) \leqslant 18.9 \varrho^n \gamma \mathbb{E}[f_0] + 3.97(x_n + 1.5\sqrt{\gamma p})^2 + \frac{10.8\gamma^2 p}{m},$$
$$\eta \sum_{k=0}^{n} \varrho^{2(n-k)} \left( \|\boldsymbol{g}_k\|_{\mathbb{L}_2}^2 + \|\boldsymbol{v}_k\|_{\mathbb{L}_2}^2 + \eta\gamma p \right) \leqslant 40.5 \varrho^n \gamma \mathbb{E}[f_0] + 8.8(x_n + 1.5\sqrt{\gamma p})^2 + \frac{21.9\gamma^2 p}{m}.$$

This yields

$$x_n \leqslant \varrho^n x_0 + 0.036 \eta^3 \sqrt{\kappa} \left( 18.9 \varrho^n \gamma \mathbb{E}[f_0] + 3.97(x_n + 1.5\sqrt{\gamma p})^2 + \frac{10.8\gamma^2 p}{m} \right)^{1/2}$$
$$+ \eta^{3/2} \left( 74 \varrho^n \gamma \mathbb{E}[f_0] + 16(x_n + 1.5\sqrt{\gamma p})^2 + \frac{40\gamma^2 p}{m} \right)^{1/2}$$
$$\leqslant \varrho^n x_0 + (0.072 \eta^3 \sqrt{\kappa} + 4\eta^{3/2}) x_n + (0.16 \eta^3 \sqrt{\kappa} + 8.7 \eta^{3/2}) \sqrt{\varrho^n \gamma \mathbb{E}[f_0]}$$
$$+ 0.12 \eta^3 \gamma \sqrt{\kappa p/m} + 6.4 \eta^{3/2} \gamma \sqrt{p/m}.$$

We assume that $\eta \kappa^{1/6} \leqslant 0.1$, which implies that

$$x_n \leqslant \varrho^n x_0 + 0.072 x_n + 0.16 \sqrt{\varrho^n \gamma \mathbb{E}[f_0]} + 0.12 \eta^3 \gamma \sqrt{\kappa p/m} + 6.4 \eta^{3/2} \gamma \sqrt{p/m}.$$

Rearranging the display leads to

$$x_n \leqslant 1.08 \varrho^n x_0 + 0.18 \sqrt{\varrho^n \gamma \mathbb{E}[f_0]} + 0.12 \eta^3 \gamma \sqrt{\kappa p/m} + 6.9 \eta^{3/2} \gamma \sqrt{p/m}.$$

Finally, we use the fact that $x_0 = \gamma \mathsf{W}_2(\nu_0, \pi)$ and $x_n \geqslant \gamma \mathsf{W}_2(\nu_n, \pi)/\sqrt{2}$ to get the claim of the theorem.

## C.3 Proofs of the technical lemmas

We now provide the proofs of the technical lemmas that we used in this section.

### C.3.1 Proof of Lemma 9

By the definition of $\boldsymbol{\vartheta}_{n+U}$, we have

$$\|\boldsymbol{\vartheta}_{n+U} - \boldsymbol{L}'_{Uh}\| \leqslant \left\| \int_0^{Uh} \left( 1 - e^{-\gamma(Uh-s)} \right) \left( \nabla f(\boldsymbol{\vartheta}_n) - \nabla f(\boldsymbol{L}'_s) \right) \mathrm{d}s \right\|$$
$$\leqslant \int_0^{Uh} \left\| \left( 1 - e^{-\gamma(Uh-s)} \right) \left( \nabla f(\boldsymbol{L}'_0) - \nabla f(\boldsymbol{L}'_s) \right) \right\| \mathrm{d}s$$
$$= Uh \int_0^1 \left( 1 - e^{-U\eta(1-t)} \right) \left\| \nabla f(\boldsymbol{L}'_0) - \nabla f(\boldsymbol{L}'_{Uht}) \right\| \mathrm{d}t$$
$$\leqslant Mh\eta U^2 \int_0^1 (1-t) \left\| \boldsymbol{L}'_0 - \boldsymbol{L}'_{Uht} \right\| \mathrm{d}t,$$

where in the last inequality we have used the Lipschitz property of $\nabla f$ and the inequality $1 - e^{-U\eta(1-t)} \leqslant U\eta(1-t)$. By taking the expectation wrt to $U$, we get

$$
\begin{aligned}
\mathbb{E}_U \|\boldsymbol{\vartheta}_{n+U} - \boldsymbol{L}'_{Uh}\|^2 &\leqslant M^2 h^2 \eta^2 \mathbb{E}_U\left[ U^4 \left\{ \int_0^1 (1-t)\|\boldsymbol{L}'_0 - \boldsymbol{L}'_{Uht}\| \,\mathrm{d}t \right\}^2 \right] \\
&\leqslant \frac{M^2 h^2 \eta^2}{3} \mathbb{E}_U\left[ U^4 \int_0^1 \|\boldsymbol{L}'_0 - \boldsymbol{L}'_{Uht}\|^2 \,\mathrm{d}t \right] \\
&\leqslant \frac{M^2 h^2 \eta^2}{3} \mathbb{E}_U[U^3] \int_0^1 \|\boldsymbol{L}'_0 - \boldsymbol{L}'_{ht}\|^2 \,\mathrm{d}t.
\end{aligned}
$$

Hence, we obtain in view of eq. (42)

$$
\begin{aligned}
\|\boldsymbol{\vartheta}_{n+U} - \boldsymbol{L}'_{Uh}\|^2_{\mathbb{L}_2} &\leqslant \frac{M^2 h^2 \eta^2}{12} \int_0^1 \|\boldsymbol{L}'_0 - \boldsymbol{L}'_{ht}\|^2_{\mathbb{L}_2} \,\mathrm{d}t \\
&\leqslant \frac{M^2 h^2 \eta^2 e^{M_\gamma \eta^2}}{12} \int_0^1 \left( \sqrt{\frac{2\gamma^2 p(ht)^3}{3}} + ht\|\boldsymbol{v}_n\|_{\mathbb{L}_2} + \frac{\gamma(ht)^2}{2}\|\nabla f(\boldsymbol{\vartheta}_n)\|_{\mathbb{L}_2} \right)^2 \mathrm{d}t \\
&\leqslant \frac{\gamma^{-2} M_\gamma^2 \eta^6 e^{M_\gamma \eta^2}}{12} \left\{ \sqrt{(2/3)\eta\gamma p} + \sqrt{1/3}\,\|\boldsymbol{v}_n\|_{\mathbb{L}_2} + \sqrt{0.05}\gamma h\|\nabla f(\boldsymbol{\vartheta}_n)\|_{\mathbb{L}_2} \right\}^2.
\end{aligned}
$$

Taking the square root of the two sides of the inequality, we get the claim of the lemma.

### C.3.2  PROOF OF LEMMA 10

By the definition of $\boldsymbol{\vartheta}_{n+1}$, we have

$$
\begin{aligned}
\|\bar{\boldsymbol{\vartheta}}_{n+1} - \boldsymbol{L}'_h\| &= \left\| \mathbb{E}_U\left[ h(1 - e^{-\gamma(h-Uh)})\nabla f(\boldsymbol{\vartheta}_{n+U}) \right] - \int_0^h (1 - e^{-\gamma(h-s)})\nabla f(\boldsymbol{L}'_s)\,\mathrm{d}s \right\| \\
&= \left\| \mathbb{E}_U\left[ h(1 - e^{-\gamma(h-Uh)})\nabla f_{n+U} \right] - h\mathbb{E}_U\left[ (1 - e^{-\gamma(h-hU)})\nabla f(\boldsymbol{L}'_{Uh}) \right] \right\| \\
&\leqslant h\mathbb{E}_U\left[ (1 - e^{-\gamma(1-U)h})\|\nabla f_{n+U} - \nabla f(\boldsymbol{L}'_{Uh})\| \right] \\
&\leqslant M_\gamma \eta^2 \mathbb{E}_U\left[ (1-U)\|\boldsymbol{\vartheta}_{n+U} - \boldsymbol{L}'_{Uh}\| \right],
\end{aligned}
$$

where in the last inequality follows from the smoothness of function $f$ and the fact that $1 - e^{-\gamma(h-Uh)} \leqslant \gamma(1-U)h$. Using the Cauchy-Schwarz inequality, we get

$$
\begin{aligned}
\|\bar{\boldsymbol{\vartheta}}_{n+1} - \boldsymbol{L}'_h\|^2 &\leqslant M_\gamma^2 \eta^4 \mathbb{E}_U[(1-U)^2]\,\mathbb{E}_U\left[ \|\boldsymbol{\vartheta}_{n+U} - \boldsymbol{L}'_{Uh}\|^2 \right] \\
&= \frac{M_\gamma^2 \eta^4}{3} \mathbb{E}_U\left[ \|\boldsymbol{\vartheta}_{n+U} - \boldsymbol{L}'_{Uh}\|^2 \right].
\end{aligned}
$$

By Lemma 9, we then obtain

$$
\begin{aligned}
\|\bar{\boldsymbol{\vartheta}}_{n+1} - \boldsymbol{L}'_h\|_{\mathbb{L}_2} &\leqslant \frac{M_\gamma \eta^2}{\sqrt{3}} \|\boldsymbol{\vartheta}_{n+U} - \boldsymbol{L}'_{Uh}\|_{\mathbb{L}_2} \\
&\leqslant \frac{M_\gamma^2 \eta^5 e^{M_\gamma \eta^2/2}}{\sqrt{3}\gamma} \left( 0.065\eta\|\boldsymbol{g}_n\|_{\mathbb{L}_2} + (1/6)\|\boldsymbol{v}_n\|_{\mathbb{L}_2} + (1/7)\sqrt{\eta\gamma p} \right).
\end{aligned}
$$

This completes the proof of the first claim.

Using the definition of $\boldsymbol{\vartheta}_{n+1}$, and the fact that the mean minimizes the squared integrated error, we get

$$
\begin{aligned}
\|\boldsymbol{\vartheta}_{n+1} - \bar{\boldsymbol{\vartheta}}_{n+1}\|_{\mathbb{L}_2} &= h\left\| (1 - e^{-\gamma h(1-U)})\nabla f_{n+U} - \mathbb{E}_U\left[ (1 - e^{-\gamma h(1-U)})\nabla f_{n+U} \right] \right\|_{\mathbb{L}_2} \\
&\leqslant h\left\| (1 - e^{-\gamma h(1-U)})\nabla f_{n+U} - \mathbb{E}_U\left[ 1 - e^{-\gamma h(1-U)} \right]\nabla f_n \right\|_{\mathbb{L}_2}.
\end{aligned}
$$

Recall that $\bar{U} = 1 - U$, combining this with the last display and the triangle inequality yields

$$\|\boldsymbol{\vartheta}_{n+1} - \bar{\boldsymbol{\vartheta}}_{n+1}\|_{\mathbb{L}_2} \leqslant h\left\|\left(1 - e^{-\eta\bar{U}}\right)\left(\nabla f_{n+U} - \nabla f_n\right)\right\|_{\mathbb{L}_2} + h\left\|\left(e^{-\eta\bar{U}} - \mathbb{E}[e^{-\eta\bar{U}}]\right)\nabla f_n\right\|_{\mathbb{L}_2}$$
$$\leqslant M_\gamma\eta^2\left\|\bar{U}(\boldsymbol{\vartheta}_{n+U} - \boldsymbol{\vartheta}_n)\right\|_{\mathbb{L}_2} + h\eta\|\bar{U}\|_{\mathbb{L}_2}\|\boldsymbol{g}_n\|_{\mathbb{L}_2}. \tag{60}$$

In view of equation 48, we get

$$\left\|(1-U)(\boldsymbol{\vartheta}_{n+U} - \boldsymbol{\vartheta}_n)\right\|_{\mathbb{L}_2}^2 = \mathbb{E}\left[(1-U)^2\left(\|(\eta/\gamma)(U\bar{\alpha}_1\boldsymbol{v}_n - U^2\eta\bar{\beta}_1\boldsymbol{g}_n)\|^2 + (2/3)U^3\eta^3p/\gamma\right)\right]$$
$$\leqslant \frac{\eta^2\|\boldsymbol{v}_n\|_{\mathbb{L}_2}^2}{15\gamma^2} + \frac{\eta^4\|\boldsymbol{g}_n\|_{\mathbb{L}_2}^2}{210\gamma^2} + \frac{\eta^3p}{90\gamma}.$$

In addition, $\|1 - U\|_{\mathbb{L}_2} = \sqrt{1/3}$. Therefore, we infer from equation 60 that

$$\|\boldsymbol{\vartheta}_{n+1} - \bar{\boldsymbol{\vartheta}}_{n+1}\|_{\mathbb{L}_2} \leqslant \frac{M_\gamma\eta^3}{\gamma}\left(\frac{\|\boldsymbol{v}_n\|_{\mathbb{L}_2}}{\sqrt{15}} + \sqrt{\frac{\eta\gamma p}{90}}\right) + \frac{\eta^2}{\gamma}\left(\frac{M_\gamma\eta^2}{\sqrt{210}} + \frac{1}{\sqrt{3}}\right)\|\boldsymbol{g}_n\|_{\mathbb{L}_2}.$$

Numerical computations complete the proof of the second claim.

By the definition equation 50 of $\boldsymbol{v}_{n+1}$, we have

$$\|\bar{\boldsymbol{v}}_{n+1} - \boldsymbol{V}_h'\|_{\mathbb{L}_2} = \gamma\left\|\mathbb{E}_U\left[he^{-\gamma(h-Uh)}\nabla f_{n+U}\right] - \int_0^h e^{-\gamma(t-s)}\nabla f(\boldsymbol{L}_s')\,\mathrm{d}s\right\|_{\mathbb{L}_2}$$
$$\leqslant \gamma\left\|he^{-\gamma(h-Uh)}\nabla f_{n+U} - he^{-\gamma(h-Uh)}\nabla f(\boldsymbol{L}_{Uh}')\right\|_{\mathbb{L}_2}$$
$$\leqslant M\gamma h\|\boldsymbol{\vartheta}_{n+U} - \boldsymbol{L}_{Uh}'\|_{\mathbb{L}_2}.$$

By Lemma 9, we obtain the third claim of the lemma.

In view of equation 47, and the fact that the expectation minimizes the mean squared error, we have

$$\|\boldsymbol{v}_{n+1} - \bar{\boldsymbol{v}}_{n+1}\|_{\mathbb{L}_2} = \gamma h\left\|e^{-\gamma h(1-U)}\nabla f_{n+U} - \mathbb{E}_U\left[e^{-\gamma h(1-U)}\nabla f_{n+U}\right]\right\|_{\mathbb{L}_2}$$
$$\leqslant \gamma h\left\|e^{-\gamma h(1-U)}\nabla f_{n+U} - \mathbb{E}_U\left[e^{-\gamma h(1-U)}\right]\nabla f_n\right\|_{\mathbb{L}_2}.$$

The last display, the notation $\bar{U} = 1 - U$ and the triangle inequality imply that

$$\|\boldsymbol{v}_{n+1} - \bar{\boldsymbol{v}}_{n+1}\|_{\mathbb{L}_2} \leqslant \gamma h\left\|e^{-\eta\bar{U}}\left(\nabla f_{n+U} - \nabla f_n\right)\right\|_{\mathbb{L}_2} + \gamma h\left\|\left(e^{-\eta\bar{U}} - \mathbb{E}_U[e^{-\eta\bar{U}}]\right)\nabla f_n\right\|_{\mathbb{L}_2}$$
$$\leqslant M\gamma h\left\|\boldsymbol{\vartheta}_{n+U} - \boldsymbol{\vartheta}_n\right\|_{\mathbb{L}_2} + \gamma h\left\|\left(e^{-\eta(1-U)} - 1\right)\nabla f_n\right\|_{\mathbb{L}_2}$$
$$\leqslant M\gamma h\left\|\boldsymbol{\vartheta}_{n+U} - \boldsymbol{\vartheta}_n\right\|_{\mathbb{L}_2} + \frac{\eta^2}{\sqrt{3}}\left\|\boldsymbol{g}_n\right\|_{\mathbb{L}_2}. \tag{61}$$

In view of equation 48, we get

$$\left\|\boldsymbol{\vartheta}_{n+U} - \boldsymbol{\vartheta}_n\right\|_{\mathbb{L}_2}^2 = \mathbb{E}\left[\|(\eta/\gamma)(U\bar{\alpha}_1\boldsymbol{v}_n - U^2\eta\bar{\beta}_1\,\boldsymbol{g}_n)\|^2 + (2/3)\eta^3U^3p/\gamma\right]$$
$$\leqslant \frac{2\eta^2\|\boldsymbol{v}_n\|_{\mathbb{L}_2}^2}{3\gamma^2} + \frac{\eta^4\|\boldsymbol{g}_n\|_{\mathbb{L}_2}^2}{10\gamma^2} + \frac{\eta^3p}{6\gamma}.$$

The last claim of the lemma follows from the previous display and equation 61.

### C.3.3 PROOF OF LEMMA 11

From equation 48, equation 49 and equation 50, it follows that

$$\gamma^2\|\boldsymbol{\vartheta}_{n+U} - \boldsymbol{\vartheta}_n\|_{\mathbb{L}_2}^2 \leqslant \|U\eta\bar{\alpha}_1\boldsymbol{v}_n - (U\eta)^2\bar{\beta}_1\,\boldsymbol{g}_n\|_{\mathbb{L}_2}^2 + (1/6)\eta^3\gamma p$$
$$\leqslant (2\eta^2/3)\|\boldsymbol{v}_n\|_{\mathbb{L}_2}^2 + (\eta^4/10)\|\boldsymbol{g}_n\|_{\mathbb{L}_2}^2 + (\eta^3/6)\gamma p \tag{62}$$

and

$$\gamma\|\boldsymbol{\vartheta}_{n+1} - \boldsymbol{\vartheta}_n\|_{\mathbb{L}_2} \leqslant \eta\|\boldsymbol{v}_n\|_{\mathbb{L}_2} + 0.5\eta^2\|\boldsymbol{g}_{n+U}\|_{\mathbb{L}_2} + \sqrt{(2/3)\eta^3\gamma p}$$
$$\leqslant \eta\|\boldsymbol{v}_n\|_{\mathbb{L}_2} + 0.5\eta^2\|\boldsymbol{g}_n\|_{\mathbb{L}_2} + 0.5M_\gamma\eta^2\gamma\|\boldsymbol{\vartheta}_{n+U} - \boldsymbol{\vartheta}_n\|_{\mathbb{L}_2} + \sqrt{(2/3)\eta^3\gamma p}$$
$$\leqslant 1.001\eta\|\boldsymbol{v}_n\|_{\mathbb{L}_2} + 0.501\eta^2\|\boldsymbol{g}_n\|_{\mathbb{L}_2} + \sqrt{0.67\eta^3\gamma p} \tag{63}$$

where in the last step we have used equation 62 and the fact that $M_\gamma \eta^2/2 \leqslant \eta^2/8 \leqslant 1/200$. A bit more precise computations also yield

$$\gamma\|\boldsymbol{\vartheta}_{n+1} - \boldsymbol{\vartheta}_n\|_{\mathbb{L}_2} \leqslant \left\{(\eta\|\boldsymbol{v}_n\|_{\mathbb{L}_2} + \eta^2\|\bar{\beta}_2\boldsymbol{g}_n\|_{\mathbb{L}_2})^2 + \tfrac{2}{3}\gamma\eta^3 p\right\}^{1/2} + \frac{\eta^2}{2}\|\boldsymbol{g}_{n+U} - \boldsymbol{g}_n\|_{\mathbb{L}_2}$$

$$\leqslant \left\{\left(\eta\|\boldsymbol{v}_n\|_{\mathbb{L}_2} + \tfrac{\eta^2}{\sqrt{3}}\|\boldsymbol{g}_n\|_{\mathbb{L}_2}\right)^2 + \tfrac{2}{3}\gamma\eta^3 p\right\}^{1/2} + \frac{M_\gamma\eta^2\gamma}{2}\|\boldsymbol{\vartheta}_{n+U} - \boldsymbol{\vartheta}_n\|_{\mathbb{L}_2}$$

$$\leqslant \left\{\left(\eta\|\boldsymbol{v}_n\|_{\mathbb{L}_2} + \tfrac{\eta^2}{\sqrt{3}}\|\boldsymbol{g}_n\|_{\mathbb{L}_2}\right)^2 + \tfrac{2}{3}\gamma\eta^3 p\right\}^{1/2} + \tfrac{1}{10}\eta^2\gamma\|\boldsymbol{\vartheta}_{n+U} - \boldsymbol{\vartheta}_n\|_{\mathbb{L}_2}.$$

Taking the squares of this inequality, we get

$$\gamma^2\|\boldsymbol{\vartheta}_{n+1} - \boldsymbol{\vartheta}_n\|_{\mathbb{L}_2}^2 \leqslant \left(\eta\|\boldsymbol{v}_n\|_{\mathbb{L}_2} + \tfrac{\eta^2}{\sqrt{3}}\|\boldsymbol{g}_n\|_{\mathbb{L}_2}\right)^2 + \tfrac{2}{3}\gamma\eta^3 p + \tfrac{1}{100}\eta^4\gamma^2\|\boldsymbol{\vartheta}_{n+U} - \boldsymbol{\vartheta}_n\|_{\mathbb{L}_2}^2$$

$$+ \tfrac{1}{5}\eta^2\gamma\left\{\left(\eta\|\boldsymbol{v}_n\|_{\mathbb{L}_2} + \tfrac{\eta^2}{\sqrt{3}}\|\boldsymbol{g}_n\|_{\mathbb{L}_2}\right)^2 + \tfrac{2}{3}\gamma\eta^3 p\right\}^{1/2}\|\boldsymbol{\vartheta}_{n+U} - \boldsymbol{\vartheta}_n\|_{\mathbb{L}_2}$$

$$\leqslant (1 + \tfrac{1}{10}\eta^2)\eta^2\left\{\left(\|\boldsymbol{v}_n\|_{\mathbb{L}_2} + \tfrac{\eta}{\sqrt{3}}\|\boldsymbol{g}_n\|_{\mathbb{L}_2}\right)^2 + \tfrac{2}{3}\eta\gamma p + \tfrac{1}{10}\gamma^2\|\boldsymbol{\vartheta}_{n+U} - \boldsymbol{\vartheta}_n\|_{\mathbb{L}_2}^2\right\}$$

$$\leqslant 1.01\eta^2\left\{\left(\|\boldsymbol{v}_n\|_{\mathbb{L}_2} + \tfrac{\eta}{\sqrt{3}}\|\boldsymbol{g}_n\|_{\mathbb{L}_2}\right)^2 + \tfrac{2}{3}\eta\gamma p + 0.1\gamma^2\|\boldsymbol{\vartheta}_{n+U} - \boldsymbol{\vartheta}_n\|_{\mathbb{L}_2}^2\right\}$$

$$\leqslant 0.68\eta^2\left(3\|\boldsymbol{v}_n\|_{\mathbb{L}_2}^2 + \eta^2\|\boldsymbol{g}_n\|_{\mathbb{L}_2}^2 + \eta\gamma p\right). \tag{64}$$

This implies that for $\gamma \geqslant 5M$ and $\eta \leqslant 1/5$, we have

$$\|\boldsymbol{v}_{n+1}\|_{\mathbb{L}_2}^2 \leqslant (1 - \bar{\alpha}_2\eta)^2\|\boldsymbol{v}_n\|_{\mathbb{L}_2}^2 - 4\eta(1 - \bar{\alpha}_2\eta)\mathbb{E}[\bar{\beta}_2\boldsymbol{v}_n^\top\boldsymbol{g}_{n+U}] + \eta^2\|\boldsymbol{g}_{n+U}\|_{\mathbb{L}_2}^2 + 2\eta\gamma p$$

$$+ 2\eta\sqrt{2\eta\gamma p}\|\boldsymbol{g}_{n+U} - \boldsymbol{g}_n\|_{\mathbb{L}_2}$$

$$\leqslant (1 - \bar{\alpha}_2\eta)^2\|\boldsymbol{v}_n\|_{\mathbb{L}_2}^2 - 4\eta(1 - \bar{\alpha}_2\eta)\mathbb{E}[\bar{\beta}_2\boldsymbol{v}_n^\top\boldsymbol{g}_{n+U}] + \eta^2\|\boldsymbol{g}_{n+U}\|_{\mathbb{L}_2}^2 + 2\eta\gamma p$$

$$+ 2M_\gamma\eta\sqrt{2\eta\gamma p}\|\gamma(\boldsymbol{\vartheta}_{n+U} - \boldsymbol{\vartheta}_n)\|_{\mathbb{L}_2}$$

$$\leqslant (1 - \eta\bar{\alpha}_2)^2\|\boldsymbol{v}_n\|_{\mathbb{L}_2}^2 - 4\eta(1 - \eta\bar{\alpha}_2)\mathbb{E}[\bar{\beta}_2\boldsymbol{v}_n^\top\boldsymbol{g}_{n+U}] + \eta^2\|\boldsymbol{g}_{n+U}\|_{\mathbb{L}_2}^2 + 2.1\eta\gamma p$$

$$+ 20(M_\gamma\eta)^2\|\gamma(\boldsymbol{\vartheta}_{n+U} - \boldsymbol{\vartheta}_n)\|_{\mathbb{L}_2}^2$$

$$\leqslant (1 - \bar{\alpha}_2\eta)^2\|\boldsymbol{v}_n\|_{\mathbb{L}_2}^2 - 2\bar{\alpha}_2\eta(1 - \eta\bar{\alpha}_2)\mathbb{E}[\boldsymbol{v}_n^\top\boldsymbol{g}_n] + 1.1\eta^2\|\boldsymbol{g}_n\|_{\mathbb{L}_2}^2 + 2.1\eta\gamma p$$

$$+ 2M_\gamma\eta\|\boldsymbol{v}_n\|_{\mathbb{L}_2}\|\gamma(\boldsymbol{\vartheta}_{n+U} - \boldsymbol{\vartheta}_n)\|_{\mathbb{L}_2} + 31(M_\gamma\eta)^2\|\gamma(\boldsymbol{\vartheta}_{n+U} - \boldsymbol{\vartheta}_n)\|_{\mathbb{L}_2}^2$$

$$\leqslant (1 - \bar{\alpha}_2\eta)^2\|\boldsymbol{v}_n\|_{\mathbb{L}_2}^2 - 2\bar{\alpha}_2\eta(1 - \eta\bar{\alpha}_2)\mathbb{E}[\boldsymbol{v}_n^\top\boldsymbol{g}_n] + 1.1\eta^2\|\boldsymbol{g}_n\|_{\mathbb{L}_2}^2 + 2.1\eta\gamma p$$

$$+ 0.2M_\gamma\eta^2\|\boldsymbol{v}_n\|_{\mathbb{L}_2}^2 + 5M_\gamma\|\gamma(\boldsymbol{\vartheta}_{n+U} - \boldsymbol{\vartheta}_n)\|_{\mathbb{L}_2}^2 + 31(M_\gamma\eta)^2\|\gamma(\boldsymbol{\vartheta}_{n+U} - \boldsymbol{\vartheta}_n)\|_{\mathbb{L}_2}^2$$

$$\leqslant (1 - \bar{\alpha}_2\eta)^2\|\boldsymbol{v}_n\|_{\mathbb{L}_2}^2 - 2\bar{\alpha}_2\eta(1 - \bar{\alpha}_2\eta)\mathbb{E}[\boldsymbol{v}_n^\top\boldsymbol{g}_n] + 1.1\eta^2\|\boldsymbol{g}_n\|_{\mathbb{L}_2}^2 + 2.1\eta\gamma p$$

$$+ 0.2M_\gamma\eta^2\|\boldsymbol{v}_n\|_{\mathbb{L}_2}^2 + 5.4M_\gamma\|\gamma(\boldsymbol{\vartheta}_{n+U} - \boldsymbol{\vartheta}_n)\|_{\mathbb{L}_2}^2.$$

Since for $\eta \leqslant 0.2$ we have $\bar{\alpha}_2 \geqslant 0.9$, we get $(1 - \bar{\alpha}_2\eta)^2 + 0.2M_\gamma\eta^2 + 5.4M_\gamma(2\eta^2/3) \leqslant (1 - 0.9\eta)^2 + 0.008\eta + 0.16\eta \leqslant 1 - 1.47\eta$. Therefore,

$$\|\boldsymbol{v}_{n+1}\|_{\mathbb{L}_2}^2 \leqslant (1 - 1.47\eta)\|\boldsymbol{v}_n\|_{\mathbb{L}_2}^2 - 2\bar{\alpha}_2\eta(1 - \eta\bar{\alpha}_2)\mathbb{E}[\boldsymbol{v}_n^\top\boldsymbol{g}_n] + 1.12\eta^2\|\boldsymbol{g}_n\|_{\mathbb{L}_2}^2 + 2.12\eta\gamma p.$$

The next step is to get an upper bound on $\mathbb{E}[\boldsymbol{v}_{n+1}^\top\boldsymbol{g}_{n+1}] - \mathbb{E}[\boldsymbol{v}_n^\top\boldsymbol{g}_n]$ in order to prove equation 53. To this end, we first note that

$$\|\boldsymbol{v}_{n+1}\|_{\mathbb{L}_2} \leqslant \|\boldsymbol{v}_n\|_{\mathbb{L}_2} + \eta\|\boldsymbol{g}_{n+U}\|_{\mathbb{L}_2} + \sqrt{2\eta\gamma p}$$

$$\leqslant \|\boldsymbol{v}_n\|_{\mathbb{L}_2} + \eta\|\boldsymbol{g}_n\|_{\mathbb{L}_2} + M_\gamma\eta\|\gamma(\boldsymbol{\vartheta}_{n+U} - \boldsymbol{\vartheta}_n)\|_{\mathbb{L}_2} + \sqrt{2\eta\gamma p}$$

$$\leqslant 1.004\left(\|\boldsymbol{v}_n\|_{\mathbb{L}_2} + \eta\|\boldsymbol{g}_n\|_{\mathbb{L}_2} + \sqrt{2\eta\gamma p}\right). \tag{65}$$

From equation 63 and equation 65, we also infer that

$$\gamma\|\boldsymbol{v}_{n+1}\|_{\mathbb{L}_2}\|\boldsymbol{\vartheta}_{n+1} - \boldsymbol{\vartheta}_n\|_{\mathbb{L}_2} \leqslant \eta\sqrt{3(1.001^2 + 0.501^2 + 0.34)}\left(\|\boldsymbol{v}_n\|_{\mathbb{L}_2}^2 + \eta^2\|\boldsymbol{g}_n\|_{\mathbb{L}_2}^2 + 2\eta\gamma p\right)$$

$$\leqslant 2.2\eta\left(\|\boldsymbol{v}_n\|_{\mathbb{L}_2}^2 + \eta^2\|\boldsymbol{g}_n\|_{\mathbb{L}_2}^2 + 2\eta\gamma p\right).$$

Therefore, this bound and some elementary computations yield

$$
\begin{aligned}
\mathbb{E}[\boldsymbol{v}_{n+1}^\top \boldsymbol{g}_{n+1}] &\leqslant \mathbb{E}[\boldsymbol{v}_n^\top \boldsymbol{g}_n] + \mathbb{E}[\boldsymbol{v}_{n+1}^\top(\boldsymbol{g}_{n+1} - \boldsymbol{g}_n)] + \mathbb{E}[(\boldsymbol{v}_{n+1} - \boldsymbol{v}_n)^\top \boldsymbol{g}_n] \\
&\leqslant \mathbb{E}[\boldsymbol{v}_n^\top \boldsymbol{g}_n] + M_\gamma \|\boldsymbol{v}_{n+1}\|_{\mathbb{L}_2} \|\gamma(\boldsymbol{\vartheta}_{n+1} - \boldsymbol{\vartheta}_n)\|_{\mathbb{L}_2} - \bar{\alpha}_2 \eta \mathbb{E}[\boldsymbol{v}_n^\top \boldsymbol{g}_n] - \eta \mathbb{E}[\boldsymbol{g}_n^\top \boldsymbol{g}_{n+U}] \\
&\leqslant (1 - \bar{\alpha}_2 \eta)\mathbb{E}[\boldsymbol{v}_n^\top \boldsymbol{g}_n] + M_\gamma \|\boldsymbol{v}_{n+1}\|_{\mathbb{L}_2} \|\gamma(\boldsymbol{\vartheta}_{n+1} - \boldsymbol{\vartheta}_n)\|_{\mathbb{L}_2} - \eta\|\boldsymbol{g}_n\|_{\mathbb{L}_2}^2 \\
&\quad + M_\gamma \eta \|\boldsymbol{g}_n\|_{\mathbb{L}_2} \|\gamma(\boldsymbol{\vartheta}_{n+U} - \boldsymbol{\vartheta}_n)\|_{\mathbb{L}_2} \\
&\leqslant (1 - \bar{\alpha}_2 \eta)\mathbb{E}[\boldsymbol{v}_n^\top \boldsymbol{g}_n] - \eta\|\boldsymbol{g}_n\|_{\mathbb{L}_2}^2 + 2.2 M_\gamma \eta\big(\|\boldsymbol{v}_n\|_{\mathbb{L}_2}^2 + \eta^2\|\boldsymbol{g}_n\|_{\mathbb{L}_2}^2 + 2\eta\gamma p\big) \\
&\quad + M_\gamma \eta^2 \|\boldsymbol{g}_n\|_{\mathbb{L}_2} \big(\tfrac{2}{3}\|\boldsymbol{v}_n\|_{\mathbb{L}_2}^2 + \tfrac{1}{10}\eta^2\|\boldsymbol{g}_n\|_{\mathbb{L}_2}^2 + \tfrac{1}{6}\eta\gamma p\big)^{1/2} \\
&\leqslant (1 - \bar{\alpha}_2 \eta)\mathbb{E}[\boldsymbol{v}_n^\top \boldsymbol{g}_n] - \eta\|\boldsymbol{g}_n\|_{\mathbb{L}_2}^2 + 2.2 M_\gamma \eta\big(\|\boldsymbol{v}_n\|_{\mathbb{L}_2}^2 + \eta^2\|\boldsymbol{g}_n\|_{\mathbb{L}_2}^2 + 2\eta\gamma p\big) \\
&\quad + 0.5 M_\gamma \eta\big(\tfrac{2}{3}\|\boldsymbol{v}_n\|_{\mathbb{L}_2}^2 + \tfrac{11}{10}\eta^2\|\boldsymbol{g}_n\|_{\mathbb{L}_2}^2 + \tfrac{1}{6}\eta\gamma p\big).
\end{aligned}
$$

Grouping the terms, and using the fact that $M_\gamma \eta \leqslant 1/25$, we arrive at

$$
\begin{aligned}
\mathbb{E}[\boldsymbol{v}_{n+1}^\top \boldsymbol{g}_{n+1}] &\leqslant (1 - \bar{\alpha}_2 \eta)\mathbb{E}[\boldsymbol{v}_n^\top \boldsymbol{g}_n] - 0.97\eta\|\boldsymbol{g}_n\|_{\mathbb{L}_2}^2 + 2.54 M_\gamma \eta\|\boldsymbol{v}_n\|_{\mathbb{L}_2}^2 + 4.5 M_\gamma \gamma \eta^2 p \\
&\leqslant (1 - \bar{\alpha}_2 \eta)\mathbb{E}[\boldsymbol{v}_n^\top \boldsymbol{g}_n] - 0.97\eta\|\boldsymbol{g}_n\|_{\mathbb{L}_2}^2 + 0.51\eta\|\boldsymbol{v}_n\|_{\mathbb{L}_2}^2 + 0.9\gamma\eta^2 p.
\end{aligned}
$$

Similarly, using the Lipschitz property of $\nabla f$ and equation 64, we get

$$
\begin{aligned}
\gamma\mathbb{E}[f_{n+1} - f_n] &\leqslant \gamma\mathbb{E}[\boldsymbol{g}_n^\top(\boldsymbol{\vartheta}_{n+1} - \boldsymbol{\vartheta}_n)] + (M_\gamma/2)\|\gamma(\boldsymbol{\vartheta}_{n+1} - \boldsymbol{\vartheta}_n)\|_{\mathbb{L}_2}^2 \\
&= \mathbb{E}[\boldsymbol{g}_n^\top(\bar{\alpha}_2 \eta \boldsymbol{v}_n - \eta^2 \bar{\beta}_2 \boldsymbol{g}_{n+U})] + 0.07\eta^2(3\|\boldsymbol{v}_n\|_{\mathbb{L}_2}^2 + \eta^2\|\boldsymbol{g}_n\|_{\mathbb{L}_2}^2 + \eta\gamma p) \\
&\leqslant \bar{\alpha}_2 \eta\mathbb{E}[\boldsymbol{v}_n^\top \boldsymbol{g}_n] - \eta^2\mathbb{E}[\bar{\beta}_2]\|\boldsymbol{g}_n\|_{\mathbb{L}_2}^2 + 0.2\eta^2\|\boldsymbol{g}_n\|_{\mathbb{L}_2}\|\gamma(\boldsymbol{\vartheta}_{n+U} - \boldsymbol{\vartheta}_n)\|_{\mathbb{L}_2} \\
&\quad + 0.07\eta^2(3\|\boldsymbol{v}_n\|_{\mathbb{L}_2}^2 + \eta^2\|\boldsymbol{g}_n\|_{\mathbb{L}_2}^2 + \eta\gamma p) \\
&\leqslant \bar{\alpha}_2 \eta\mathbb{E}[\boldsymbol{v}_n^\top \boldsymbol{g}_n] - \eta^2\mathbb{E}[\bar{\beta}_2]\|\boldsymbol{g}_n\|_{\mathbb{L}_2}^2 + 0.1\eta^4\|\boldsymbol{g}_n\|_{\mathbb{L}_2}^2 + 0.1\|\gamma(\boldsymbol{\vartheta}_{n+U} - \boldsymbol{\vartheta}_n)\|_{\mathbb{L}_2}^2 \\
&\quad + 0.07\eta^2(3\|\boldsymbol{v}_n\|_{\mathbb{L}_2}^2 + \eta^2\|\boldsymbol{g}_n\|_{\mathbb{L}_2}^2 + \eta\gamma p) \\
&\leqslant \bar{\alpha}_2 \eta\mathbb{E}[\boldsymbol{v}_n^\top \boldsymbol{g}_n] - 0.468\eta^2\|\boldsymbol{g}_n\|_{\mathbb{L}_2}^2 + 0.1\eta^4\|\boldsymbol{g}_n\|_{\mathbb{L}_2}^2 \\
&\quad + 0.07\eta^2\|\boldsymbol{v}_n\|_{\mathbb{L}_2}^2 + 0.01\eta^4\|\boldsymbol{g}_n\|_{\mathbb{L}_2}^2 + 0.02\eta^3\gamma p \\
&\quad + 0.07\eta^2(3\|\boldsymbol{v}_n\|_{\mathbb{L}_2}^2 + \eta^2\|\boldsymbol{g}_n\|_{\mathbb{L}_2}^2 + \eta\gamma p).
\end{aligned}
$$

Grouping the terms, and using the fact that $M_\gamma \eta^2 \leqslant 1/50$, we arrive at

$$
\gamma\mathbb{E}[f_{n+1} - f_n] \leqslant \bar{\alpha}_2 \eta\mathbb{E}[\boldsymbol{v}_n^\top \boldsymbol{g}_n] - 0.46\eta^2\|\boldsymbol{g}_n\|_{\mathbb{L}_2}^2 + 0.28\eta^2\|\boldsymbol{v}_n\|_{\mathbb{L}_2}^2 + 0.09\eta^3\gamma p.
$$

This completes the proof of the lemma.

