$\vartheta_k$ and $v_k$ will refer to $\vartheta_k^{\text{KLMC}}$ and $v_k^{\text{KLMC}}$, respectively. We will also use the following shorthand notation:

$$f_n = f(\vartheta_n), \quad g_n = \nabla f_n = \nabla f(\vartheta_n), \quad \eta = \gamma h, \quad M_\gamma = M/\gamma.$$

The advantage of dealing with $\eta$ instead of $h$ is that the former is scale-free.

Note that the iterates of KLMC satisfy

$$v_{n+1} = (1 - \alpha\eta)v_n - \alpha\eta\, g_n + \sqrt{2\gamma\eta}\,\sigma\xi_n \tag{27}$$

$$\vartheta_{n+1} = \vartheta_n + \gamma^{-1}\eta\big(\alpha v_n - \beta\eta g_n + \sqrt{2\gamma\eta}\,\tilde{\sigma}\bar{\xi}_n\big), \tag{28}$$

where

$$\alpha = \frac{1 - e^{-\eta}}{\eta} \in (0, 1), \qquad \beta = \frac{e^{-\eta} - 1 + \eta}{\eta^2} \in (0, 1/2),$$

$$\sigma^2 = \frac{1 - e^{-2\eta}}{2\eta} \in (0, 1), \qquad \tilde{\sigma}^2 = \frac{2(1 - 2\eta + 2\eta^2 - e^{-2\eta})}{(2\eta)^3} \in (0, 1/3)$$

and $\xi_n, \bar{\xi}_n$ are two $\mathcal{N}_p(\mathbf{0}, \mathbf{I}_p)$-distributed random vectors independent of $(\vartheta_n, v_n)$.

Since we assume throughout this section that $2Mh \leqslant 0.1$, $\gamma \geqslant 2M$ and $\kappa \geqslant 10$, we have

$$\alpha = \frac{1 - \exp(-\eta)}{\eta} \geq 0.95, \quad \text{and} \quad mh = \frac{Mh}{\kappa} \leqslant \frac{Mh}{10} \leqslant \frac{1}{200}.$$

The latter, in particular, implies the following bound for $\varrho$:

$$1 - mh \leqslant \varrho = e^{-mh} \leqslant 1 - 0.99mh = 1 - 0.99m\eta/\gamma. \tag{29}$$

For any sequence $\omega = (\omega_n)_{n\in\mathbb{N}}$ of real numbers, we denote by $S_n(\omega)$ the $\rho$-discounted sum $\sum_{k=0}^n \rho^{n-k}\omega_k$. Below we present a simple lemma for the function $S_n(\cdot)$ that we will use repeatedly in this proof.

**Lemma 3** (Summation by parts). *Suppose $\omega = (\omega_n)_{n\in\mathbb{N}}$ is a sequence of real numbers and define $S_n^{+1}(\omega) := \sum_{k=0}^n \varrho^{n-k}\omega_{k+1}$. Then, the following identity is true*

$$S_n^{+1}(\omega) = \omega_{n+1} - \varrho^{n+1}\omega_0 + \varrho S_n(\omega).$$

*Proof.* The proof is based on simple algebra:

$$S_n^{+1}(\omega) = \omega_{n+1} + \sum_{j=1}^n \varrho^{n-j+1}\omega_j = \omega_{n+1} + \varrho\left(S_n(\omega) - \varrho^n\omega_0\right). \qquad \square$$

### B.1   Exponential mixing of continuous-time kinetic Langevin diffusion

Consider the kinetic Langevin diffusions

$$d\mathbf{L}_t = \mathbf{V}_t\, dt \qquad d\mathbf{V}_t = -\gamma\mathbf{V}_t\, dt - \gamma\nabla f(\mathbf{L}_t)dt + \sqrt{2}\gamma\, d\mathbf{W}_t \tag{30}$$

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

\big\|\big(1 - e^{-\eta\bar{U}}\big)\big(\nabla f_{n+U} - \nabla f_n\big)\big\|_{\mathbb{L}_2} + h\big\|\big(e^{-\eta\bar{U}} - \mathbb{E}[e^{-\eta\bar{U}}]\big)\nabla f_n\big\|_{\mathbb{L}_2}$$
$$\leqslant M_\gamma\eta^2\big\|\bar{U}(\boldsymbol{\vartheta}_{n+U} - \boldsymbol{\vartheta}_n)\big\|_{\mathbb{L}_2} + h\eta\|\bar{U}\|_{\mathbb{L}_2}\|\boldsymbol{g}_n\|_{\mathbb{L}_2}. \tag{60}$$

In view of equation 48, we get

$$\big\|(1-U)(\boldsymbol{\vartheta}_{n+U} - \boldsymbol{\vartheta}_n)\big\|_{\mathbb{L}_2}^2 = \mathbb{E}\Big[(1-U)^2\Big(\|(\eta/\gamma)(U\bar{\alpha}_1\boldsymbol{v}_n - U^2\eta\bar{\beta}_1\boldsymbol{g}_n)\|^2 + (2/3)U^3\eta^3 p/\gamma\Big)\Big]$$
$$\leqslant \frac{\eta^2\|\boldsymbol{v}_n\|_{\mathbb{L}_2}^2}{15\gamma^2} + \frac{\eta^4\|\boldsymbol{g}_n\|_{\mathbb{L}_2}^2}{210\gamma^2} + \frac{\eta^3 p}{90\gamma}.$$

In addition, $\|1 - U\|_{\mathbb{L}_2} = \sqrt{1/3}$. Therefore, we infer from equation 60 that

$$\|\boldsymbol{\vartheta}_{n+1} - \bar{\boldsymbol{\vartheta}}_{n+1}\|_{\mathbb{L}_2} \leqslant \frac{M_\gamma\eta^3}{\gamma}\Big(\frac{\|\boldsymbol{v}_n\|_{\mathbb{L}_2}}{\sqrt{15}} + \sqrt{\frac{\eta\gamma p}{90}}\Big) + \frac{\eta^2}{\gamma}\Big(\frac{M_\gamma\eta^2}{\sqrt{210}} + \frac{1}{\sqrt{3}}\Big)\|\boldsymbol{g}_n\|_{\mathbb{L}_2}.$$

Numerical computations complete the proof of the second claim.

By the definition equation 50 of $\boldsymbol{v}_{n+1}$, we have

$$\|\bar{\boldsymbol{v}}_{n+1} - \boldsymbol{V}_h'\|_{\mathbb{L}_2} = \gamma\Big\|\mathbb{E}_U\big[he^{-\gamma(h-Uh)}\nabla f_{n+U}\big] - \int_0^h e^{-\gamma(t-s)}\nabla f(\boldsymbol{L}_s')\,\mathrm{d}s\Big\|_{\mathbb{L}_2}$$
$$\leqslant \gamma\Big\|he^{-\gamma(h-Uh)}\nabla f_{n+U} - he^{-\gamma(h-Uh)}\nabla f(\boldsymbol{L}_{Uh}')\Big\|_{\mathbb{L}_2}$$
$$\leqslant M\gamma h\|\boldsymbol{\vartheta}_{n+U} - \boldsymbol{L}_{Uh}'\|_{\mathbb{L}_2}.$$

By Lemma 9, we obtain the third claim of the lemma.

In view of equation 47, and the fact that the expectation minimizes the mean squared error, we have

$$\|\boldsymbol{v}_{n+1} - \bar{\boldsymbol{v}}_{n+1}\|_{\mathbb{L}_2} = \gamma h\Big\|e^{-\gamma h(1-U)}\nabla f_{n+U} - \mathbb{E}_U\big[e^{-\gamma h(1-U)}\nabla f_{n+U}\big]\Big\|_{\mathbb{L}_2}$$
$$\leqslant \gamma h\Big\|e^{-\gamma h(1-U)}\nabla f_{n+U} - \mathbb{E}_U\big[e^{-\gamma h(1-U)}\big]\nabla f_n\Big\|_{\mathbb{L}_2}.$$

The last display, the notation $\bar{U} = 1 - U$ and the triangle inequality imply that

$$\|\boldsymbol{v}_{n+1} - \bar{\boldsymbol{v}}_{n+1}\|_{\mathbb{L}_2} \leqslant \gamma h\big\|e^{-\eta\bar{U}}\big(\nabla f_{n+U} - \nabla f_n\big)\big\|_{\mathbb{L}_2} + \gamma h\big\|\big(e^{-\eta\bar{U}} - \mathbb{E}_U[e^{-\eta\bar{U}}]\big)\nabla f_n\big\|_{\mathbb{L}_2}$$
$$\leqslant M\gamma h\big\|\boldsymbol{\vartheta}_{n+U} - \boldsymbol{\vartheta}_n\big\|_{\mathbb{L}_2} + \gamma h\big\|\big(e^{-\eta(1-U)} - 1\big)\nabla f_n\big\|_{\mathbb{L}_2}$$
$$\leqslant M\gamma h\big\|\boldsymbol{\vartheta}_{n+U} - \boldsymbol{\vartheta}_n\big\|_{\mathbb{L}_2} + \frac{\eta^2}{\sqrt{3}}\big\|\boldsymbol{g}_n\big\|_{\