# OpenReview forum: "Langevin Monte Carlo for strongly log-concave distributions: Randomized midpoint revisited"
_ICLR.cc/2024/Conference — ICLR 2024 poster_

### Official Review · Reviewer_gtQ6 · 2023-10-18

**Soundness:** 3 good
**Presentation:** 4 excellent
**Contribution:** 3 good
**Rating:** 6
**Confidence:** 2

**Summary:**

The authors considered Langevin Monte Carlo method in the problem of sampling from a target distribution that has a smooth strongly log-concave density. Specifically, the authors developed a novel proof technique that led to nonasymptotic $W_2$ error bounds for the randomized midpoint method for the Langevin Monte Carlo (RLMC) and the randomized midpoint method for the kinetic Langevin Monte Carlo (RKLMC). The upper bounds are competitive with the best available results for LMC and are free from a term that’s linearly dependent on the sample size. The authors also provided a nonasymptotic $W_2$ error bounds for the kinetic Langevin Monte Carlo (KLMC) algorithm that has an improved dependence on the condition number. Numerical experiments were conducted.

**Strengths:**

The paper is well-organized and clearly written. The authors did a good job discussing backgrounds and intuitions. The proposed error bounds improved upon existing results, and the novel proof technique itself has the potential to be used to re-examine other existing analyses. The paper includes sufficient comparison with and reference to related results/literature. The authors also addressed several limitations.

**Weaknesses:**

A major part of this paper’s contribution is removing the (square root of) $mnh$ for the error bounds of RLMC and RKLMC, yet I’m not clear on how important this is. The removal of this term, claimed by the authors, is an important step toward extending these results to potentials that are not strongly convex. Similarly, in the comments for the result for RKLMC, the authors claimed that not requiring the algorithm to be initialized at the minimizer of the potential is important for extending the method to non-convex potentials. However, as the authors pointed out in the discussion, strong convexity seems to be an essential assumption for these results. Therefore, I’m a bit unclear on the significance of this paper’s contribution.

**Questions:**

Please see the weakness part. In particular, how would the extension to non-convex or non-strongly convex potentials depend on the authors’ proposed methods?

---

> ### Author Response · Authors · 2023-11-22
>
> We thank the reviewer for the valuable feedback. Please find out our response below.
>
> 1. >`A major part of this paper’s contribution is removing the (square root of) $mnh$ for the error bounds of RLMC and RKLMC, yet I’m not clear on how important this is. The removal of this term, claimed by the authors, is an important step toward extending these results to potentials that are not strongly convex. Similarly, in the comments for the result for RKLMC, the authors claimed that not requiring the algorithm to be initialized at the minimizer of the potential is important for extending the method to non-convex potentials.`
>
> Let us provide some additional explanations related to these claims. First, let us explain the importance of removing the time horizon $T = nh$ from the upper bound on the discretization error. For most non-convex potentials, the associated Langevin process is not geometrically ergodic, but it can still be ergodic with a polynomial mixing rate. This implies that the time horizon necessary to achieve $\epsilon$-accuracy for the continuous-time process depends polynomially on $1/\epsilon$ (as opposed to the logarithmic dependence in the geometrically ergodic setting). Hence, removing from the discretization error a factor that is polynomial in $T$ would lead to an improvement of the upper bound on the overall sampling error that scales polynomially in $1/\epsilon$.
>
> Likewise, the insistence on the initial point being at the minimizer is tailored to the (strong) convexity assumption, ensuring the computationally efficient determination of such a minimizer. Clearly, this is not the case when dealing with a non-convex potential function.
>
> Therefore, eliminating the term $T = nh$ and removing the requirement for the initialization to be at the minimizer constitute crucial steps in generalizing the results to the non-convex case.
>
>
> 2. >`However, as the authors pointed out in the discussion, strong convexity seems to be an essential assumption for these results. Therefore, I’m a bit unclear on the significance of this paper’s contribution...
> Please see the weakness part. In particular, how would the extension to non-convex or non-strongly convex potentials depend on the authors’ proposed methods?`
>
> One of the prevalent approaches to relaxing strong convexity is to replace it with some isoperimetric inequalities like LSI (Vempala and Wibisono, 2019) or Poincaré inequality (Chewi et al. 2020). These conditions keep the contractivity of the Langevin diffusion without directly assuming strong convexity.
> To analyze non-strongly convex methods (Dalalyan et al. 2022) and (Karagulyan et al. 2020) have proposed a quadratic penalization which renders the potential strongly convex.
>
> In both cases, having explicit non-asymptotic and most importantly reproducible theoretical guarantees is quintessential to both directions. Thus, we believe that our methods will be relevant also in future work.

---

### Official Review · Reviewer_FzXn · 2023-10-29

**Soundness:** 3 good
**Presentation:** 2 fair
**Contribution:** 2 fair
**Rating:** 5
**Confidence:** 4

**Summary:**

The paper is an in-depth study on the randomized Langevin algorithms, particularly emphasizing the Randomized Midpoint technique in LMC. It is claimed that the bounds are superior to those previously established in the literature.

**Strengths:**

The paper consider widely-used randomized midpoint discretizations, offering notable enhancements over current standards, particularly in terms of condition number dependency and the stability of bounds. A key advantage of these bounds is their explicit numerical constants. The Randomized Midpoint is important  to achieve optimal convergence rates in first-order algorithms. The potential of extending this technique to other domains also presents an intriguing avenue for exploration.

**Weaknesses:**

While the paper presents some intriguing results, most of them primarily offers incremental advancements in the field. It applies the Randomized Midpoint technique to lessen discretization errors, which in turn marginally enhances the rate of convergence. However, the mathematical methods employed are not particularly interesting or surprising.

The empirical evidence looks weak to justify the criticality of the Randomized Midpoint in standard LMC.

**Questions:**

Can you provide any empirical evidence that the current algorithm improves in real world?

---

> ### Author Response · Authors · 2023-11-22
>
> We thank the reviewer for their valuable feedback. Please find our point-by-point response below.
>
> 1. >`While the paper presents some intriguing results, most of them primarily offers incremental advancements in the field. It applies the Randomized Midpoint technique to lessen discretization errors, which in turn marginally enhances the rate of convergence. However, the mathematical methods employed are not particularly interesting or surprising.`
>
> We firmly believe that our results and techniques are valuable. Although the proof technique appears straightforward, it is nontrivial, demanding additional efforts and meticulous handling of intermediate terms (refer to the detailed explanation of display (14) in the submission). Notably, our coherent treatment of the randomized midpoint method yields enhanced convergence guarantees for RLMC and RKLMC, and provides explicit dependence on the initialization and small constants. Furthermore, our technique eliminates the requirement for the algorithm to be initialized at the minimizer of the potential and removes the dependence of the upper bound on the time horizon $T = nh$. We believe that these improvements are crucial for envisioning to extend previously existing results to the case of non-convex potentials. Indeed, for a non-convex potential, it is not realistic to assume that the algorithm is initialized at a minimizer of the potential function. In addition, if the non-convex potential is such that the resulting Langevin process is not geometrically ergodic, then the time horizon necessary to achieve $\epsilon$-accuracy for the continuous-time process depends polynomially on $1/\epsilon$. Hence, removing from the discretization error a factor that is
> polynomial in $T$ would lead to an improvement of the upper bound on the overall sampling error that scales polynomially in $1/\epsilon$.
>
> Furthermore, we demonstrated that our proof technique, applied to the analysis of the KLMC algorithm, leads to an enhanced upper bound on the sampling error compared to the results available in the literature. It is worth noting that many prominent researchers have worked on this problem, and our innovative approach, involving a smart application of the discrete integration by parts argument, improves upon their results.
>
> 2. >`The empirical evidence looks weak to justify the criticality of the Randomized Midpoint in standard LMC.`
>
> We note that the target density considered in the empirical studies satisfies a stricter set of assumptions. However, in this work, we aim to analyze the theoretical convergence in the worst-case scenario. In this context, the extent to which the randomized midpoint method outperforms its vanilla version depends on the properties of the target density.
>
> 3. >`Can you provide any empirical evidence that the current algorithm improves in real world?`
>
> We note that the primary focus of this submission is to present a thorough and refined theoretical analysis of midpoint randomization applied to (kinetic) Langevin Monte Carlo. While we acknowledge that demonstrating a more concrete and tangible real-world application could enhance the practical significance of our findings, delving into the applied aspects of midpoint randomization presents challenges. Factors such as the measurement of Wasserstein distance, estimation of smoothness and strong convexity parameters, etc., may require extensive efforts. Consequently, we regard this avenue as a direction for future research.

---

### Official Review · Reviewer_4YVq · 2023-11-01

**Soundness:** 3 good
**Presentation:** 3 good
**Contribution:** 3 good
**Rating:** 8
**Confidence:** 3

**Summary:**

This paper considers the randomized midpoint discretization for the kinetic Langevin diffusion for sampling from a target distribution with smooth and strongly log-concave density. A non-asymptotic upper bound on the W_2 error of this discretization is obtained. A bound on Euler discretization for the kinetic Langevin process is also obtained.

**Strengths:**

The paper provides strong error bounds on randomized midpoint discretization for the kinetic Langevin diffusion for sampling from a target distribution with smooth and strongly log-concave density. The bounds substantially improve upon earlier results, have explicit constant and transparent reliance on the initialization, and don't require starting at the minimizer of the potential. The proof technique is novel and is based on summation by part.

**Weaknesses:**

It would be nice if some simulations in higher dimensions p could be included in Section 5.

**Questions:**

In the second paragraph after (7), "a close" might be "close"?

Would the qualitative behavior of numerical experiments change when the dimension p increases?

It seems that two notations d and p are used for dimension in Section 5.

---

> ### Author Response · Authors · 2023-11-22
>
> We thank the reviewer for the positive and encouraging feedback.
>
> 1. >`It would be nice if some simulations in higher dimensions p could be included in Section 5.`
>
> We thank the reviewer for raising this question. Calculating the Wasserstein-2 distance in a high dimensional setting poses significant challenges. Furthermore, increasing the dimensionality may implicitly alter the strong convexity and smoothness parameters. This is the rationale behind our focus on low dimensions.
>
>
> 2. Potential typos:
>
> We thank the reviewer for pointing out the typos, and we corrected them in the revision of the paper.

---

> > ### Comment · Reviewer_4YVq · 2023-11-22
> >
> > Thank you for your response and revision!

---

### Official Review · Reviewer_Y55L · 2023-11-07

**Soundness:** 3 good
**Presentation:** 4 excellent
**Contribution:** 2 fair
**Rating:** 6
**Confidence:** 3

**Summary:**

This paper revisits the problem of sampling from strongly log-concave continuous distributions in high dimensions. A classical sampling algorithm for this task is the “Langevin Monte Carlo”  MCMC method which uses Langevin diffusion, a stochastic differential equation (SDE) that models the motion of a particle in a fluid. This paper provides an improved analysis of the randomized discretization scheme called “randomized midpoint method” for the aforementioned SDE. The results rely on the assumption that the magnitude of the eigenvalues of the potential function (i.e logarithm of the distribution’s density function) is both upper and lower bounded, while the ratio $\kappa$ between these upper and lower bounds appears in the bounds presented. Specifically, faster convergence of the randomized midpoint method for Langevin diffusion is shown for sufficiently small ratio $\kappa$. Similarly, the authors analyze the randomized midpoint method for the kinetic Langevin Monte Carlo method to obtain improved bounds, although with some slightly stronger conditions and with the advantage of not having to find a minimizer of the potential function for initialization.

**Strengths:**

This is a well written paper that provides improved results on existing algorithms for Langevin sampling, which can be potentially applicable in a wide range of problems.

**Weaknesses:**

Some of the results are potentially still not optimal as the authors also suggest. It would also be nice to see how the improved bounds can be applied to some more concrete problems even for theoretical results.


Minor comments:
-Page 2, line 8: “strongly”->”strong”
-Page 2, line 22: “at”->”a”
-Page 2, “notation” paragraph, line 4: “semi-definite positive”->”positive semi-definite”
-Page 6, line 5: “designatex”->”designate”

**Questions:**

Do you an example application of the improved analysis do get better results for a specific problem?

---

> ### Author Response · Authors · 2023-11-22
>
> We thank the reviewer for their valuable feedback. Please find our response below.
>
> 1. >`Some of the results are potentially still not optimal as the authors also suggest. `
>
> While our bound offers the best-known dependence on the condition number for Kinetic Langevin Monte Carlo, it's essential to note that the optimal dependence remains an open question.
>
> 2. >`It would also be nice to see how the improved bounds can be applied to some more concrete problems even for theoretical results...
> Do you an example application of the improved analysis do get better results for a specific problem?`
>
> The primary focus of our work is to present a thorough and refined theoretical analysis of midpoint randomization applied to Langevin Monte Carlo. While we acknowledge that demonstrating a more concrete and tangible real-world application could enhance the practical significance of our findings, delving into the applied aspects of midpoint randomization presents challenges. Factors such as the calculation of Wasserstein distance in high dimension, measurement of the distributions of the samples and the true underlying distribution, estimation of smoothness and strong convexity parameters, etc., may require extensive efforts. Consequently, we regard this avenue as a direction for future research.
>
>
> 3. Potential typos:
>
> We thank the reviewer for pointing out the typos, and we corrected them in the revision of the paper.

---

### Meta-Review · Area_Chair_Ccnq · 2023-12-11

**Metareview:**

The manuscript analyzes the randomized midpoint method for strongly log-concave distributions. Compared with previous analysis, the work helps further clarify the underlying principle of the methods; on the other hand, the contribution is somewhat incremental, as the RMM has been analyzed in a few earlier works. Overall the paper is solid and hence the AE recommends acceptance.

**Justification For Why Not Higher Score:**

The contribution is somewhat incremental as the randomized midpoint method has been analyzed by a few papers already.

**Justification For Why Not Lower Score:**

The paper is solid and the new analysis offers some further insights into the algorithm.

---

### Decision · Program_Chairs · 2024-01-16

Accept (poster)